# Blocking cancer-fibroblast mutualism inhibits proliferation of endocrine therapy resistant breast cancer

Jason I Griffiths [1,2,6✉], Feng Chi[1,6], Elena Farmaki[1], Eric F Medina[1], Patrick A Cosgrove [1], Kimya L Karimi [1], Jinfeng Chen[1], Vince K Grolmusz [1], Frederick R Adler[2,3], Qamar J Khan[4], Aritro Nath[1], Jeffrey T Chang [5] & Andrea H Bild [1✉]

## Abstract

In early-stage estrogen receptor-positive (ER + ) breast cancer, resistance to endocrine therapy (ET) and CDK4/6 inhibitors (CDK4/6i) often involve a shift away from estrogen-driven proliferation. The nature and source of compensatory growth signals driving cancer proliferation remain unknown but represent direct therapeutic targets of resistant cells. By analyzing single-cell RNA-sequencing data from serial biopsies of patient tumors, we elucidated compensatory growth signaling pathways activated in ET + CDK4/6i-resistant cancer cells, along with the intercellular growth signal communications within the tumor microenvironment. In most patient tumors, resistant cancer cells increased ERBB growth pathway activity during treatment, only partially through ERBB receptor upregulation. Concurrently, fibroblasts within the tumor increased ERBB ligand communication with cancer cells, as they differentiated to a proliferative and mesenchymal phenotype in response to TGFβ signals from cancer cells. In vitro model systems demonstrated molecularly how therapy induces a mutualistic cycle of crosstalk between cancer cells and fibroblasts, fostering a growth factor-rich tumor microenvironment circumventing estrogen reliance. We show that ERBB inhibition can break this cancer-fibroblasts mutualism, targeting an acquired sensitivity of resistant cancer cells.

**Keywords** Endocrine Resistance; Cancer-fibroblast-mutualism; ERBB Signaling; TME-communication
**Subject Categories** Cancer; Chromatin, Transcription & Genomics

## Introduction

Hormone receptor-positive (estrogen receptor-positive (ER+) and/or progesterone receptor-positive (PR+)) breast cancer constitute 70–80% of breast malignancies (Rozeboom et al, 2019). In ER+ breast cancer cells, estrogen receptor overexpression stimulates cell cycle progression through cyclin-dependent kinase 4 and 6 (CDK4/6) activation which drives cancer cell proliferation. Primary treatments for these tumors are endocrine therapies that either deplete endogenous and locally metabolized estrogen using aromatase-inhibition (AI), or drugs that block ER activity through direct modulation or degradation (Rani et al, 2019; Zhao et al, 2016; Jameera Begam et al, 2017). Endocrine therapy resistance emerges after treatment in the majority of metastatic breast cancer tumors and at least 33% of early-stage (II/III) non-metastatic tumors (Osborne, 1998; Early Breast Cancer Trialists' Collaborative Group (EBCTCG), 2005). Combining endocrine therapy with CDK4/6 inhibitors has improved metastatic ER+ breast cancer disease control, significantly extending progression-free and overall survival (Piezzo et al, 2020). However, this combination has shown variable efficacy for treating patients with early-stage disease across the MonachE, PALLAS, Penelope-B, and FELINE clinical trials (Johnston et al, 2020; Mayer et al, 2021; Loibl et al, 2021; Khan et al, 2020). To uncover cancer cell-intrinsic or tumor microenvironment (TME) mechanisms conferring resistance to endocrine and CDK4/6 inhibition therapies in earlier-stage patient tumors, detailed molecular interrogation of serial biopsies across patient cohorts is needed (Hanker et al, 2020; Watt and Goel, 2022).

In a prior study of a smaller cohort of early-stage ER+ breast cancer patient tumors, we revealed that cancer resistance to endocrine and CDK4/6 inhibition therapy involves cancer cells shifting from estrogen-mediated signaling to alternative growth signal-mediated proliferation (Griffiths et al, 2021). Furthermore, we showed that cancer cells exhibited cell cycle reactivation during treatment, as measured clinically by KI-67 antigen expression, tumor size trajectories and through transcriptional changes in cell

[1]Department of Medical Oncology & Therapeutics, City of Hope National Medical Center, 1500 East Duarte Road, Duarte, CA 91010, USA. [2]Department of Mathematics, University of Utah 155 South 1400 East, Salt Lake City, UT 84112, USA. [3]School of Biological Sciences, University of Utah 257 South 1400 East, Salt Lake City, UT 84112, USA. [4]Division of Medical Oncology, Department of Internal Medicine, The University of Kansas Medical Center, Kansas City, KS 66160, USA. [5]Department of Integrative Biology and Pharmacology, School of Medicine, School of Biomedical Informatics, UT Health Science Center at Houston, Houston, TX 77030, USA. [6]These authors contributed equally: Jason I Griffiths, Feng Chi. ✉E-mail: jasonigriff@gmail.com; jgriff@coh.org; abild@coh.org

cycle signaling. This analysis revealed that post-treatment cancer cells are resistant and have overcome endocrine therapy-induced cytostatic effects through cell cycle reactivation, as opposed to being cytostatic persister cells. The alternative growth signals driving reactivation of proliferation represent direct therapeutic targets to overcome resistance (Dominiak et al, 2020). Cancer cell proliferation can be stimulated by a variety of growth-promoting receptors of exogenous hormones (e.g., progesterone, androgen and insulin) and locally produced growth factors (GF) such as neuregulins (NRGs), epidermal growth factor (EGF), insulin-like growth factors (IGFs), fibroblast growth factors (FGFs), and transforming growth factor β members (TGFβ) (Almaraz Postigo and Montero, 2023; Masuda et al, 2012; Christopoulos et al, 2015; Santolla and Maggiolini, 2020; Kretzschmar, 2000). A substantial literature exists on the oncogenic properties of each of these growth pathways and numerous drugs exist to target them, though with inconsistent clinical efficacy (Navid et al, 2020; Biswas et al, 2007; Ekyalongo and Yee, 2017). Despite their documented role in cancer proliferation, it is currently unclear which pathways contribute to GF signal-mediated proliferation.

In addition to upregulating growth receptors, cancer cells can also produce cognate GF ligands and hormone signals to stimulate proliferation through autocrine and/or paracrine signaling (e.g., Emond et al, 2023). Cancer-associated fibroblasts, macrophages and endothelial cells can also provide growth signals that contribute to cancer growth (Chatterjee et al, 2019; Lee et al, 2014; Castellaro et al, 2019). Identifying the primary sources of growth signals may reveal targets to block endocrine resistance. Yet, little is known about the intercellular signaling shaping endocrine and CDK4/6 inhibitor resistance in patient tumors as they progress on endocrine therapy (see Griffiths et al, 2025).

It is increasingly appreciated that cancer cells engineer the TME to promote cancer growth through local interactions with non-cancer cell types (Martinez-Outschoorn et al, 2019; Ren et al, 2018; Heppner and Shekhar, 2014). Mutually beneficial mutualistic interactions are of particular concern when they enhance proliferation of both cancer and pro-tumor non-cancer cell types by generating an oncogenic positive feedforward loop (Wu et al, 2019). Ecological theory predicts that such oncogenic mutualisms may become stronger under more stressful treatment conditions (Bertness and Callaway, 1994; Hammarlund and Harcombe, 2019; Michalet et al, 2014). Such unintended treatment consequences could more rapidly induce resistance.

We hypothesize that endocrine therapy can stimulate oncogenic mutualisms within tumors, fostering compensatory GF signaling that fuels the proliferation of resistant cells. To investigate the mechanisms underlying endocrine and CDK4/6 inhibitor resistance in early-stage ER+ breast cancer cells, we employ a hybrid computational and experimental strategy. This approach integrates patient-derived single-cell RNA sequencing to explore acquired resistant phenotypes and intercellular crosstalk within tumors as patients undergo therapy and utilizes cell-based assays to assess mechanisms of resistance and interactions between cancerous and non-cancerous cells. This method enables detailed insights into critical resistance signaling pathways and new therapy approaches. We utilize a dataset of 424,581 single cells, including both cancer and non-cancer cells, serially collected from 173 tumor biopsies from 62 patients taken prior to, during and after treatment. Across

patient tumors, resistant cancer cells consistently converge on a proliferation phenotype driven by ERBB pathway activation, associated with intensified ligand–receptor (LR) communication between cancer and fibroblast cells. Experimentally, we confirmed that endocrine therapy induces the upregulation of compensatory ERBB receptors in cancer cells. In addition, under therapy cancer cells stimulate fibroblast differentiation to a more mesenchymal and proliferative state, leading to increased ERBB ligand provision. Patient-derived insights, validated across independent cohorts, and multiple in vitro model systems of endocrine and ribociclib resistance support the role of ERBB signaling activation in ER+ breast cancer growth during endocrine therapy. Experiments also support the ecological prediction that the oncogenic mutualism between cancer and fibroblasts intensifies under treatment. Experimentally blocking this mutualism, by inhibiting the ERBB pathway, targets GF signal-mediated proliferation and effectively controls cancer growth.

# Results

## Overview of patient cohort, patient genomic profiling, and experimental studies

We studied tumors from postmenopausal women with node-positive or >2 cm ER+ and/or PR +, HER2- breast cancer enrolled on the FELINE clinical trial (clinicaltrials.gov # NCT02712723). This trial assessed the efficacy of combining endocrine therapy with CDK4/6 inhibition in the neoadjuvant setting (Fig. 1, top left). Patients ($n = 120$) were randomized to receive either endocrine therapy alone (letrozole alone= letrozole + placebo) ($n = 40$ patients) or in combination with CDK4/6 inhibition (combination ribociclib= ribociclib + letrozole) ($n = 80$ patients). Patients were treated for 6 months and biopsies were collected at pre-treatment (day 0), early in treatment at follow-up (day 14), and post treatment (surgery around day 180). Biopsies from half of the patients in each treatment arm were previously randomly assigned to a hypothesis-generating discovery cohort (Griffiths et al, 2021). The remaining samples were designated for use as a validation cohort to replicate and independently verify key observations of the discovery cohort.

Independently for each patient cohort, scRNAseq (10X platform) was performed on serially collected tumor samples following consistent procedures (detailed in Griffiths et al, 2021). Stringent quality control ensured high coverage, low mitochondrial content, and lack of doublets. We obtained high-quality transcriptomic profiles of 424,581 single cells (41% discovery cohort, 59% validation cohort) from 173 serial patient tumor biopsies. Cancer and non-cancer cells were serially sampled from pre- and post-treatment biopsies of 62 patient tumors. Of these tumors, 35 were from the discovery cohort patients (23 received combination ribociclib; 12 received letrozole alone), and 27 were from validation cohort patients (16 received combination ribociclib; 11 received letrozole alone). High-quality scRNAseq transcriptional profiles were used for the analysis of cell type, phenotype, communication, and composition. Broad cell types were annotated using singleR (Aran et al, 2019), cancer cells were identified by their copy number amplification using inferCNV (Tickle et al, 2019) and granular immune subtypes were determined using ImmClassifier (Liu et al,

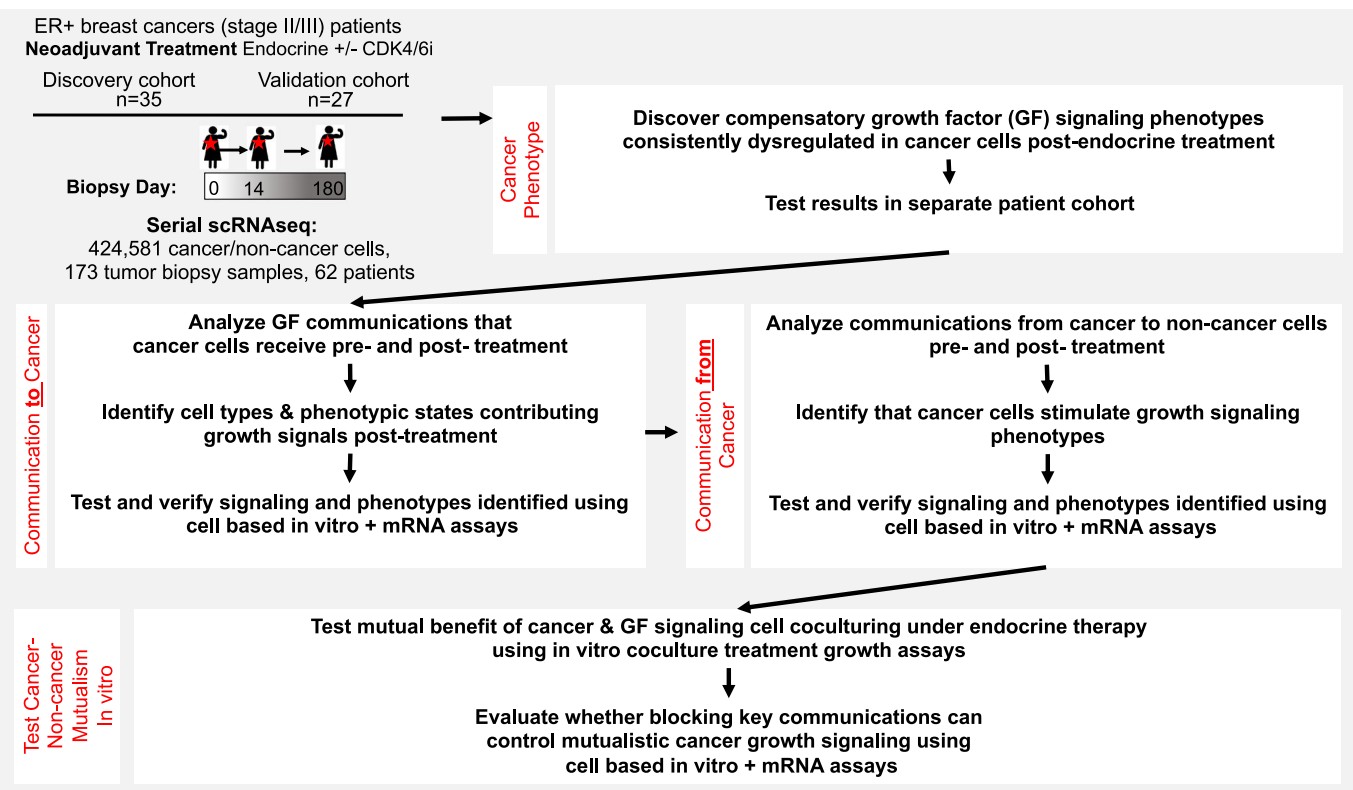

**Figure 1. Approach to understand compensatory growth factor (GF) signal-mediated proliferation in endocrine therapy-resistant ER+ breast cancer using scRNAseq analysis of serially sampled tumor biopsies of clinical patient cohorts.**

Tumor biopsies were provided by patients (*n* = 62) that all received neoadjuvant endocrine +/− CDK4/6 inhibitor treatment for 180 days, with samples taken on day 0 (pre-treatment), day 14 (early follow-up) and day 180 (post treatment). Patients were randomly stratified between a discovery and validation cohort (Discover= 35 patients, 12 letrozole alone + 23 combination ribociclib; Validation=27 patients, 11 letrozole alone + 16 combination ribociclib). Multiple timepoint scRNAseq data was obtained from the serial tumor biopsies and used to analyze the cell-type composition and within cell-type phenotypic heterogeneity of each tumor sample (see "Methods"). Compensatory GF signaling phenotypes were detected in the endocrine +/− CDK4/6i-resistant cancer cells remaining post treatment across patients. For 41 patients (*N* = 16 discovery + 25 validation) with cancer cells (*n* > 20 cells) sequenced in paired pre- (day 0) and post treatment (day 180) tumor biopsies, ssGSEA scores quantified cancer phenotypes. Hierarchical random regression detected pathways consistently dysregulated in the resistant cells observed post treatment compared to the pre-treatment sensitive cancer cells of that tumor. The model structure accounted for: (i) pre-existing phenotype differences in pathway activity among patient tumors and (ii) tumor-specific phenotypic response to therapy. Resistant cancer phenotypes, identified in the discovery cohort, were then verified in the independent validation cohort. Communications supporting cancer cell proliferation under endocrine treatment were measured from serial scRNAseq data. The ligand–receptor communication analysis measured networks of signaling between the phenotypically diverse populations of cancer and non-cancer cell types constituting each tumor. Phenotypically diverse subpopulations of each cell type (e.g. fibroblasts, endothelial, myeloid, diploid epithelial, cancer and T cells) were resolved using UMAP analysis and enumerated within each tumor sample. Ligand–receptor (LR) communication pathways (*n* = 1444) were defined by known protein–protein interactions (LR communication database of Ramilowski et al, 2015). The strength of communication between cell types were quantified by the contribution of cells in each subpopulation to the total ligand signal production in the tumor and the corresponding cognate receptor expression of the receiving cell (see "Methods"). We analyzed the total strength of GF communications each cancer cell received from the many phenotypically diverse subpopulations of cells constituting the tumor. The networks of GF communication across tumors identified cell types predominantly contributing supplementary GF signals to cancer cells following endocrine therapy. We then experimentally tested patient-derived predictions about the GF communications supporting cancer proliferation under endocrine treatment. Experimental evolution of drug-resistant ER+ breast cancer cell lines in vitro yielded three endocrine therapy-resistant (ETR) lines and three paired combination ribociclib-resistant (CRR) lines. We tested whether expected resistance phenotypes emerged upon endocrine and CDK4/6 inhibition treatment by measuring gene expression and signal transduction through protein phosphorylation using qPCR and western blot assays. Using 3D spheroid coculture experiments with fluorescently labeled cancer and non-cancer cell types, we tested whether predicted cell types and GF communications facilitated cancer proliferation. We then further analyzed communication networks to identify the key cell populations stimulating GF signaling, with cancer cells being identified as a dominant supplier of these stimuli. We tested these predictions using in vitro assays with cancer and non-cancer cells. We verified that cancer cells drive phenotypic differentiation and activation of GF signaling in predicted cell types, using combined qPCR and imaging experiments. By modulating the presence/abundance of cancer and GF signaling cell types and measuring each population's growth using serial imaging, we uncovered the mutualistic interaction by which both cell types enhance the growth of one another during endocrine treatment. Finally, we evaluated how inhibiting the communications driving GF signal-mediated proliferation of cancer cells can block oncogenic mutualisms and control spheroid growth/morphological development.

2021). Cell-type-specific marker gene expression and UMAP/TSNE analyses verified annotations (Becht et al, 2018).

Examination of the phenotypic evolution of cancer cells from the initial discovery cohort of patients has shown that post-treatment cancer cells are resistant and have overcome treatment-induced cytostatic effects through cell cycle reactivation, as opposed to being cytostatic persister cells (Griffiths et al, 2021). To uncover the compensatory GF signal-mediated proliferation phenotype of these endocrine + CDK4/6i-resistant cancer cells, we profiled tumors of a much larger cohort of patients (the discovery and

validation cohorts). Using this combined (discovery + validation cohort) dataset, we analyzed changes in single-cell pathway activity between cancer cells of pre-treatment tumor biopsies and the remnant cancer cells in post-treatment tumor biopsies from 41 matched tumor samples (day 0 versus 180; $n > 20$ cancer cells). Of these, 17 patients were treated with combination ribociclib, and 24 patients received letrozole alone.

Consistent dysregulation of growth factor signaling in resistant cancer cells during treatments across patients was determined using single sample Gene Set Enrichment Analysis (ssGSEA) scores (Liberzon et al, 2011; Hänzelmann et al, 2013) (Fig. 1, top right). This was analyzed using hierarchical random regression (see "Methods": "Identifying consistent resistance phenotypes across tumors") to account for the paired pre- and post-treatment tumor sampling and the shared history of cells within each tumor (Krzywinski et al, 2014; Henderson, 1982; Bates et al, 2015). Pathways dysregulated in resistant cancer cells were identified in the discovery cohort and tested in the validation cohort.

To decipher the cell types and signals that drive resistant phenotypes, we inferred ligand–receptor signaling using the scRNAseq ligand–receptor data (Fig. 1, middle) (Kumar et al, 2018; Armingol et al, 2021; Griffiths et al, 2025) (see "Methods": "Deciphering cell type communications within patient tumors"). This analysis revealed how both phenotypic and compositional changes within tumors during treatment impacted the strength of ligand–receptor communications received by individual cancer or non-cancer cells by following changes in ligand contribution into the TME and activation of corresponding receptors. This approach identified the cell types providing the majority of GF signals to support proliferation in post-treatment resistant cancer cells (Fig. 1, middle left) and uncovered how these facilitatory GF signaling cells were activated (Fig. 1, middle right).

To experimentally test patient-derived findings, we developed six in vitro 3D spheroid models of acquired endocrine resistance or combination endocrine plus cell cycle therapy resistance. We selected three ER+ breast cancer cell lines (CAMA-1, MCF-7, and T47D) with above-average endocrine resistance, based on proliferation-corrected endocrine sensitivity estimates from a published large-scale drug screen (Hafner et al, 2017) (Data ref: Hafner 2017) (Appendix Fig. S1). We refer to these parental cell lines with relatively low endocrine sensitivity as having endocrine therapy resistance (ETR; parental lines). We then used long-term experimental evolution of each cell line under ribociclib to develop three paired cancer cell lines with joint endocrine and CDK4/6i resistance (evolved lines with combination ribociclib resistance (CRR)) (Grolmusz et al, 2020; Emond et al, 2023). Although each parental cell line already exhibited considerable resistance to endocrine therapy (4-hydroxytamoxifen and Letrozole), ribociclib selection pressure further increased endocrine resistance to fulvestrant (Appendix Fig. S2). These cell lines were selected to reflect the post-treatment cancer cells resistant to endocrine +/− CDK4/6i therapy experienced during the FELINE trial. We expected that across the replicate model systems and resistance states, the cell lines should exhibit the endocrine + CDK4/6i resistance phenotype observed in patient tumors. For each of the three paired cancer cell lines, we used gene expression and protein phosphorylation assays to test whether compensatory GF signal-mediated proliferation phenotypes (predicted from scRNAseq analyses of patient tumors) emerged in vitro under endocrine

and CDK4/6i treatment. Further, using coculture experiments, we tested the role of hypothesized cell-cell communications with non-cancer cells (fibroblasts) predicted to drive GF signal-mediated proliferation (Fig. 1, bottom). By modulating the presence and abundance of cancer and non-cancer cell types and measuring each population's growth using serial imaging, we tested the mutualistic interactions predicted to enhance growth under treatment.

Finally, we assessed the therapeutic efficacy of inhibiting the identified mechanisms of GF signal-mediated proliferation to: (i) target the resistance phenotype of persistent cancer cells observed in patient tumors and (ii) block growth-promoting cancer-non-cancer cell communications underpinning an oncogenic mutualism that fuels cancer cell growth. We used serial spheroid imaging to measure the growth of cancer monocultures and cancer-non-cancer cocultures with and without targeted inhibition to assess control of cancer growth during treatment.

### Endocrine and CDK4/6i-resistant cells upregulate the ERBB pathway during treatment, driving GF signal-mediated proliferation

We first identified cellular pathways consistently dysregulated in resistant cancer cells persisting at end of treatment (EOT) across the cohorts of patient tumors, through analysis of ssGSEA enrichment scores of gene sets in the Molecular Signatures Database (MSigDB; C2 and hallmark pathways). Pathway gene sets were identified that were recurrently dysregulated in cancer cells of post-treatment tumor samples compared to cancer cells of patient-matched pre-treatment tumor samples, using hierarchical random regression (Fig. EV1A). Most pathways were identified to be linked to ERBB signaling and within a cell these pathways were highly correlated (Fig. EV1B). We therefore constructed a composite pathway score to measure ERBB activity across these pathways (see "Methods": "Measuring composite phenotypes scores").

Activation of the ERBB growth factor signaling pathway was identified in post-treatment resistant cancer cells in the majority of patient tumors across treatment arms (73% of patients (30/41) showed significant activation in post-treatment versus pre-treatment paired cancer samples). This result was consistent across patient tumors in both the discovery (Fig. 2A, top panel) and validation cohort (Fig. 2A, bottom panel) (Tables EV1 and EV2; Fig. EV1C,D). Activation of the ERBB signaling pathway was also observed to a lesser extent in diploid epithelial cells of these post-treatment tumor samples (Fig. EV1C).

Differential expression of individual genes (versus the pathway level sets of genes above) between pre-treatment (naive) and post-treatment (resistant) cancer cells was also analyzed using a hierarchical regression approach and expression changes were then confirmed in the independently profiled validation cohort (Fig. EV1D). This analysis revealed that resistant cells upregulated transcription factors downstream of the ERBB signaling pathway including AP-1 components, FOS, FOSB, JUN, and JUND, and also SOX9 and EHF; all of which are known promoters of cancer proliferation (Wee and Wang, 2017; LaMuraglia et al, 1988). Activation of transcription factors downstream of ERBB signaling was observed in tumors from both the discovery and validation cohort (Table EV3). Similarly, cancer cells remaining after 6 months of therapy in both discovery and validation cohort tumors lost expression of cyclin D (CCND1) and genes involved in estrogen-dependent growth (e.g., PGR and MAPT are genes associated with

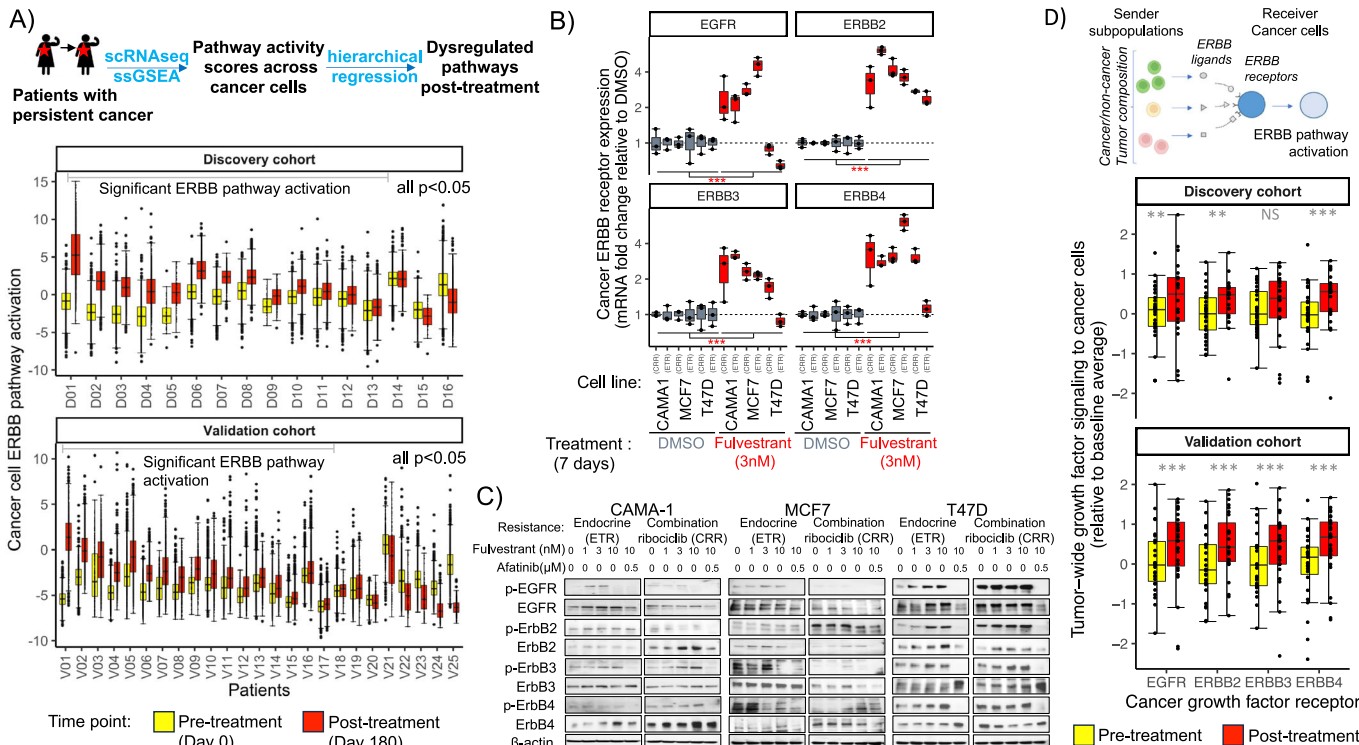

**Figure 2. Endocrine and CDK4/6i-resistant cells of patient tumors upregulate the ERBB pathway during treatment.**

(A) Phenotype of endocrine and CDK4/6i-resistant patient cancer cells. Consistent activation of intracellular ERBB pathway activity in cancer cells (points=single cells) resistant to treatment (red) compared to treatment-naive cancer cells (yellow) across patient tumors (x axis) in both the discovery and validation cohorts (top versus bottom panel). Pathway scores are a composite of 47 ERBB activation ssGSEA signatures from the C2 collection (listed in Fig. EV1). Significant increases in ERBB activity during treatment were identified using hierarchical regression and ANOVA after accounting for multiple comparisons using false discovery rate (FDR) P-value correction. (Discovery::Activation: Est=0.01, se=0.0029, df=16, $t = 3.55$, $P = 0.00261$) (Validation::Activation: Est=0.006, se=8.5e-5, df=24, $t = 70.83$, $P = 2e\text{-}16$). Patient-specific ERBB activation during treatment was tested using linear models (statistics in Tables EV1 and EV2) and tumors with significant activation indicated by gray bar. Box elements represent median (center line), upper/lower quartiles (hinges), 1.5*interquartile range (whiskers=minima/maxima) of ERBB activity within a sample. Sample size= 134,218 cells (87,913 pre-treatment; 46,305 post treatment), 41 paired pre-/post-treatment tumor samples. Patient groups: Discovery cohort treated with letrozole ($n = 7$)/combination ribociclib ($n = 9$); Validation cohort treated with letrozole ($n = 10$)/combination ribociclib ($n = 15$). (B) Validation in vitro that ER+ breast cancer cells broadly upregulate ERBB receptor gene expression under endocrine treatment. Box plot of fold change of ERBB growth factor receptor mRNA expression (panels), measured by qPCR in three paired cancer cell lines (x axis: CAMA-1, MCF-7 and T47D endocrine therapy-resistant (ETR) and combination ribociclib-resistant (CRR)) under endocrine therapy (fulvestrant 3 nM) compared to DMSO control (colors = treatment). Endocrine therapy broadly increases ERBB growth factor receptor expression after 7 days of treatment compared to DMSO control (linear mixed model expression change under:: EGFR: Est=1.22, se=0.28, df=32, $t = 4.38$, $P = 0.0001$, ERBB2:Est=2.72, se=0.30, df=32, $t = 9.21$, $P = 1.6e\text{-}10$, ERBB3:Est=1.12, se=0.18, df=32, $t = 6.20$, $P = 6e\text{-}7$, ERBB4:Est=2.26, se=0.36, df=32, $t = 6.34$, $P = 4e\text{-}7$; overall $P$ indicated by asterisks (significance of difference in ERBB expression between treatments across cell types measured by t-statistic-based test following Satterthwaite method)). Box elements represent median (center line), upper/lower quartiles (hinges), 1.5*interquartile range (whiskers=minima/maxima) of ERBB receptor fold change versus control. Sample size=144 qPCR samples, 6 cell lines, 3 replicates per gene ($n = 4$) and treatment ($n = 2$). (C) Validation in vitro that the ERBB pathway is activated through increased ERBB protein phosphorylation of receptors under endocrine treatment in a dose-dependent manner. Phosphorylated and total ERBB protein levels (measured by Western blot analysis) in ($n = 3$) paired cancer cell lines (CAMA-1, MCF-7 and T47D endocrine therapy-resistant (ETR) and combination ribociclib-resistant (CRR) lines) following 3 days of treatment (in 2D culture of 1% FBS) under increasing doses of endocrine therapy (fulvestrant: 0–10 nM) or pan-ERBB inhibitor (afatinib: 0/0.5 μM). Overall, ERBB activation and phosphorylation increased under endocrine therapy across ribociclib-resistant (CRR) and sensitive (ETR) cell lines, but with mixed extent in specific cancer cell lines (greater in CAMA-1 and T47D; lesser in MCF-7). Afatinib inhibits phosphorylation and reduces total concentration of ERBB receptors in each of the cancer cells. (D) Boxplots showing the increase in ERBB growth factor signaling to cancer cells from across the TME during treatment. Points indicate the strength of ERBB communication received by cancer cells via an ERBB receptor (x axis) in a given tumor sample. Box elements represent median (center line), upper/lower quartiles (hinges), 1.5*interquartile range (whiskers=minima/maxima) of these ERBB communications across tumors. Increases in ERBB signaling between pre- and post treatment (color=treatment timepoint) were identified using a linear mixed-effects model (with t-statistic-based test following the Satterthwaite method). The discovery and validation cohort (top versus bottom panel) show consistent increases in growth factor signaling to cancer cells from across the TME via all four ERBB receptor pathways (Discovery:: EGF:Est=0.264, se=0.084, df=658, $t = 3.14$, $P = 0.0018$, ERBB2:Est=0.242, se=0.089, df=665, $t = 2.71$, $P = 0.0069$, ERBB3:Est = −0.012, se=0.117, df=684, $t = \text{-}0.11$, $P = 0.917$, ERBB4:Est=0.319, se=0.089, df=665, $t = 3.57$, $P = 0.0003$) (Validation:: EGF:Est=0.398, se=0.078, df=1371, $t = 5.14$, $P = 3.2e\text{-}7$, ERBB2:Est=0.593, se=0.083, df=1371, $t = 7.16$, $P = 1.3e\text{-}12$, ERBB3:Est=0.402, se=0.109, df=1370,t = 3.68, $P = 2.4e\text{-}4$, ERBB4:Est=0.555, se=0.083, df=1371, $t = 6.70$, $P = 3.1e\text{-}11$). Receptors predicted to be bound to either NRG1-3, EGF, AREG, HBEGF, TGFA, EMA4D, HLA-A, EFNB1, ICAM1, GNAI2, CDH1, AREG, ANXA1, ADAM17. Largest increase in binding via: NRG1-3, EGF, and HBEGF. Sample size: Discovery:: 32 patient tumors ($n = 51$ pre-/post-treatment samples), Validation:: 28 patient tumors ($n = 54$ pre-/post-treatment samples). All statistical tests two-tailed. Source data are available online for this figure.

the luminal subtype (Kensler et al, 2019)), potentially reflecting a bypass of both endocrine and CDK4/6i treatment targets. Given these changes, we focused on ERBB receptor expression and found broad upregulation in post-treatment tumors compared to pre-treatment tumors, with ERBB2 and ERBB4 being most consistently activated in post-treatment resistant cells of discovery and validation cohort tumors.

As post-treatment resistant cancer cells show activation of the ERBB signaling pathway, we took a parallel experimental approach to test whether endocrine and CDK4/6i treatments drive compensatory ERBB receptor expression in vitro. We examined three paired endocrine therapy-resistant ER+ breast cancer cell lines in which the parental line is CDK4/6i sensitive and the evolved paired line is CDK4/6i-resistant. Across all lines, qPCR validated significant upregulation of all four ERBB receptor genes after one week of treatment with endocrine, CDK4/6i and combination therapy compared to DMSO control (Figs. 2B and EV2A). Endocrine therapy induced a more significant increase in ERBB receptor expression than CDK4/6 inhibition.

We then measured if increased expression of ERBB receptor during endocrine therapy led to increased ERBB receptor protein levels. Using Western blot analysis, we confirmed that fulvestrant treatment induced higher levels of ERBB receptors across cancer cell lines, although the extent was much greater in CAMA-1 and T47D cell lines and less in MCF-7 (Fig. 2C). Furthermore, endocrine therapy-induced greater phosphorylation of ERBB receptors, showing rapid dose-dependent compensatory activation of ERBB signaling across endocrine and combination ribociclib-resistant cell lines (again with more varied activation in MCF-7). The pan-ERBB inhibitor afatinib was able to diminish ERBB phosphorylation and control spheroid growth when combined with endocrine therapy (fulvestrant) across endocrine and combination ribociclib-resistant cell lines (Fig. EV2B). The results presented are from one experiment and were validated in repeated experiments (Fig. EV2C). Together, the genomic analyses of patient tumors and experiments both show that ERBB signaling is an alternative proliferative pathway activated by endocrine therapy in endocrine-resistant ER+ breast cancer cells.

To investigate the intercellular ERBB growth signaling dysregulation in post-treatment tumors, that may contribute to ERBB pathway activation, we used a ligand–receptor (LR) gene expression algorithm to decipher the communication between cell types. We first dissected the broad cell types (e.g., fibroblasts, diploid epithelial, cancer, endothelial, myeloid, and T cells) into phenotypically distinct subpopulations by combining intrinsic dimension estimation (packing number estimation (Erba et al, 2019)) and UMAP (Becht et al, 2018) to uncover the major axes of phenotypic heterogeneity within each cell type from ssGSEA phenotype scores. Next a curated LR communication database was used to define a set of 1444 LR communication pathways based on known protein–protein interactions (Ramilowski et al, 2015).

We assessed the strength of ERBB ligand–receptor communications received by cancer cells in each tumor sample from across all cancer and non-cancer cell-type populations in the TME (tumor-wide signaling) using the extended-expression product method (Aran et al, 2019; Armingol et al, 2021; Griffiths et al, 2025). For each LR communication pathway, we quantified the contribution of cells in each subpopulation to the total ligand signal production in the tumor. We then measured the total strength of signaling each

cell received from the many phenotypically diverse subpopulations of cells constituting the tumor, given its cognate receptor expression. This analysis provided a summary of all crosstalk received by cancer cells from across cells in each tumor (tumor-wide communication) and uncovered the network of communication among all cell types in the population.

We compared the tumor-wide ERBB signaling communications received by cancer cells in tumors sampled pre- and post treatment using linear models. This analysis revealed upregulation of ERBB growth factor signaling to cancer cells post treatment, with increased communications received via all ERBB growth factor receptors (EGFR/ERBB1, HER2/ERBB2, HER3/ERBB3, and HER4/ERBB4) (Fig. 2D). Increased ERBB signaling to cancer cells from across the TME was seen in both the discovery and the validation cohort and supports the observed upregulation of the ERBB pathway.

### Fibroblasts are a primary source of ERBB growth signals

To determine which cells in the tumor provide these GF signals, we compared the contribution of each cancer and non-cancer cell type to the post-treatment increase in cancer cells GF signaling within each tumor (Fig. 3A). In tumors from both the discovery and the validation cohorts, fibroblasts emerged as the primary source of additional GF signals to cancer cells. Tumors on average showed a 1.2-fold increase in the strength of ERBB LR communications from fibroblasts to cancer cells during treatment (95% CI = 1.1–1.3). Across patient tumors, fibroblast-to-cancer cell ERBB communication was consistently increased via various ERBB receptors (Fig. 3B). This result was significant in both the discovery and validation cohorts. These results are consistent with prior findings that ERBB ligand expression is significantly higher in fibroblasts than in cancer cell lines (T47D and MCF-7) and higher in stromal versus epithelial compartments of various laser capture microdissected (LCM) breast cancer tumor samples (Berdiel-Acer et al, 2021). Together, these findings suggest that fibroblasts are the primary source of additional ERBB signals which bind to upregulated ERBB receptors of cancer cells post treatment.

Due to increased crosstalk from fibroblasts to endocrine-resistant cancer cells, we next determined if there are specific fibroblast phenotypes that disproportionately provide the ERBB ligand signals. Phenotypic diversity across fibroblast populations in ER+ breast tumors was first characterized from scRNAseq gene expression using intrinsic dimensionality estimation. UMAP dimension reduction and trajectory-based analysis of dynamic changes in gene expression (Erba et al, 2019; Becht et al, 2018; Van den Berge et al, 2020). These analyses showed that the major axes of fibroblast phenotypic variation reflected differentiation to a highly mesenchymal phenotype (high hallmark EMT ssGSEA scores and high collagen and fibronectin expression) and activation of EGFR-driven proliferation (high biocarta EGF and ERK proliferation ssGSEA scores) (Meran et al, 2011; Midgley et al, 2013; Wu et al, 2020; Glabman et al, 2022) (Fig. EV3A; Appendix Fig. S3). To support this interpretation, unbiased clustering was performed to identify fibroblast subpopulations and differential expression confirmed the upregulation of mesenchymal and EGFR response markers genes in clusters that aligned closely with the major axes of the UMAP landscape (Appendix Fig. S4). We then compared ERBB signaling of fibroblasts across phenotypic states using multiple regression and ANOVA and found that ERBB ligand upregulation

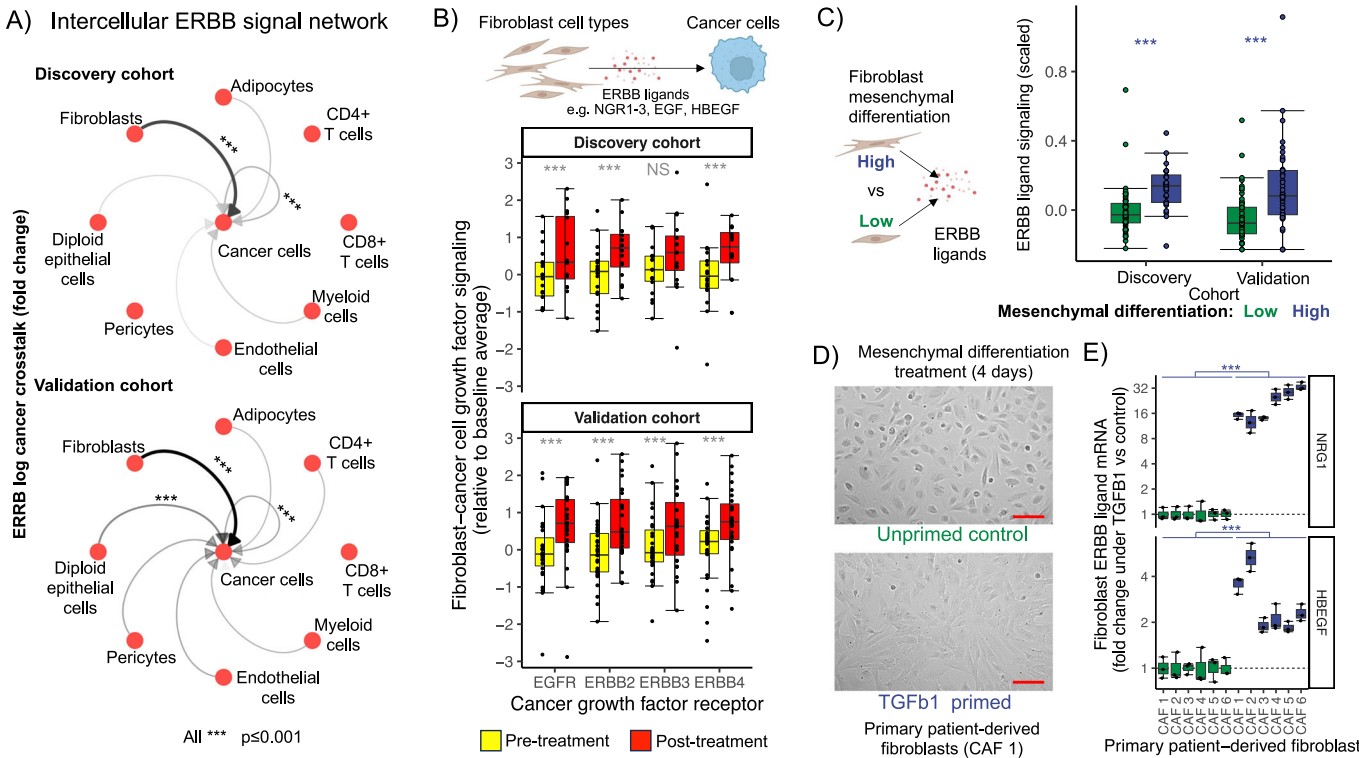

**Figure 3. Fibroblasts differentiated to a highly mesenchymal phenotype provide supplementary ERBB signals to facilitate cancer growth.**

(A) Fold change in ERBB crosstalk (edge width/opacity) of each cell type (nodes) with cancer cells post-treatment relative to baseline in the discovery and validation cohort (top versus bottom panel). Signaling direction indicated by arrows (Ligands and receptors listed in (B)). Significance tested using a linear model and indicated by asterisks (Discovery:: Fibroblasts: Est=0.061, se=0.022, df=126, t = 2.761, P = 0.007, Cancer cells: Est=0.046, se=0.021, df=126, t = 2.164, P = 0.03)(Validation:: Fibroblasts: Est=0.127, se=0.044, df=204, t = 2.900, P = 0.004, Cancer cells: Est=0.093, se=0.043, df=204, t = 2.134, P = 0.03, Diploid epithelial cells: Est=0.101, se=0.044, df=204, t = 2.310, P = 0.02). (B) Boxplots showing the increase in ERBB growth factor signaling from fibroblasts to cancer cells following treatment. Points indicate the strength of fibroblast to cancer communication (via a given receptor: x axis) in pre- and post-treatment tumor samples (color). Box elements represent median (center line), upper/lower quartiles (hinges), 1.5*interquartile range (whiskers=minima/maxima) of these ERBB communications across tumors. Increases in fibroblast to cancer crosstalk assessed using a linear mixed-effects model. The discovery and validation cohort (top versus bottom panel) show consistent increases in fibroblast growth factor signaling to cancer cells via all four ERBB receptor pathways (Discovery::EGF: Est=0.510, se=0.107, df=439, t = 4.75, P = 2.8e-6, ERBB2: Est=0.563, se=0.107, df=392, t = 5.24, P = 2.6e-7, ERBB3:Est=0.257, se=0.141, df=397, t = 1.83, P = 0.07, ERBB4: Est=0.488, se=0.107, df=386, t = 4.56, P = 6.4e-6)(Validation:: EGF: Est=0.670, se=0.076, df=1202, t = 8.78, P = 2e-16, ERBB2: Est=0.812, se=0.085, df=1202, P = 2e-16, ERBB3: Est=0.457, se=0.115, df=1201, P = 7.6e-5, ERBB4: Est=0.605, se=0.084, df=1201, P = 1.3e-5). Receptors predicted to be bound to either NRG1-3, EGF, AREG, HBEGF, TGFA, EMA4D, HLA-A, EFNB1, ICAM1, GNAI2, CDH1, AREG, ANXA1, ADAM17). (C) Boxplot showing the average ERBB ligand signaling of fibroblasts with high versus low mesenchymal differentiation (color) in tumor samples of the discovery and validation cohort (x axis). Box elements represent median (center line), upper/lower quartiles (hinges), 1.5*interquartile range (whiskers=minima/maxima) of fibroblast ERBB ligand expression across tumor samples. Points indicate median log ERBB signaling within a fibroblast type (high/low mesenchymal differentiation) in a specific sample across ligands (NGR1-3, EGF, HBEGF, AREG, TGFA, EMA4D, HLA-A, EFNB1, ICAM1, GNAI2, CDH1, AREG, ANXA1, and ADAM17). Fibroblasts increase ERBB ligand signaling upon differentiation to a highly mesenchymal phenotype (Discovery:: ERBB ligand activation linear model: Est=0.13, se=0.03, df=78, t = 4.1, P = 8.7e-5, n = 238,620 fibroblasts grouped by differentiation within 80 tumor samples (24 with highly differentiated fibroblasts; 56 with lowly differentiated fibroblasts)) (Validation:: ERBB ligand activation linear model: Est=0.17, se=0.039, df=95, t = 4.44, P = 2.5e-5, n = 164,577 fibroblasts grouped by differentiation within 97 tumor samples (41 with highly differentiated fibroblasts; 56 with lowly differentiated fibroblasts)). Sample sizes in (A–C): Discovery:: 76,246 tumor-derived single cells (47,275 pre-treatment, 28,971 post treatment) from 20 patients with paired pre- and post-treatment samples (n = 40), Validation:: 163,245 tumor-derived single cells (96,659 pre-treatment, 66,586 post treatment) from 28 patients with paired pre- and post-treatment samples (n = 56). (D) Representative images (×200 microscopy) of the mesenchymal morphological change of patient-derived fibroblasts when treated with TGFB1 (20 ng/mL) for 4 days (2D culture in 10% FBS medium). Scale bars = 50 μm. Sample size: n = 6 patient-derive fibroblast populations. (E) Comparison of ERBB ligand gene expression (HBEGF, NRG1 growth factors) of six primary patient-derived fibroblast populations when treated with TGFB1 (20 ng/mL) for 4 days compared to unprimed controls. Gene expression quantified by qPCR after performing microscopy. All six patient-derived fibroblast populations increased ERBB ligand expression when primed (Linear model estimates of log fold change under TGFB1:: NRG1:: CAF1: Est=2.72, se=0.077, df=4, t = 35.13, P = 3.9e-6, CAF2: Est=2.53, se=0.15, df=4, t = 17.26, P = 6.6e-5, CAF3: Est=2.64, se=0.08, df=4, t = 32.69, P = 5.2e-6, CAF4: Est=3.22, se=0.15, df=4, t = 21.02, P = 3.0e-5, CAF5: Est=3.35, se=0.10, df=4, t = 33.53, P = 4.72e-6, CAF6: Est=3.51, se=0.07, df=4, t = 50.67, P = 9.1e-7, HBEGF:: CAF1: Est=1.27, se=0.08, df=4, t = 15.04, P = 0.0001, CAF2: Est=1.67, se=0.12, df=4, t = 13.64, P = 0.0002, CAF3: Est=0.64, se=0.058, df=4, t = 11.10, P = 0.0004, CAF4: Est=0.74,se=0.14, df=4, t = 5.35, P = 0.0059, CAF5: Est=0.62,se=0.082, df=4, t = 7.51, P = 0.0017, CAF6: Est=0.83, se=0.077, df=4, t = 10.73, P = 0.0004). Asterisks show Fishers combined P-value significance (NRG1: $\chi^2$ = 141.67, P = 2.75e-24, HBEGF: $\chi^2$ = 89.77, P = 5.47e-14). Box elements represent median (center line), upper/lower quartiles (hinges), 1.5*interquartile range (whiskers=minima/maxima) of ERBB ligand fold change versus control. Sample size: n = 72 samples, 6 primary fibroblast lineages, 2 treatments, 3 replicates, 2 genes. All statistical tests two-tailed. Source data are available online for this figure.

(especially NRG1/2 and HBEGF) was commensurate with a shift in fibroblast differentiation to a highly mesenchymal phenotype in both the discovery and validation cohorts (Figs. 3C and EV3B,C). The highly mesenchymal fibroblasts also had greater collagen and fibronectin expression (Fig. EV3D) which supports extracellular matrix modification and cancer growth (Papait et al, 2022; Glabman et al, 2022; Erdogan and Webb, 2017). Furthermore, we found that during endocrine and combination ribociclib treatment fibroblasts across tumors showed activation of EGF-stimulated proliferation (Fig. EV3C). We observed a significant increase in the fraction of EGFR-activated fibroblasts post treatment and found that an increased frequency of this cell type was associated with greater post-treatment residual tumor burden (pathologically measured longest tumor length) (Appendix Fig. S3). It is known that EGF activation supports mesenchymal differentiation, primarily through TGF$\beta$ stimulation (Midgley et al, 2013; Simpson et al, 2010). Consistent with this, we observed upregulation of TGF$\beta$R1-3 receptors in EGF-activated fibroblasts which would support the transition to the highly mesenchymal (Fig. EV3A).

We then experimentally tested if TGF$\beta$ signaling stimulated mesenchymal differentiation as well as increased ERBB ligand signaling in patient-derived cancer-associated fibroblasts (Fig. 3D,E). We cultured fibroblasts isolated from six primary ER+ patient tumor samples for 4 days under TGF$\beta$1 or DMSO control treatments (called TGF$\beta$1 primed/unprimed). Using transmitted-light brightfield microscopy, we observed induction of the elongated morphology of mesenchymal fibroblasts in all TGF$\beta$1 primed patient-derived fibroblast populations but not in the unprimed controls (Fig. 3D) (Rogers et al, 2022) (Fig. EV4A). Using qPCR, we also showed that primed patient-derived fibroblasts and MRC5 fibroblast cell lines have increased EMT and fibrosis markers under TGF$\beta$1 treatment (Fig. EV4B,C). In all six patient-derived populations, TGF$\beta$1 -primed fibroblasts upregulated both neuregulin-1 (NRG1) and human basic epidermal growth factor (HBEGF) (Fig. 3E). Together these experimental results align with the patient analyses and indicates that the growth ligands predicted to drive ERBB-dependent proliferation of resistant patient cancer cells can be produced by highly mesenchymal fibroblasts.

### Cancer cells stimulate fibroblast differentiation and ERBB signaling

As ERBB ligand signaling in fibroblasts was associated with mesenchymal differentiation and EGF-stimulated proliferation, we next assessed which cell types may provide TGF$\beta$ and EGF crosstalk stimuli to fibroblasts in patient tumors. We compared the overall strength of mesenchymal differentiation communications received by fibroblasts from each cell type, using the LR communication analysis (Fig. 4A). Cancer cells were found to be the major source of TGF$\beta$/EGF stimuli of mesenchymal fibroblast differentiation in both the discovery and validation cohorts. This result indicates that cancer cells may modify fibroblast function by stimulating mesenchymal differentiation through increased TGF$\beta$, which then promotes their own oncogenic ERBB-dependent proliferation.

We next explored the specific ligand communications by which post-treatment resistant cancer cells may stimulate mesenchymal fibroblast differentiation and ERBB ligand provision. We compared the strength of TGF/EGF communications to fibroblasts from cancer cells compared to those sent to fibroblasts by non-cancer cells, using linear models (Fig. 4B). This analysis showed that

resistant cancer cells stimulated greater mesenchymal fibroblast differentiation through their increased provision of TGF$\beta$1–3 and HBEGF ligands compared to non-cancer cells. The increased TGF$\beta$1–3 cancer to fibroblast crosstalk was found in both the discovery and validation cohort tumors. Cancer and non-cancer cells did not provide significantly different amounts of HBEGF stimulus in the validation cohort. However, this result was largely due to high contributions from diploid epithelial cells that may have been cancerous but lacked clear copy number alteration and a myeloid cell subpopulation that may reflect regulatory macrophages (known producers of HBEGF (Edwards et al, 2009)) (Fig. 4A).

We next performed parallel in vitro experiments to test if endocrine therapy drives cancer cells to stimulate mesenchymal fibroblast differentiation, promoting fibroblast ERBB signaling. We focused on endocrine impacts as increased fibroblast differentiation and GF signaling was identified in the patient tumor-derived cancer cells independent of whether endocrine was combined with ribociclib treatment or not. We first assessed in vitro whether cancer cells, grown in monoculture, upregulate mesenchymal differentiation signals during endocrine therapy (Fig. 4C). We measured TGF$\beta$1 and HBEGF gene expression in the three endocrine-resistant cell line pairs (CDK4/6i sensitive (parental; ETR) vs resistant (evolved; CRR)) after 7 days of fulvestrant treatment compared to the DMSO control. Under fulvestrant treatment, the resistant CAMA-1 and MCF-7 cell lines (but not T47D) upregulated TGF$\beta$ or HBEGF ligands, as measured by qPCR.

Given the relationship between cancer cell TGF$\beta$1 and HBEGF upregulation and mesenchymal fibroblast differentiation, we next assessed how endocrine therapy modulates cancer cell signaling in the local presence of fibroblasts. We cultured the three pairs of endocrine and combination ribociclib-sensitive and resistant cancer cell lines either in monoculture or in coculture with fluorescently labeled fibroblasts (MRC5) and monitored population growth for 3 days under fulvestrant or DMSO control treatments. We then separated the labeled cell types using FACS sorting and measured cancer cell expression of TGF$\beta$1 and HBEGF across treatments and coculture conditions using qPCR (Fig. 4D). Without endocrine treatment, all three pairs of cancer cell lines upregulated HBEGF when cocultured with fibroblasts compared to monoculture controls, but not all cocultured populations upregulated TGF$\beta$1. In contrast, under endocrine therapy, all cocultured cancer cell lines upregulated TGF$\beta$1 and most upregulated HBEGF, compared to treatment monoculture controls. This result indicates that cancer cells stimulate mesenchymal fibroblast differentiation when they encounter fibroblasts. Further, endocrine therapy can induce cancer cells to drive greater mesenchymal fibroblast differentiation. Finally, using qPCR, we verified that fibroblasts show an increased mesenchymal phenotype after being cocultured with cancer cells for 3 days, with increased collagen levels (COL1A1: collagen type I alpha 1 chain) compared to fibroblast monocultures (Fig. EV4D).

Together the in vitro evidence supported the patient-derived findings and allowed us to test specific mechanistic hypotheses. These results indicate an oncogenic positive feedback loop whereby cancer cells promote fibroblast proliferation via HBEGF signaling. In turn, fibroblasts induce cancer cells to produce TGF$\beta$, stimulating mesenchymal fibroblast differentiation and GF signaling which promotes further ERBB-dependent cancer growth.

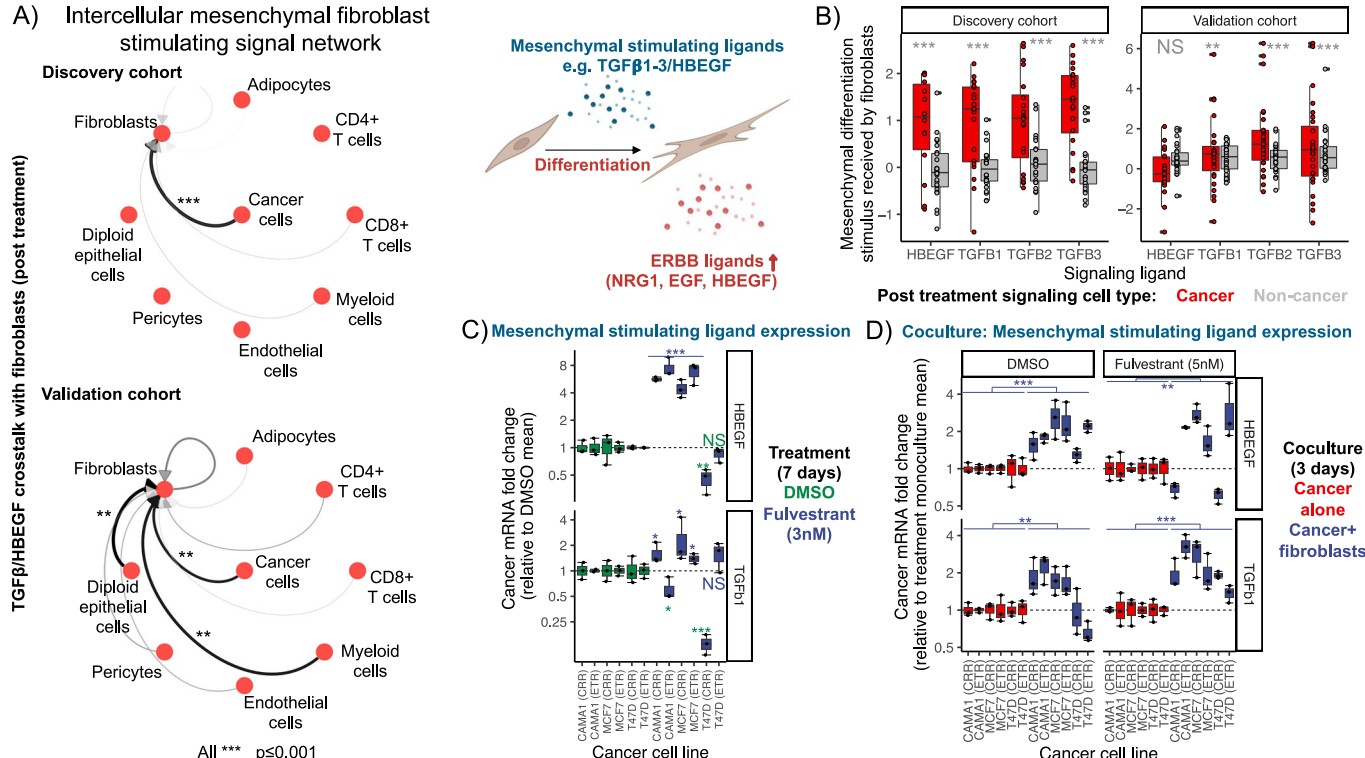

**Figure 4. Cancer cells stimulate fibroblast differentiation to a highly mesenchymal state, activating facilitatory ERBB signaling.**

(A) Network graph showing the strength of mesenchymal differentiation stimulating crosstalk from each cell type (nodes) to fibroblasts via the TGF Beta and HBEGF ligand–receptor communication pathways. Edge width/opacity proportional to overall communication between cell types across pathways post treatment. Signaling direction indicated by arrows (ligands and receptors listed in (B)). In both the discovery and validation cohorts, cancer cells provide the majority of mesenchymal stimulating signals to fibroblasts post treatment (Discovery:: mesenchymal stimulation ANOVA: Cancer cells: Est=0.434, se=0.044, df=173,t = 9.88, P = 2e-16, n = 44,531 tumor cells, 25 patients) (Validation:: mesenchymal stimulation ANOVA: Cancer cells: Est=0.164, se=0.063, df=211, t = 2.61, P = 0.0096, Diploid epithelial cells: Est=0.172, se=0.063, df=211, t = 2.75, P = 0.0065, Myeloid cells: Est=0.188, se=0.063, df=173, t = 3.00, P = 0.0031). Sample size: n = 66,586 tumor cells, 27 patients. (B) Boxplot showing the strength of mesenchymal differentiation signaling received by fibroblasts, post treatment, from cancer cells (red) compared to non-cancer cells (gray). Points indicate the strength of mesenchymal differentiation signals received by fibroblasts of a specific tumor via distinct HBEGF /TGFB1-3 ligand–receptor communication pathways (receptors on x axis; ligands=TGFB1-3, HBEGF, CDH1, INHBA, GNAI2). Box elements represent median (center line), upper/ lower quartiles (hinges), 1.5*interquartile range (whiskers=minima/maxima) of these signals across tumors. In both the discovery and validation panels), cancer cells provide greater mesenchymal differentiation signals to fibroblasts than non-cancerous cell types (Discovery cancer stimulus linear model:: HBEGF: Est=0.903, se=0.200, df=162, t = 4.53, P = 1.2e-5, TGFB1:Est=0.940, se=0.167, df=162, t = 5.63, P = 7.7e-8, TGFB2:Est=0.982, se=0.167, df=162, t = 5.88, P = 2.3e-8, TGFB3:Est=1.32, se=0.167, df=162, t = 7.90, P = 4e-13; n = 7629 fibroblasts, 23002 cancer cells, 21,529 non-cancer cells, 25 patients) (Validation cancer mesenchymal stimulus linear model:: HBEGF: Est = −0.18, se=0.30, df=198, t = −0.60, P = 0.55, TGFB1:Est=0.72, se=0.267, df=198, t = 2.70, P = 0.007, TGFB2:Est=1.53, se=0.267, df=198, t = 5.72, P = 3.8e-8, TGFB3:Est=1.03, se=0.267, df=198, t = 3.89, P = 0.00014). Sample size: n = 10,848 fibroblasts, 26215 cancer cells, 40371 non-cancer cells, 27 patients. (C) Boxplot showing upregulation of TGFB1 or HBEGF gene expression (mesenchymal differentiation signals) in 4/6 cancer cell lines (x axis; CAMA-1, MCF-7 and T47D endocrine therapy-resistant (ETR) and combination ribociclib-resistant (CRR)) following 7 days of fulvestrant (3 nM) treatment (blue) relative to DMSO treated controls (green).(Linear model estimates of log fold change under fulvestrant: HBEGF:: CAMA-1 CCR: Est=1.73, se=0.07, df=4, t = 24.57, P = 1.6e-5, CAMA-1 ETR: Est=2.01, se=0.13, df=4, t = 15.27, P = 1.1e-4, MCF-7 CCR: Est=1.48, se=0.18, df=4, t = 8.04, P = 0.0013, MCF-7 ETR: Est=1.89, se=0.12, df=4, t = 15.13, P = 1.1e-4, T47D CCR: Est = -0.82, se=0.14, df=4, t = −6.06, P = 0.003, T47D ETR: Est = −0.17, se=0.08, df=4, t = −2.27, P = 0.085, TGFB1:: CAMA-1 CCR: Est=0.46, se=0.15, df=4, t = 3.06, P = 0.038, CAMA-1 ETR: Est = −0.51, se=0.12, df=4, t = −4.13, P = 0.014, MCF-7 CCR: Est=0.77, se=0.272, df=4, t = 2.84, P = 0.047, MCF-7 ETR: Est=0.33, se=0.078, df=4, t = 4.26, P = 0.013, T47D CCR: Est = −2.00, se = 0.19, df = 4, t = −10.66, P = 4.4e-4, T47D CCR: Est=0.41, se = 0.19, df = 4, t = 2.21, P = 0.091). Sample size: n = 72 samples, 6 cancer cell lineages, 2 treatments, 3 replicates, 2 genes. Box elements represent median (center line), upper/lower quartiles (hinges), 1.5*interquartile range (whiskers=minima/maxima) of gene expression measured by qPCR for replicate populations (points). Cell cultured in 2D with 1% FBS. (D) Boxplot showing upregulation of TGFB1 or HBEGF gene expression (mesenchymal differentiation signals) in cancer cell lines (x axis; CAMA-1, MCF-7, and T47D endocrine therapy-resistant (ETR) and combination ribociclib-resistant (CRR)) when cocultured with fibroblasts (MRC5; blue) relative to when grown in monoculture (red). Increased mesenchymal differentiation signaling observed in cancer-fibroblast cocultures after 3 days of treatment with either fulvestrant (5 nM) (right) or DMSO control (left). (Linear mixed-effects model estimates of log fold change in coculture, (i) under DMSO:: HBEGF: Est=0.63, se=0.077, df=29, t = 8.20, P = 4.8e-9, TGFB1: Est=0.32, se=0.11, df=29, t = 2.96, P = 0.006, (ii) under fulvestrant (3 nM):: HBEGF: Est=4.0, se=0.14, df=29, t = 2.95, P = 0.006, TGFB1: Est=0.74, se=0.08, df=29, t = 8.53, P = 2.16e-9; overall P indicated by asterisks). Sample size: n = 72 samples, 6 cancer cell lineages, 2 treatments, 3 replicates, 2 genes. Box elements represent median (center line), upper/lower quartiles (hinges), 1.5*interquartile range (whiskers=minima/maxima) of gene expression measured by qPCR for replicate populations (points). Cell cultured in 3D with 1% FBS. All statistical tests two-tailed. Source data are available online for this figure.

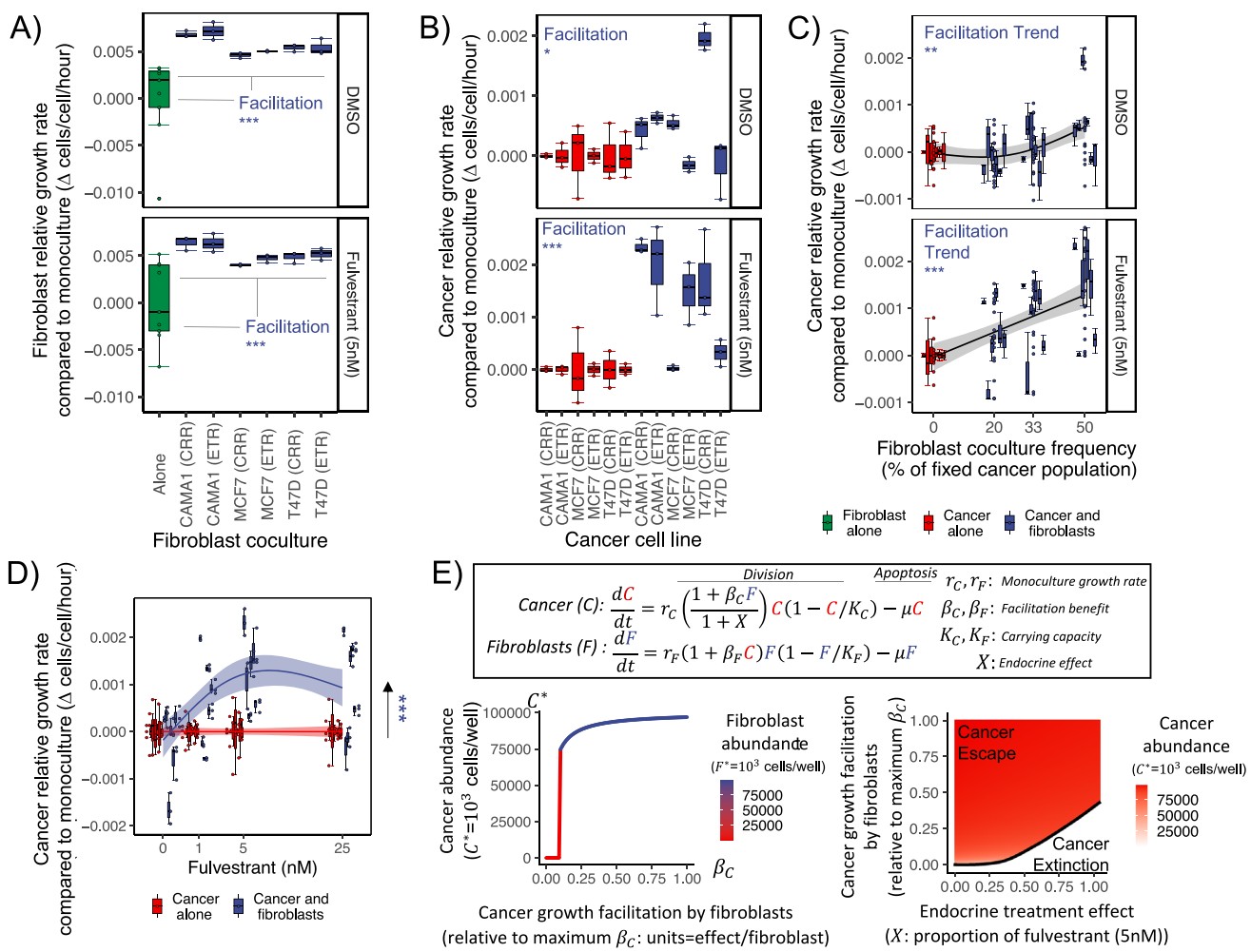

## Endocrine therapy-induced oncogenic cancer-fibroblast mutualism

We have found evidence that the TGFβ and ERBB signals that cancer and fibroblast exchange create bidirectional positive interactions that enhance the growth of both cell types. Such an ecological mutualism may confer resistance to endocrine treatment by engaging the ERBB pathway in cancer cells, bypassing the dependence on ER signaling (Bertness and Callaway, 1994). To test this cancer-fibroblast mutualism, we checked whether fibroblasts could promote the proliferation of cancer cells and vice versa. To do this, we measured the proliferation of fluorescently labeled fibroblasts (MRC5) and cancer cells when cultured alone (fibroblast or cancer monocultures) or together (50:50 coculture) for 14 days in 3D in vitro spheroids. We performed experiments under endocrine therapy and DMSO control conditions to assess whether the mutualism was constitutive or context-dependent (Fig. 5A,B). Relative growth rates measured the rate of change in cell abundance relative to the current cell abundance per unit of time. We measured the treatment/coculture effects on relative growth rate to quantify the proportional increase (positive values) or decrease (negative values) in cells per cell per hour (change in exponential growth rate) under coculture conditions compared to the

monoculture baseline. A mutualism results in increased per capita proliferation of both interacting populations, therefore we expected to observe increased relative growth rates of both fibroblasts and cancer cells when cocultured. Indeed, fibroblast growth was facilitated by each of the three pairs of cancer cell lines under endocrine and DMSO control conditions (Fig. 5A). Similarly, fibroblasts facilitated the growth of each of the three pairs of cancer cell lines, with cancer growth being facilitated more strongly under endocrine therapy (Fig. 5B).

A second prediction of the mutualism hypothesis is that cancer cell growth should be facilitated more strongly as the abundance of cocultured fibroblasts increases. To test this prediction, we repeated the coculture experiment with differing initial fibroblast numbers but fixed the initial cancer abundance (Fig. 5C). As expected, cancer cell growth rate accelerated with an increased initial fibroblasts ratio across each of the three pairs of cancer cell lines. Again, fibroblasts facilitated cancer growth more strongly under endocrine treatment than DMSO.

A third ecological prediction of the well-established stress gradient hypothesis is that the cancer-fibroblast mutualism should become stronger under more stressful treatment conditions, before collapsing at

◀ **Figure 5. Cancer and fibroblast cell mutualism.**

(A) Fibroblast growth facilitated by cocultured cancer cells. Boxplots showing the growth of MRC5 fibroblast populations when monocultured (green) compared to when cocultured with each of three cancer cell line pairs (blue: x axis; CAMA-1, MCF-7, and T47D endocrine therapy-resistant (ETR) and combination ribociclib-resistant (CRR)). Box elements represent median (center line), upper/lower quartiles (hinges), and 1.5*interquartile range (whiskers=minima/maxima) of difference in growth rates (cells/cell/hour) of replicate fibroblast populations (points=replicates) compared to the monoculture mean per treatment. Facilitation of fibroblast growth by CAMA-1, MCF-7, and T47D cancer cell lines evident both with or without endocrine therapy (rows: 5 nM fulvestrant vs DMSO) (linear mixed-effects estimate of fibroblast coculture growth increase (overall difference of mono versus coculture growth): (i) under DMSO:: Est=5.73e-3, se=6.36e-4, df=25, $t = 9.00$, $P = 2.5e-9$; (ii) under fulvestrant (5 nM):: Est=5.26e-3, se=6.77e-4, df=9.4,t = 7.77, $P = 2.1e-5$; overall $P$ indicated by asterisks). Sample size: $n = 54$ spheroids (measured at 5 timepoints): 18 fibroblast alone (9 replicates, 2 treatments), 36 fibroblast-cancer cocultures (3 replicates, 6 cancer cell lines, 2 treatments). All replicates were initially seeded with an equal number of fibroblast cells. Cocultures were seeded with additional cancers (1:1 ratio). Fibroblast growth measured over 14 days under 1% FBS. (B) Cancer growth facilitated by cocultured fibroblasts. Boxplots showing the growth of three cancer cell line pairs (x axis; CAMA-1, MCF-7 and T47D endocrine therapy-resistant (ETR) and combination ribociclib-resistant (CRR)) when cocultured with fibroblasts (blue) compared to cancer monocultures (red). Box elements represent median (center line), upper/lower quartiles (hinges), and 1.5*interquartile range (whiskers=minima/maxima) of difference in growth rates (cells/cell/hour) of replicate cancer populations (points=replicates) compared to the monoculture mean per cell line and treatment. Facilitation of CAMA-1, MCF-7 and T47D cancer cell lines evident both with or without endocrine therapy (rows: 5 nM fulvestrant vs DMSO) (Linear mixed effect cancer coculture growth increase: (i) under DMSO:: Est=5.45e-4, se=1.77e-4,df=7.95, $t = 3.07$, $P = 0.0155$; (ii) under fulvestrant (5 nM):: Est=1.31e-3, se=2.18e-4, df=8.46, $t = 6.02$, $P = 2.5e-4$; overall $P$ indicated by asterisks). All replicates were initially seeded with an equal number of cancer cells. Cocultures were seeded with additional fibroblasts (1:1 ratio). Cancer growth measured over 14 days under 1% FBS. Sample size: $n = 72$ spheroids (3 replicates, 6 cancer cell lines, 2 fibroblast abundances (present/absent), 2 treatments) measured across 5 timepoints. (C) Cancer growth facilitation increased with fibroblasts abundance. Boxplots showing the growth of the three cancer cell line pairs as the frequency of cocultured fibroblasts was increased from 0-50% the initial number of cancer cells (x axis). Box elements represent median (center line), upper/lower quartiles (hinges), and 1.5*interquartile range (whiskers=minima/maxima) of difference in growth rates (cells/cell/hour) of replicate cancer populations (points=replicates) compared to the monoculture mean per cell line and treatment. Trends in the cancer growth facilitation with increasing fibroblast abundance were measured for each treatment using generalized additive models (expectations=lines; 95% confidence interval of uncertainty=shaded region). When cocultured with increasing numbers of fibroblasts (blue bars; x axis), cancer growth rate increased across all six cell lines compared to growth in monoculture (red) (Linear mixed effect: growth acceleration: DMSO:: Est=1.1e-5, se=3.56e-6, df=68.3, $t = 3.07$, $P = 0.003$; Fulvestrant (5 nM):: Est=2.64e-5, se=4.58e-6, df=34.6, $t = 5.76$, $P = 1.69e-06$. Sample size: $n = 144$ spheroids (3 replicates, 6 cell lines (CAMA-7, MCF-7, T47D sensitive or resistant), 4 fibroblast abundances, 2 treatments) measured across 5 timepoints. (D) Endocrine treatment amplified fibroblast facilitation of cancer growth. Boxplots showing the growth of the three cancer cell line pairs when cocultured with fibroblasts (blue; 1:1 ratio) compared to cancer monocultures (red) across a range of doses of endocrine therapy (fulvestrant: 0–25 nM) (x axis). Box elements represent median (center line), upper/lower quartiles (hinges), 1.5*interquartile range (whiskers=minima/maxima) growth rates (cells/cell/hour) of replicate cancer populations (points=replicates) compared to the monoculture means per cell line and treatment. Trends of the facilitation of cancer growth by fibroblasts with increasing fulvestrant dose were measured using a generalized additive model (expectations=lines; 95% confidence interval of uncertainty=shaded region). When cocultured (blue), cancer growth rates were increasingly facilitated by fibroblasts at higher fulvestrant doses compared to the growth of monocultures (red) (Linear mixed effect: increased growth facilitation with fulvestrant dose:: Est=1.9e-4, se=5.38e-5, df=130, $t = 3.53$, $P = 5.75e-4$). Sample size: $n = 144$ spheroids (3 replicates, 6 cell lines (CAMA-1, MCF-7, T47D sensitive or resistant), 2 fibroblast abundances (present/absent), 4 treatments (fulvestrant; see x axis) measured across 5 timepoints. (E) Dynamical mutualism model predicts that blocking cancer growth facilitation by fibroblasts breaks the oncogenic mutualism and controls coculture growth. Using the in vitro mono and coculture growth data, we parameterized a differential equation model (top) describing the mutualistic interaction of cancer and fibroblasts and their population growth under endocrine therapy. We explored how cancer growth facilitation by fibroblasts impacted the equilibrium cancer abundance. On the left-hand side, we predicted cancer abundance ($C^*$: steady state) when the facilitation effect of fibroblasts ($\beta_C$) is reduced from the estimated maximum amount down to zero. The endocrine drug effect was fixed to the half-maximal value ($X = 0.5$). Coloration indicates the corresponding fibroblast steady-state abundance ($F^*$). On the right-hand side, we explored the impact on cancer cell abundance of blocking the facilitation effect of fibroblasts ($\beta_C$) across a gradient of effective endocrine doses between zero ($X = 0$) and 5 nM fulvestrant ($X = 1$). This analysis predicts that blocking the facilitation of cancer growth should break the oncogenic mutualism and allow control of cancer growth (cancer extinction below black line) at lower endocrine doses. Inferred parameter values: Cancer monoculture growth rate = $r_C = 1.1e-3$ (net proliferation = 1e-4 cells/cell/hour), Fibroblast monoculture growth rate = $r_F = 1e-5$ cells/cell/h, Facilitation effect on cancer growth: $\beta_C = 4.19e-4$, Facilitation effect on fibroblast growth: $\beta_F = 0.207$, Maximal endocrine cytostatic effect: X = 1, Death rate: $\mu=1e-3$ cells/cell/hour, Carrying capacity: $K_C/K_F = 1e-5$ thousand cells). All statistical tests two-tailed. Source data are available online for this figure.

the "edge of life" of one participant (Bertness and Callaway, 1994; Hammarlund and Harcombe, 2019; Michalet et al, 2014). This outcome is expected when evolutionary pressures select for cooperative interactions that ameliorate stress (e.g., (Emond et al, 2023)). To test this prediction, we repeated the coculture experiment under increasing doses of endocrine therapy (fulvestrant 0–25 nM concentrations) (Fig. 5D). As expected, endocrine therapy reduced cancer growth. However, when we compared cancer growth in coculture to that of monocultures grown under the same treatment, we found that fibroblasts increasingly supported cancer growth as the fulvestrant dose increased. In each of the three pairs of cancer cell lines, fibroblast facilitation of growth increased up to 5 nM fulvestrant, with all cancer cell lines growing faster in coculture than monoculture. Finally, at the highest dose (25 nM fulvestrant), we observed a reduction of fibroblast facilitation in three cancer cell lines as the mutualism breaks down.

Together, these results support that the communications identified in the patient tumor-derived scRNAseq analyses of cell crosstalk may have identified an oncogenic mutualism that emerges under endocrine therapy to facilitate cancer cell growth. We formalized this logic into a differential equation model of cancer-

fibroblast mutualism (following (Wu et al, 2019)) (Fig. 5E). The model describes the cancer and fibroblast population growth ($r_i$ and $K_i$ represent intrinsic growth rate and carrying capacity of cell type i), with each cell type facilitating the proliferation of the other ($\beta_C$ = cancer facilitation/fibroblast; $\beta_F$ = fibroblast facilitation/cancer cells). By including the anti-proliferative effect of endocrine therapy ($X$), the model formalizes the stress gradient hypothesis. Steady-state analysis reproduced the expectation that cancer-fibroblast mutualism should broadly increase cancer growth and maintain greater cancer cell abundances at higher endocrine doses. Further, the model also indicates that blocking the ERBB-mediated facilitation of cancer growth by fibroblasts should break the mutualism and allow greater control of coculture growth at lower doses of endocrine therapy.

### Targeting the cancer-fibroblast mutualism and ERBB-dependent proliferation of resistant cancer cells using ERBB inhibition

As the mathematical model predicted that blocking the mutualism should control cancer growth in the endocrine therapy-resistant cancer cells from patient tumors, we examined the treatment

efficacy of the pan-ERBB inhibitor afatinib. We compared the growth of the three cell line pairs when monocultured or cocultured with fibroblasts in 3D spheroids for 14 days under endocrine (fulvestrant), ERBB inhibitor (afatinib) or combination treatments (Fig. 6). If ERBB inhibition blocks the cancer-fibroblast mutualism, afatinib treatment should reduce the growth rate of both cancer and fibroblast cells under afatinib treatment. We observed that afatinib reduced the growth rate of cocultured fibroblasts similar to that of the untreated (DMSO) fibroblast monocultures, indicating a blockage of mutualistic growth (Fig. 6A). Fibroblast growth in monoculture was also impacted by afatinib, indicating a role of ERBB signaling in fibroblast proliferation.

The growth of cancer cells was reduced even more effectively by afatinib, indicating specificity towards cancer cells (Fig. 6B). Cocultured cancer growth rate was typically reduced to well below that of the untreated (DMSO) cancer monocultures. Under afatinib treatment, cocultured fibroblasts provided weak support to accelerate cancer growth of several cancer cell lines (CAMA-1 CRR/ETR) but hampered the growth of other cell lines (MCF-7 CRR/ETR) when compared to afatinib-treated cancer monocultures. As a result, no significant benefit of fibroblast coculture was detectable across cancer cell lines under afatinib treatment. In addition, cocultured cancer growth was overall substantially slowed relative to untreated (DMSO) conditions for all cancer cell lines. The above results were qualitatively similar when comparing combination afatinib and fulvestrant treatment with the fulvestrant alone control, but with slower cancer growth rates and spheroid shrinkage frequently achieved (Fig. EV5A). This finding demonstrates that blocking ERBB signaling can increase endocrine sensitivity.

We next examined afatinib's impact on cancer-fibroblast spheroid morphology and cell interactions using a more physiologically structured 3D collagen matrix system (see "Methods") (Figs. 6C and EV5B,C). Fluorescent imaging compared the spatial distribution of cancer and fibroblast cells after 2 weeks of treatment with afatinib or DMSO control. We observed that control-treated cancer spheroids had a distinct invasive morphology, with cancer cells growing along fibroblast outgrowth spikes (Fig. 6C: upper panels per condition). Endocrine treatment partially decreased cancer spheroid growth but not the spikey morphology, likely because of its weak impact on fibroblast growth and migration (Fig. EV5A,B). ERBB inhibition blocked cancer proliferation, halting spheroid outgrowth, but also prevented the invasive fibroblast outgrowth and spiked morphology and was also effective in combination with endocrine therapy (Fig. EV5B). Focusing closely on the invasive front of cancer-fibroblast cocultures (Fig. 6C, lower panels per condition) we observed an elongated mesenchymal morphology of outgrowing fibroblasts in control cocultures. Afatinib treatment caused a near-complete loss of these mesenchymal fibroblasts (Fig. 6C: lower right panels per cell type). Together these results indicate that pan-ERBB inhibition can modulate cancer-fibroblast interactions and block the cancer-fibroblast mutualism fueling ERBB-dependent proliferation.

## Discussion

In this study, we discovered a role of ERBB pathway activation in driving GF signal-mediated proliferation of endocrine and CDK4/

6i-resistant ER+ breast cancer cells (Fig. 6D). We show how cancer cells can modulate fibroblast function under endocrine treatment by stimulating mesenchymal fibroblast differentiation and amplifying their provision of ERBB ligand signals, which in turn bind to ERBB receptors on cancer cells to drive growth factor medicated proliferation independent of estrogen signaling. We find that treatment strengthens an oncogenic mutualism between cancer and fibroblast cells that subsequently supports endocrine therapy resistance. We highlight the multiple beneficial roles that ERBB inhibitors can play in blocking this mutualism by targeting the GF signaling-rich environment that promotes resistance through ERBB activity in cancer cells.

A significant amount of research on endocrine therapy resistance is focused on cancer cell-intrinsic signaling. In this study, we discovered that high levels of extrinsic GF activity in the TME as a whole is experienced by endocrine therapy-resistant cancer cells. These findings are discoverable due to the serial patient tumor scRNAseq data and relatively new methods for discovering interactions between cell populations and by leveraging multiple timepoints during treatment to identify acquisition of resistance and crosstalk with other cells in the microenvironment. Further, the identification of physiologically relevant resistance traits can be supported by integrated analyses of patient data and experimental systems.

ERBB ligands and receptors have a myriad of roles in cancer and non-cancer tumor-associated cells. For example, a recent study showed that prostate CAFs secrete neuregulin-1 (NRG1) to promote anti-androgen therapy resistance through activation of HER3 cancer cell signaling (Zhang et al, 2020). Indeed, a diversity of other tumor-associated cells can acquire or promote oncogenic phenotypes following ERBB signaling dysregulation in cancer (Schumacher et al, 2017; Wu et al, 2021; Zhang et al, 2016; Kerneur et al, 2022). In this study, we find that two ERBB receptor ligands are transcriptionally upregulated by CAFs after endocrine treatment (NRG1 and HBEGF) and contribute to cancer cell endocrine therapy resistance. CAF secretion of these ligands can be regulated by cancer cell upregulation of TGFβ, a factor that promotes fibroblast to myofibroblast transition and activation of mesenchymal fibroblast proliferation, migration and tumor fibrosis (Shi et al, 2020; Calon et al, 2014; Barnett and Vilar, 2018; Mueller and Fusenig, 2004). This feedforward bypass pathway from endocrine to growth factor dependence may provide an opportunity to inhibit a CAF:cancer cell mutualistic mechanism of resistance and prolong and/or increase response to endocrine therapy.

While it is estimated that up to 50% of stage II and III high-risk patients will progress to a metastatic state, these earlier-stage patients have higher likelihood of being cured than later-stage metastatic patients. Recent advances in therapies for progressive ER + patients include many targeted therapies, one of which is the anti-HER2 (ERBB2) chemotherapy antibody-drug conjugate trastuzumab deruxtecan. This therapy was approved by the FDA in August 2022 for patients with unresectable/metastatic HER2-low breast cancer based on the phase 3 DESTINY-Breast04 study (Modi et al, 2022). Our data show that HER2 and other ERBB receptors such as ERBB4 are upregulated in tumors that grow during treatment with endocrine and cell cycle therapies (Griffiths et al, 2021). While the mechanisms by which trastuzumab deruxtecan works in ER + /HER− tumors are still being determined, its efficacy in this population is compelling and could reflect higher

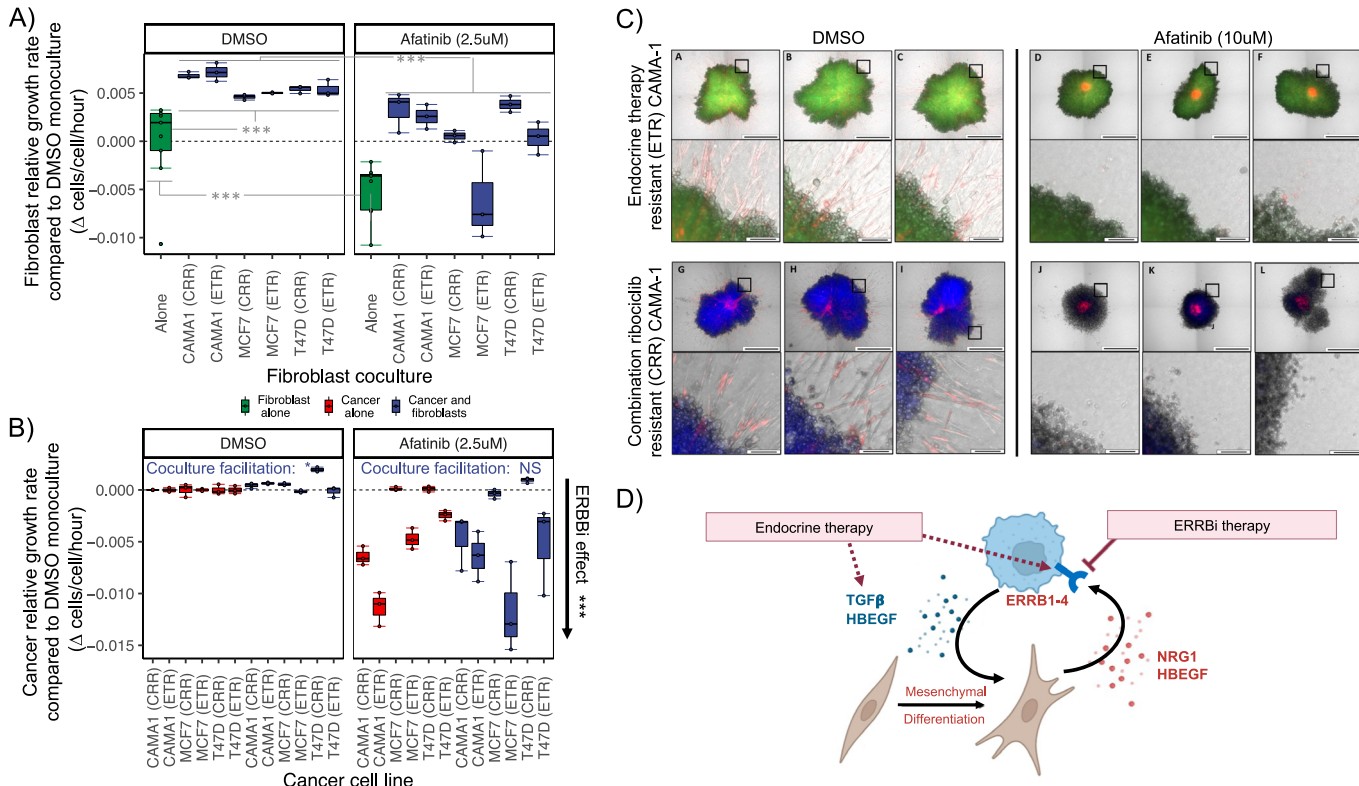

**Figure 6.   ERBB inhibition targets diverse resistant cancer cell lines, blocks the cancer-fibroblast mutualism fueling ERBB-dependent proliferation and controls fibroblast-guided invasion.**

(A) Afatinib controls cocultured fibroblast growth which is otherwise accelerated by cancer cells. Boxplots showing the growth of fibroblast (MRC5) populations in monocultured (green) compared to when cocultured with one of six cancer cell lines (blue: *x* axis; CAMA-1, MCF-7, and T47D endocrine therapy-resistant (ETR) and combination ribociclib-resistant (CRR)). Box elements represent median (center line), upper/lower quartiles (hinges), and 1.5*interquartile range (whiskers=minima/maxima) of difference in growth rates (cells/cell/hour) of replicate fibroblast populations (points=replicates) compared to the DMSO monoculture mean (dashed line). Fibroblast growth was facilitated by all cancer cell lines (left-hand panel: DMSO) (Linear mixed effect fibroblast coculture growth increase under DMSO:: Est=5.7e-3, se=6.4e-4, df=25, *t* = 9.0, *P* = 2.5e-9). Afatinib controlled fibroblast growth (right-hand panel: afatinib 2.5 nM), reducing cocultured growth rates to the rate of the untreated fibroblast monocultures across all six cancer coculture conditions (Linear mixed effect reduction of cocultured fibroblast growth under afatinib:: Est = −0.0049, se=0.00075, df=29, *t* = -6.68, *P* = 2.5e-7; equivalent fibroblast growth of afatinib-treated cocultures and untreated monocultures:: Est=1e-3, se=1.5e-3, df=22.4, *t* = 0.68, *P* = 0.505; *P* indicated by asterisks). Afatinib also reduced monoculture fibroblast growth rates compared to DMSO control (Linear mixed effect:: Est = −5.0e-3, se=1.5e-3, df=14, *t* = −3.35, *P* = 0.0047; *P* indicated by asterisks). All replicates were initially seeded with an equal number of fibroblast cells. Cocultures were seeded with additional cancers (1:1 ratio). Fibroblast growth measured over 14 days under 1% FBS. Sample size: *n* = 54 spheroids (measured at 5 timepoints): 18 fibroblast alone (9 replicates, 2 treatments), 36 fibroblast-cancer cocultures (3 replicates, 6 cancer cell lines, 2 treatments). (B) Afatinib blocks the facilitation of cancer cell growth by fibroblasts, preventing mono and cocultured cancer cell growth. Boxplots showing the growth of three cancer cell line pairs (*x* axis; CAMA-1, MCF-7, and T47D endocrine therapy-resistant (ETR) and combination ribociclib-resistant (CRR)) when cocultured with fibroblasts (blue) compared to cancer monocultures (red). Box elements represent median (center line), upper/lower quartiles (hinges), and 1.5*interquartile range (whiskers=minima/maxima) of difference in growth rates (cells/cell/hour) of replicate cancer populations compared to the DMSO monoculture mean per cell line (dashed line) (points=replicates). The facilitation of CAMA-1, MCF-7 and T47D cancer cell lines evident under DMSO (left-hand panel: Linear mixed effect estimates of cancer growth facilitation when cocultured under DMSO:: Est=5.45e-4, se=1.63e-4, df=29, *t* = 3.33, *P* = 0.0024) was controlled by afatinib (Linear mixed effect estimates of cancer facilitation by fibroblasts being blocked under afatinib (2.5 μM): no coculture benefit:: Est=4.4e-4, se=9.6e-4, df=29, *t* = −0.46, *P* = 0.65). Growth of all 6 cancer cell lines was substantially reduced under afatinib treatment compared to DMSO control (right-hand panel blue bars all lower than left; Linear mixed effect:: Est = −4.9e-3, se=7.5e-4, df=29, *t* = −6.68, *P* = 2.5e-7). All replicates were initially seeded with an equal number of cancer cells. Cocultures were seeded with additional fibroblasts (1:1 ratio). Cancer growth measured over 14 days under 1% FBS. Sample size: *n* = 72 spheroids (3 replicates, 6 cell lines, 2 fibroblast abundances (present/absent), 2 treatments) measured across 5 timepoints. (C) Afatinib halts cancer-fibroblast spheroid outgrowth and eliminates the mesenchymal population, impeding expansion of the invasive front. Fluorescence images of replicate populations (*n* = 3) comparing the spatial distribution of cancer and fibroblast cells after 7 days of treatment with DMSO control (left) or afatinib (right) in endocrine therapy-resistant (ETR) (top) and combination ribociclib-resistant (CRR) (bottom) CAMA-1 cancer cells cocultured with fibroblasts (MRC5). Subpanels for each cell type and condition show replicate populations (columns) as 2 × 2 tiled maximum intensity projection images (top row, scale = 1000 μm) and close-ups of the invasive front (bottom row, ×10 zoom subsets, scale = 100 μm). (D) Schematic model summarizing the validated mechanism of ERBB-dependent proliferation providing resistance of cancer cells to endocrine therapy through an induced oncogenic mutualism with fibroblasts. In most patient tumors, growth factor signal-mediated proliferation is driven by compensatory ERBB growth factor receptor upregulation by cancer cells along with fibroblast provision of cognate ligand signals. Under endocrine therapy, cancer cells also increase TGF Beta signaling, stimulating a mesenchymal fibroblast phenotype with increased ERBB growth factors production, leading TME growth factor enrichment. Endocrine therapy induces an oncogenic mutualism (indicated by the positive feedback circle) fueling cancer and fibroblast growth. ERBB inhibition can break this mutualism, preventing growth factor signal-mediated cancer cell proliferation and controlling invasion. All statistical tests two-tailed. Source data are available online for this figure.

levels and dependency of endocrine-resistant cancer cells on GF signaling. Indeed, experiments show that the pan-ERBB inhibitor afatinib can block both cancer and CAF growth during endocrine therapy (Fig. 6). Further research into the effectiveness of other ERBB inhibitors, including pan-ERBB inhibitors such as neratinib, may provide an approach to block one of the most common resistance traits from emerging in these tumors during endocrine therapy.

We have discovered that ERBB pathway activation is a recurrent resistance phenotype of endocrine-resistant cancer cells that allows GF signal-mediated proliferation. ERBB inhibitors can play a dual role in targeting endocrine-resistant cells. First, they target a common cancer cell-intrinsic resistance mechanism by blocking compensatory GF signaling induced by endocrine therapy. Second, they break an oncogenic mutualism between cancer and fibroblasts, reducing the supply of GF signals and slowing fibroblast-guided invasion. These findings suggest a novel approach to overcome endocrine therapy resistance by targeting the cancer-fibroblast microenvironment.

# Methods

### Reagents and tools table

| Reagent/resource | Reference or source | Identifier or catalog number |
| --- | --- | --- |
| **Experimental models** | | |
| 2003CAMA1 cells (*H. sapiens*) | ATCC | HTB-21 |
| MCF-7 cells (*H. sapiens*) | ATCC | HTB-22 |
| T47D cells (*H. sapiens*) | ATCC | HTB-133 |
| MRC5 cells (*H. sapiens*) | ATCC | CCL-171 |
| HEK-293 cells (*H. sapiens*) | ATCC | CRL-1573 |
| CAMA-1 ETR/CAMA-1_V2 | Grolmusz et al, 2020; Emond et al, 2023 | |
| MCF-7 ETR/MCF-7_V2 | Emond et al, 2023 | |
| T47D (ETR)/T47D_V2 | This study | |
| CAMA-1 (CRR)/CAMA-1_riboR_Cer2 | Grolmusz et al, 2020; Emond et al, 2023 | |
| MCF-7 (CRR)/MCF-7-1_riboR_Cer2 | Emond et al, 2023 | |
| T47D (CRR)/T47D_riboR_Cer2 | This study | |
| MRC5_C2 | This study | |
| **Recombinant DNA** | | |
| LeGO-V2 | Addgene, Weber et al, 2008 | plasmids #27340 |
| LeGO-Cer2 | Addgene, Weber et al, 2008 | plasmids #27338 |
| LeGO-C2 | Addgene, Weber et al, 2008 | plasmids #27339 |
| **Antibodies** | | |
| Anti-rabbit IgG, HRP-linked Antibody | Cell Signaling Technology | #7074 |
| Anti-mouse IgG, HRP-linked Antibody | Cell Signaling Technology | #7076 |

| Reagent/resource | Reference or source | Identifier or catalog number |
| --- | --- | --- |
| Phospho-EGF Receptor (Tyr1068) (D7A5) XP Rabbit mAb | Cell Signaling Technology | #3777 |
| Phospho-HER2/ErbB2 (Tyr1221/1222) (6B12) Rabbit mAb | Cell Signaling Technology | #2243 |
| Phospho-HER3/ErbB3 (Tyr1289) (D1B5) Rabbit mAb | Cell Signaling Technology | #2842 |
| Phospho-HER4/ErbB4 (Tyr1284)/EGFR (Tyr1173) (21A9) Rabbit mAb | Cell Signaling Technology | #4757 |
| EGFR-Specific Polyclonal rabbit antibody | Proteintech | 18986-1-AP |
| HER2/ErbB2 (29D8) Rabbit mAb | Cell Signaling Technology | #2165 |
| HER3/ErbB3 (1B2E) Rabbit mAb | Cell Signaling Technology | #4754 |
| HER4/ErbB4 (111B2) Rabbit mAb | Cell Signaling Technology | #4795 |
| Goat anti-β-actin (C4) | Santa Cruz Biotechnology | c-47778 |
| FITC labeled anti-human CD90 | Miltenyibiotec | 130-117-684 |
| PE labeled anti-human EpCAM | Miltenyibiotec | 130-113-264 |
| **Oligonucleotides and sequence-based reagents** | | |
| PCR primers | | |
| RT-qPCR assay | This study | See Table EV4 |
| **Chemicals, enzymes, and other reagents** | | |
| DPBS, no calcium, no magnesium | Gibco | 14190144 |
| RPMI 1640 Medium | Gibco | 11875093 |
| DMEM, high glucose, pyruvate | Gibco | 11995065 |
| Cytiva HyClone™ Dulbecco's Modified Eagles Medium | Cytiva | SH3028401 |
| Renaissance Essential Tumor Medium with Supplement | Cellaria | CM-0001 |
| Fetal bovine serum (FBS) | Millipore Sigma | 12306C |
| Trypsin-EDTA (0.25%), phenol red | Gibco | 25200056 |
| Antibiotic–Antimycotic (100X) | Gibco | 15240062 |
| collagen CellMatrix Type I-A | Nitta Gelatin Inc | 631-00653 |
| HEPES pH 7.4 | Thermo Fisher Scientific | 15630106 |
| Mycoplasma-negative using the Mycoalert PLUS Mycoplasma detection kit | Lonza | LT07-703 |
| Ribociclib (LEE011) | Selleckchem | S7440 |
| Fulvestrant (ICI-182780) | Selleckchem | S1191 |
| Afatinib (BIBW2992) | Selleckchem | S1011 |

| Reagent/resource | Reference or source | Identifier or catalog number |
|---|---|---|
| Letrozole (CGS 2026) | Selleckchem | S1235 |
| (Z)-4-Hydroxytamoxifen [(Z)-4-OHT] | Selleckchem | S8956 |
| TGFβ1 | Peprotech | 100-21 |
| RIPA Lysis and Extraction Buffer | Thermo Scientific | 89901 |
| PMSF Protease Inhibitor | Thermo Scientific | 36978 |
| Halt™ protease and phosphatase inhibitor cocktail (100×) | Thermo Scientific | 78440 |
| Pierce™ BCA Protein Assay Kit | Thermo Scientific | 23225 |
| Laemmli SDS sample buffer, reducing (6X) | Thermo Scientific | J61337.AD |
| 4–20% Mini-PROTEAN TGX™ Precast Protein Gels, 15-well, 15 µl | Bio-rad | 4561096 |
| Precision Plus Protein™ Dual Color Standards | Bio-rad | 1610374EDU |
| Ponceau S Staining Solution | Thermo Scientific | A40000278 |
| Fisher BioReagents™ Bovine Serum Albumin (BSA) Protease-free Powder | Fisher Scientific | BP9703100 |
| SuperSignal™ West Pico PLUS Chemiluminescent Substrate | Thermo Scientific | 35478 |
| SuperSignal™ West Femto Maximum Sensitivity Substrate | Thermo Scientific | 3096 |
| Restore™ PLUS Western Blot Stripping Buffer | Thermo Scientific | 46430 |
| Lipofectamine 3000 Transfection Reagent | Thermo Scientific | L3000015 |
| CellTiter-Glo® Luminescent Cell Viability Assay Kit | Promega | G7573 |
| Fast SYBR™ Green Master Mix | Thermo Scientific/Applied Biosystems™ | 4385617 |
| RNeasy Plus Mini Kit | QIAGEN | 74136 |
| High-Capacity cDNA Reverse Transcription Kit | Thermo Scientific/Applied Biosystems™ | 4368814 |
| Dimethyl Sulfoxide, Fisher BioReagents™ | Fisher Scientific | BP231-100 |
| **Nuclei Isolation and snRNA-Seq reagents** | | |
| DNA LoBind™ Tubes | Eppendorf | 022431021 |
| 1 M Tris-HCl, pH 7.8, sterile | Teknova | T1078 |
| Sodium Chloride, 5 M, RNAse free | Thermo Scientific | AAJ60434AK |
| Calcium Chloride, 1 M | G-Biosciences | R040 |
| Magnesium Chloride, 1 M | G-Biosciences | R004 |
| Bovine Serum Albumin, 10% Solution, Nuclease Free | EMD Millipore | 126615 |
| Igepal ca-630 | MP Biomedicals | 02198596-CF |

| Reagent/resource | Reference or source | Identifier or catalog number |
|---|---|---|
| DAPI (4',6-diamidino-2-phenylindole, dihydrochloride) | Invitrogen | D1306 |
| EDTA, 0.5 M | VWR | E522-100ML |
| UltraPure™ DNase/RNase-Free Distilled Water | Invitrogen | 10977015 |
| Necrostatin-1 | Sigma-Aldrich | N9037 |
| HPN-07 | Sigma-Aldrich | SML2163 |
| QVD-OPH | AstaTech, Inc. | ATE959438252 |
| Sodium Hydroxybutyrate | Sigma-Aldrich | 54965 |
| PBS, pH 7.4 | Gibco | 10010 |
| SUPERase·In™ RNase Inhibitor (20 U/µL) | Invitrogen | AM2696 |
| EASYstrainer Cell Strainer, 40 µm | Greiner Bio-One | 542040 |
| Qubit™ dsDNA HS Assay Kit | Invitrogen | Q32851 |
| Ethanol, 200 proof, Mol. Bio Grade | Fisher Scientific | BP28184 |
| Chromium Single Cell 3' GEM, Library & Gel Bead Kit v3, 16 rxns | 10X Genomics | 1000075 |
| Chromium Chip B Single Cell Kit, 48 rxns | 10X Genomics | 1000153 |
| **Software** | | |
| Gen5 3.14 software | https://www.agilent.com/en/product/cell-analysis/cell-imaging-microscopy/cell-imaging-microscopy-software | |
| Zen software (3.6 Blue edition) | https://www.zeiss.com/microscopy/us/products/software/zeiss-zen-lite.html | |
| R software (version 4.3.1) | R Core Team (2023) | |
| ImageJ | https://imagej.nih.gov/ij/index.html | |
| **Instrument** | | |
| Cytation 5 imager | Biotek Instruments | |
| Widefield Zeiss Observer 7 | Carl Zeiss | |
| Zeiss LSM 880 with Airyscan | Carl Zeiss | |
| Tecan Infinite M1000 reader | Tecan | |
| Chromium | 10X Genomics | |
| **Materials** | | |
| Tubes and Flat Caps, strips of 8 | Thermo Scientific | AB1182 |
| MicroAmp™ Optical 384-Well Reaction Plate with Barcode | Thermo Scientific/Applied Biosystems™ | 4309849 |
| MicroAmp™ Optical Adhesive Film | Thermo Scientific/Applied Biosystems™ | 4311971 |

| Reagent/resource | Reference or source | Identifier or catalog number |
|---|---|---|
| Corning™ 384-Well, Cell Culture-Treated, Flat-Bottom, Low Flange Microplate | Corning | 3764 |
| Corning™ 96-Well, Ultra-Low Binding, U-Shaped-Bottom Microplate | Corning | 4520 |
| Corning™ Costar™ 6-well Clear TC-treated Multiple Well Plates, Sterile | Corning | 3516 |
| Millicell Standing Cell Culture Inserts, (0.4 µm PTFE hydrophilic) | Millipore Sigma | PICM03050 |

# Revealing mechanisms of GF signal-mediated proliferation in endocrine-CDK4/6 inhibitor-resistant cancer cells in the patient setting

### Clinical trial, patient cohorts, and serial tumor biopsies

Patient tumor core biopsies obtained serially during treatment were collected prospectively under Clinical Trial #NCT02712723 (Khan, 2022), during a randomized, placebo controlled, multicenter investigator-initiated trial directed by Dr. Qamar Khan at the University of Kansas Medical Center (IND #127673). Sample procurement conformed to the principles set out in the Department of Health and Human Services Belmont Report. The trial entitled FELINE studied Femara (letrozole) plus ribociclib (LEE011) or placebo as neoadjuvant endocrine therapy for women with ER-positive, HER2-negative early breast cancer. Postmenopausal women with pathologically confirmed non-metastatic, operable, invasive ER breast cancer (tumor size >2 cm; ≥66% ER+ cells or ER Allred score 6–8 and HER2-negative by ASCO-CAP guidelines) were enrolled at 10 centers across the United States.

One hundred and twenty patients were randomized equally across three treatment arms (40:40:40), receiving: (A) letrozole plus placebo, (B) letrozole plus ribociclib 600 mg daily for 21 out of 28 days of each cycle or (C) letrozole plus ribociclib 400 mg continuously. Blinding was achieved through placebo administration. Protocol therapy continued until the day before surgery (~day 180). During treatment, patients provided three mandatory core tumor biopsies (via 14-gauge needle) at screening (pre-treatment: Day 0), cycle 1 follow-up (early follow-up: Day 14), and at end of trial (Post treatment: Day 180). Biopsy samples were immediately snap-frozen and embedded in optimal cutting temperature (OCT). All patients gave informed consent. Protocols were approved by the University of Kansas Institutional Review Board (protocol #CLEE011XUS10T) and followed in accordance with the Declaration of Helsinki.

The 120 patients were previously divided into two equally sized cohorts: a hypothesis-generating discovery cohort and a validation cohort (Griffiths et al, 2021). Patient demographic metrics and tumor treatment response outcomes were determined at the onset of that study and remained unchanged. The discovery and validation cohorts were independently sequenced following equivalent procedures and in subsequent analyses the validation cohort was used to replicate and independently verify key observations first detected in the discovery cohort.

### Single nuclei RNA sequencing, processing, and cell-type annotation

Single-cell RNA sequencing was performed on single nuclei suspensions extracted from OCT-embedded core tumor biopsies using 10X Genomics Chromium platform (as previously described (Griffiths et al, 2021; Sei et al, 2018)). Sample preparation and data acquisition were consistent between cohorts. Sequence reads were processed with BETSY and CellRanger v3.0.2. and aligned to reference genome (GRChg38) using STAR v2.6.0 8, providing, for each sample, a gene-barcode count matrix of unique molecular identifiers (UMIs) for each gene in each cell (Liao et al, 2014).

Transcriptional profiles of 424,581 single cells were obtained from 173 serial tumor samples across 62 patients (35 from discovery cohort (12 letrozole alone, 23 combination ribociclib) and 27 from validation cohort (11 letrozole alone, 16 combination ribociclib)). This followed application of stringent quality controls to ensure high coverage, low mitochondrial content, and doublet removal (as described in (Griffiths et al, 2021, 2025)). Patients whose biopsy samples were excluded had either: low RNA integrity (RIN scores), insufficient cells of high quality or did not provide serial timepoint measurements. On average, we recovered 2.75 (out of 3) timepoint samples per patient tumor. Cancer and non-cancer cell types were annotated using unbiased clustering and differential expression analysis in combination with application of singleR, ImmClassifier and inferCNV pipeline. Annotations were verified using UMAP analysis and cell-type-specific marker genes expression (Aran et al, 2019; Tickle et al, 2019; Becht et al, 2018; Liu et al, 2021) (annotation workflow detailed in (Griffiths et al, 2021, 2025)).

### Defining phenotypes using single sample Gene Set Enrichment Analysis (ssGSEA)

The count matrix of each cell type was filtered to keep genes expressed in at least 10 cells and normalized using the "zinbwave" R package v1.8.0 (Risso et al, 2018), using total number of counts, gene length and GC-content as covariates (K = 2, X = "~log (total number of counts)", V = " ~ GC-content + log (gene length)", epsilon=1000, normalizedValues = TRUE). The ssGSEA scores of 50 hallmark signatures (MSigDB, hallmark) and 4725 curated pathway signatures (MSigDB, c2) (Liberzon et al, 2011) were calculated for each cell based on the normalized count matrix using R package "GSVA" v1.30.0 (kcdf = "Gaussian", method = 'ssgsea') (Hänzelmann et al, 2013).

### Identifying consistent resistance phenotypes across tumors

To determine how cancer cell phenotypes evolved during treatment, we first identified patient tumors with paired pre- and post-treatment scRNAseq data and a persistent cancer cell population at EOT (>20 cancer cells profiled in both samples). No sample bias was detected when we compared the faction of responding/non-responding tumors between the patients with paired samples and the full cohort. Cancer cells from these patients ($n = 134,218$ cells (87,913 pre-treatment; 46,305 post treatment), 41 paired pre-/post-treatment tumor samples) were analyzed using hierarchical regression to assess how the distribution of single-cell phenotypes within each tumor evolved pre- and post treatment, allowing identification of consistently emerging phenotypes.

For each of the Hallmark and C2-level ssGSEA pathway signatures (phenotype P), the shift in pathway activity of cancer cells ($j$) in tumor $i$ over time ($t$) was examined across the trial using a hierarchical random effects model (Bolker et al, 2009) with the

following linear predictor and error structure:

$$P_{ij} = (\beta_0 + u_{0i}) + (\beta_t + u_{ti})t + e_{ij}$$

$$\begin{bmatrix} u_{0i} \\ u_{ti} \end{bmatrix} \sim N(0, \Omega_u)$$

$$\Omega_u = \begin{bmatrix} \sigma_{u_0}^2 & \sigma_{u_{0t}} \\ \sigma_{u_{0t}} & \sigma_{u_t}^2 \end{bmatrix}$$

$$e_{ij} \sim N(0, \sigma_{e_0}^2)$$

The model estimated the expected pre-treatment pathway activity across the cohort of patient tumors ($\beta_0$) and the expected activation of the pathway during treatment ($\beta_t$). Tumor-specific pathway activity of cancer cells pre-treatment ($u_{0i}$) and tumor-specific responses to treatment ($u_{ti}$) were accounted for.

Pathway (in)activation in response treatment was identified using likelihood ratio tests with false discovery rate (FDR) significance correction applied to P-value estimates. The likelihood of the full model was compared against that of the nested null model assuming no change in pathway activity (fixing $\beta_t = 0$). The likelihood function for each of these models was:

$$L(\hat{P}_{ij}, \sigma_D) = \prod_{I=1}^{N} \frac{1}{\sqrt{2\pi\sigma_D^2}} e^{-\frac{(P_{ij} - \hat{P_{ij}})^2}{2\sigma_D^2}},$$

were $\hat{P}_{ij}$ is the expected pathway activity, $\sigma_D^2$ is the variance and N is the number of single cells. Significant pathway activation (non-zero $\beta_t$) was identified using a two-tailed $t$ test, with the Satterthwaite method used to perform degree of freedom, t-statistic and P-value calculations, using the "lmerTest" R package (Kuznetsova et al, 2017). The hierarchical regression model described the paired pre- and post-treatment sampling of each tumor and this non-independence of cells within a sample determined the effective degrees of freedom in statistical tests. This revealed the phenotypic changes of post-treatment-resistant cancer cells that were consistent across patients. The same analysis was repeated at the gene level examining log(1+x) transformed single-cell expression scores. For each patient with paired pre- and post-treatment cancer cell (>20 cancer cells profiled in both samples) and persistent cancer cell population at EOT, we quantified the change in average square root transformed gene expression following treatment. We visualized this change in expression for the 25 most up and downregulated genes across patients (genes with greatest expected fold change post treatment across patients and FDR-corrected significance).

### Measuring composite phenotype scores

For pathways with correlated activity within cells, PCA was used to perform dimension reduction. The higher dimensional set of pathway scores related to a key phenotype (e.g., ERBB signaling) was projected into a low-dimensional latent space, preserving the global structure and providing a summary of the dominant patterns of variation across single cells. Top principal component dimensions were used to measure the composite phenotype score, by

quantifying the major axis of phenotypic variation that correlates with diverse ERBB ssGSEA pathways.

## Deciphering cell-type communications within patient tumors

### Characterizing within cell-type phenotypic heterogeneity

Diverse non-cancer cell subpopulations and phenotypically heterogeneous cancer lineages were characterized on a cell-type-by-cell-type basis. For each broad cell type, we generated a cell-type-specific UMAP projection, based on ssGSEA pathway scores (4775 pathways per cell; c2+hallmark). We used UMAP dimension reduction as it outperforms linear unsupervised embedding techniques in preserving local structure with the phenotype landscape and provides global quantification of heterogeneity. The use of ssGSEA scores contributed to dimension reduction, and the zinbwave' normalization process also corrected for biases associated with single-cell read counts and gene length and GC-content. The intrinsic (latent) dimensionality of the UMAP model was determined for each cell type using the packing number estimator (Aran et al, 2019; Erba et al, 2019; Becht et al, 2018). We then subdivide each cell type into subtypes of at least 30 cells with coherent phenotypes and of equal interval width along each phenotype axis. This approach allowed cell types with relatively continuous phenotypic variation to be subdivided into an ordered set of cell states along multiple axes of phenotypic heterogeneity and maintains phenotype covariance structure. We then calculated the relative abundance of each subpopulation of each major cell type within a tumor sample ($\hat{P}_i$ is the vector of the proportion of cells of subtype $i$ in a tumor sample).

### Defining ligand–receptor communication pathways

We next used a curated LR communication database (Ramilowski et al, 2015) to define a set of 1444 LR communication pathways between cells based on known protein–protein interactions. We calculated activity of each LR communication pathway ($k = 1{:}1444$) between each pair of sending ($i$) and receiving ($j$) subpopulations ($i \rightarrow j$) within a tumor. Single-cell ligand and receptor expression for genes in pathway $k$ were extracted, and for each cell subpopulation ($i$) we used mean CPM as a metric of signal production ($x_k$) and mean CPM receptor expression to quantify signal receipt by a focal cell ($y_{jk}$). Following the expression product method, multiplication of these two gives the average individual-level cell-cell interactions between two typical cells of the signaling sending and receiving subpopulation. To account for the relative abundance of each signaling cell subtype, signaling received by a cell of subpopulation ($j$) from cells of subpopulation $i$ via LR pathway $k$ was measured by: $C_{i \rightarrow j,k}(x_{ik}, y_{jk}) = (y_{jk} x_{ik} \hat{P}_i)$.

### Strength of communication between cell types

To measure communications that a focal cell received from broad cell-type populations via the LR pathway $k$, signals from phenotypically diverse subpopulations ($1:n$ ligand-producing subpopulations of a broad cell type) were totaled, and signals to receiving cell subpopulations were averaged (across $1:m$ signal-receiving subpopulations of a broad cell type) proportional to the receiving subpopulation's abundance, using the weighted average:

$$C_{i_{1:n} \rightarrow \bar{j},k}\left(x_{i_{1:n}k}, y_{j_{1:m}k}\right) = \frac{\sum_{z=1}^{m}\left(C_{i_{1:n} \rightarrow j_z k}\left(x_{i_{1:n}}, y_{j_z k}\right)\hat{P}_{j_z}\right)}{\sum_{z=1}^{m}\left(\hat{P}_{j_z}\right)}.$$

We define this as the strength of communication from cells of one cell-type population to a typical cell of the signal-receiving cell type. This measures the stimulation a focal cell receives from different cellular components of the TME accounting for tumor composition and cellular phenotypic heterogeneity in signaling.

As an example, signaling received using receptor $y$ by a fibroblast (*Fibro*) with phenotype $z$ from $n$ heterogeneous cancer cell populations (*Cancer*$_{1:n}$) producing differing amount of ligand $x$ is given by:

$$C_{Cancer_{1:n} \rightarrow Fibro_z, k}\left(x_{Cancer_{1:n}k}, y_{Fibro_z k}\right) = y_{Fibro_z k} \sum_{i=1}^{n} x_{Cancer_i k} \hat{P}_{Cancer_i}$$

The strength of communication a typical fibroblast in a sample ($\overline{Fibro}$) received from the diversity of cancer cells is given by:

$$C_{Cancer_{1:n} \rightarrow \overline{Fibro}, k}\left(x_{Cancer_{1:n}k}, y_{Fibro_{1:m}k}\right) =$$

$$\frac{\sum_{z=1}^{m}\left(C_{Cancer_{1:n} \rightarrow Fibro_z, k}\left(x_{Cancer_{1:n}k}, y_{Fibro_z k}\right)\hat{P}_{Fibro_z}\right)}{\sum_{z=1}^{m}\left(P_{\widehat{Fibro_z}}\right)}$$

The strength of communication was calculated between each broad cell-type pair for each LR communication pathway separately for each pre- and post-treatment tumor sample. Finally, due to the distinct potency of each communication pathway to modulate cellular phenotype and behavior, communication pathway scores were standardized across patients (mean=0, sd=1), preventing communication inferences from being dominated by broadly highly expressed LR pairs.

### Cell-type communication pathway analysis: identifying communications responding to treatment

For each LR communication pathway, we contrasted the strength of communication between cell types in tumors pre- versus post treatment. We used log-linear model to describe temporal trends in cell-type communication across tumors (comparing $\log(1 + C_{i_{1:n} \rightarrow \bar{j}, k}(x_{i_{1:n}k}, y_{j_{1:m}k}))$ scores of tumors before and after treatment). Significant changes in the strength of cell-type communication between cell types during treatment were identified using ANOVA on the endpoint data (Day 0 and 180 separately) and accounting for multiple comparisons using false discovery rate (FDR) p-value correction.

### Summarizing communication across multiple ERBB LR communication pathways

Many signaling ligands bind to receptors to activate convergent intracellular signal transduction pathways. To analyze temporal changes in signaling between cell types across multiple functionally related LR pathways, we applied a linear mixed model. Specifically, this method identified the overall trend in the strength of communication ($\log(1 + C_{i_{1:n} \rightarrow \bar{j}, k}(x_{i_{1:n}k}, y_{j_{1:m}k}))$) a cell type received, with a covariate separating trends in signaling via each receptor. A hierarchical random effects component was additionally included to account for the variable trend of each specific LR pathway and the non-independence that multiple LR pathways were measured within each tumor sample. Significant trends in the strength of communication across multiple LR pathways were identified with dysregulation of communication via each receptor being separately identified using a two-tailed $t$ test, and with the Satterthwaite method used to calculate the degree of freedom, t-statistic and $P$ value. We used this approach to identify changes in ERBB growth

factor signaling received by cancer cells from: (i) specific cell types such as fibroblasts and (ii) across all cells in the TME, summing signaling contributions of every subpopulation in a tumor sample.

### Visualizing major sources of GF communications between cell types

The average strength of communication from one cell type to another via multiple related LR pathways (e.g. all ERBB ligand–receptor pairs) was measured by averaging the transformed communication scores ($\overline{C}_{i_{1:n} \rightarrow \bar{j}} = mean(\log(1 + C_{i_{1:n} \rightarrow \bar{j}, k}(x_{i_{1:n}k}, y_{j_{1:m}k})))$) within each tumor sample. We then calculated the fold change in average communication between cell types pre- and post treatment. To construct a network visualization of the expected GF communication between cell types, we summarized the mean across tumors. Post-treatment GF communication networks between cell type were represented by directed weighted network graphs using ggraph r package (v2.1.0).

### Fibroblast phenotype reconstruction

The diversity of fibroblast phenotypes was examined through intrinsic dimensionality estimation and UMAP analysis of scRNAseq transcriptional profiles ($\log(1 + CPM)$). Genes with greater than 5% coverage in fibroblast cells were used. The packing number estimator (Erba et al, 2019) was first applied to estimate the intrinsic dimensionality of the fibroblast phenotype landscape (i.e., the number of major axes of phenotypic variation). UMAP analysis was then applied to project the fibroblasts transcriptional profiles into this number of dimensions. We verified that the major axes of phenotype variation identified were weakly correlated and therefore reflect biologically distinct features. The phenotype reflected by each dimension were identified through trajectory-based analysis, to make use of the near-continuous resolution of scRNAseq phenotypic heterogeneity. Following the tradeSeq approach (Van den Berge et al, 2020), generalized additive models were used to identify genes with nonlinearly and dynamically (smoothly) changing expression along each axis. The significance of dynamical changes in gene expression were determined by the F value of the predictor smooth term (penalized thin plate regression spline). The 25 most dynamically varying genes were used to interpret the biology of phenotypic gradients. These dynamic changes in expression were verified by examining correlation with ssGSEA pathway scores and through gene overlay of marker genes. The key phenotypes reflected mesenchymal fibroblast differentiation (high hallmark EMT ssGSEA scores, and high collagen, fibronectin and ECM secretion protein expression) (Wu et al, 2020; Glabman et al, 2022) and activation of EGFR-driven proliferation (high biocarta EGF and ERK proliferation ssGSEA scores) (Meran et al, 2011; Midgley et al, 2013) (Fig. EV3A) (Appendix Figs. S3 and S4).

### Comparing ERBB growth factor signaling across fibroblast differentiation states

We first classified fibroblasts into high and low mesenchymal differentiated states. Highly differentiated cells were defined as having a greater than average UMAP mesenchymal differentiation score (UMAP score of zero). A plateau of the increase in collagen and fibronectin markers was seen above this cutoff. We then extracted the fibroblast scRNAseq gene expression of ERBB ligand genes detected in the communication analysis (NRG1-3, EGF, AREG, HBEGF, TGFA, EMA4D, HLA-A, EFNB1, ICAM1, GNAI2, CDH1, AREG, ANXA1, ADAM17). The ERBB signaling of each fibroblast cell was then estimated by averaging across the

expression of these genes after each was log(1+x) transformed, scaled, and centered (mean=0, sd=1). To account for the non-independence of fibroblasts within a tumor sample, we finally averaged the ERBB signaling of highly and lowly mesenchymal differentiated fibroblasts within each tumor sample. We then compared the ERBB signaling between highly and lowly mesenchymal differentiated populations across tumors using ANOVA.

## Validating patient-derived predictions in replicate in vitro model systems

We tested findings from the single-cell patient data analyses in separate in vitro 3D spheroid growth systems. We developed in vitro models of acquired endocrine and cell cycle therapy resistance in estrogen receptor-positive (ER +) breast cancer cell lines. The breast cancer cell lines CAMA-1, MCF-7, T47D and the lung fibroblast cell line MRC5 were obtained from the American Type Culture Collection (ATCC). The CAMA-1, MCF-7, MRC5 cell lines were grown in DMEM with a 10% FBS and 1% antibiotic–antimycotic solution, while T47D was cultured in RPMI with 10% FBS and 1% antibiotic–antimycotic solution. Regular testing for mycoplasma contamination was conducted using the commercially available Myco Alert kit from Lonza. All cell lines were verified by STR profiling. The drugs listed below were acquired from Selleck Chemicals: Ribociclib (S7440, CDK4/6 inhibitor), Afatinib (S1011, targeting ErbB1-4), and Fulvestrant (S119, estrogen receptor antagonist).

### Identification and generation of endocrine and combination ribociclib-resistant cell lines

We selected three ER+ breast cancer cell lines (CAMA-1, MCF-7 and T47D) with above-average endocrine therapy resistance (ETR), based on published proliferation-corrected endocrine sensitivity estimates from a large-scale drug screen (Hafner et al, 2017) (Appendix Figs. S1 and S2). Three combination ribociclib-resistant (CRR; evolved) cell lines were generated through the long-term culture of parental ETR cell lines in increasing concentrations of ribociclib for 6–12 months (Grolmusz et al, 2020; Emond et al, 2023). These cell lines were then maintained in culture medium supplemented with 250 nM ribociclib for CAMA-1 ribociclib-resistant cells, 1.5 μM ribociclib for MCF-7 ribociclib-resistant cells, and 5 μM ribociclib for T47D ribociclib-resistant cells. To quantify the extent of resistance, we performed dose–response assays with increasing concentrations of fulvestrant or ribociclib in endocrine therapy and combination ribociclib-resistant cells. Combination ribociclib-resistant cells showed a significantly reduced response to fulvestrant and ribociclib compared to endocrine therapy-resistant cells (Appendix Fig. S2).

### Lentiviral labeling of fibroblast and endocrine or combination ribociclib-resistant cancer cells

Endocrine and combination ribociclib-resistant cancer cells and fibroblasts (MRC5) were labeled with lentivirus to express a distinct fluorescent protein for monitoring each population's growth when cultured (mCherry fluorescence for fibroblast cells, Venus fluorescence for ribociclib-sensitive (endocrine therapy-resistant) cancer cells, and Cerulean fluorescence for combination ribociclib-resistant cancer cells). Briefly, lentiviruses incorporating Venus (LeGO-V2), Cerulean (LeGO-Cer2) and mCherry (LeGO-C2)

fluorescent proteins were generated using Lipofectamine 3000 reagent (Thermo Fisher Scientific) according to the manufacturer's instructions. LeGO-V2, LeGO-Cer2 and LeGO-C2 vectors were gifts from Boris Fehse (Addgene plasmids #27340, #27338, and #27339) (Weber et al, 2008). Fibroblast cell line MRC5 was transduced with mCherry-containing lentivirus using reverse transduction, resulting in MRC5-C2. Endocrine therapy-resistant (ETR: ribociclib-sensitive) and combination ribociclib-resistant (CRR) cancer cell line pairs of CAMA-1, MCF-7, and T47D were, respectively, transduced with Venus- (for ribociclib-sensitive cell lines) or Cerulean- (for ribociclib-resistant cell lines) containing lentivirus. This resulted in three fluorescently labeled pairs of endocrine therapy-resistant (ETR) cell lines (CAMA-1 ETR/ CAMA-1_V2, MCF-7 ETR/MCF-7_V2, and T47D (ETR)/ T47D_V2) or combination endocrine and ribociclib-resistant cell lines (CAMA-1 (CRR)/CAMA-1_riboR_Cer2, MCF-7 (CRR)/ MCF-7-1_riboR_Cer2, T47D (CRR)/T47D_riboR_Cer2). For virus transduction, 1 ml of polybrene-containing cell suspension of 200,000 cells were plated in a well of a six-well plate, where 0.5 ml of viral aliquot was previously dispensed. Cells were incubated for 48 h at 37 °C and 5% $CO_2$, after which cells were washed, and fresh regular culture medium was applied. Fluorescently labeled cells were selected using fluorescence-activated cell sorting after further subculture of transduced cells to attain homogeneously labeled cell populations.

### Cancer monoculture ERBB inhibitor effect assays

We initially assessed the anti-proliferative efficacy of the pan-ERBB inhibitor afatinib in each of the cancer cell lines using 3D spheroid growth assays. For cancer cell monocultures, spheroids were treated with fulvestrant plus afatinib in medium containing 10% FBS for a total of 18 days, with imaging performed every 2–3 days. Cells were plated at a density of 2000 cells per well for CAMA-1 and 5000 cells for MCF-7 and T47D in a volume of 100 μL of medium. After 24 h, an additional 100 μL of medium containing 2× the concentration of the drug being tested was added. For the experiment, imaging and media change was then performed every 4th and 7th day of the week.

### Cancer-fibroblast coculture experiments

To assess the ability of fibroblasts to facilitate cancer growth, we performed 14-day 3D cancer-fibroblast coculture experiments using the MRC5 fibroblast cell line. For each cancer cell line, we compared the rate of cancer cell number increase between mono- and cocultures initiated with the same number of cancer cells. Briefly, 10,000 cancer cells and different ratios of fibroblast cells (1:4, 1:2, and 1:1, 0:1 fibroblast to cancer cell) were plated with phenol red-free DMEM containing 5% FBS and antibiotic–antimycotic solution in 96-well round-bottom ultra-low attachment spheroid microplate (Corning, Cat. No.: 4520). Replicate populations were also plated with 10,000 fibroblasts and no cancer cells to examine monoculture fibroblast growth. After 24 h, spheroids were imaged, washed twice, and replenished with fresh phenol red-free DMEM containing 1% FBS and treatment drugs. Subsequently, imaging and media change was performed every 4th and 7th day of the week.

### Quantifying cell abundance and facilitation

Spheroids were then treated every three days and concurrently imaged on 5 occasions (Days: 1, 4, 8, 11,14). Imaging was

performed using Cytation 5 imager (Biotek Instruments) gathering signal intensity from brightfield, YFP (for Venus fluorescence), Texas Red (for mCherry fluorescence), and CFP FRET V2 (for Cerulean fluorescence) channels (Grolmusz et al, 2020). Raw data processing and image analysis were performed using Gen5 3.14 software (Biotek Instruments). Briefly, the stitching of $2 \times 2$ montage images and Z-projection using focus stacking was performed on raw images followed by spheroid area analysis. Fluorescence intensities were used to quantify each cell type's abundance over time after constructing an experimental standard curve to map known cell abundances to fluorescence intensities using linear regression. Each replicate population's cancer growth rate was measured using an exponential growth model, fitted using least squares. Fibroblast growth rate was measured using the same approach. Facilitation of one cell type by another was identified by an increase in growth rate under coculture, compared to monoculture, using linear mixed-effects models to account for cancer cell line-specific coculture responses. We applied the mutualism quantification analysis to spheroid growth data from cocultures with patient-derived macrophages and CD8 + T cells. As expected, these negative control cell types showed no mutualism, with significant growth suppression being detected across cancer cell lines in CD8 + T-cell cocultures. Comparisons of individual cell line growth rates were also made between mono- and coculture conditions using ANOVA. To assess the ability to intensify or block fibroblast facilitation of cancer growth, we also treated mono- and cocultures with fulvestrant (0, 1, 5, or 25 nM as shown in figures) or afatinib (2.5–10 µM) and across treatments we compared cancer and fibroblast growth rates between monocultures and cocultures, again using linear mixed-effects models.

### Imaging cancer-fibroblast coculture morphology: SPIKE assay embedded fluorescent images

To explore the morphological impacts of fibroblast coculturing on spheroid growth and invasion, we cocultured cancer and fibroblast cells in a physiologically relevant matrix structure utilizing a transwell collagen-based matrix (Spheroid Phenotypes In Key Environments (SPIKE) assay). First, 20,000 cancer cells (CAMA-1 endocrine therapy-resistant (ETR) or combination ribociclib-resistant (CRR)) and 10,000 fibroblasts (MRC5) per well were grown for 48 h to allow for cell aggregation and spheroid formation by growing in an Ultra-low attachment spheroid plate (Costar, Cat# 4520). After 48 h, spheroids were transplanted into the collagen matrix using a two-step process as previously described (Ootani et al, 2009) and modified as described below. Using 30 mm transwell inserts (0.4 µm PTFE hydrophilic, Millipore Sigma, Cat# PICM03050) placed in a 6-well tissue culture plate, 1 mL of 1× DPBS (Gibco, Cat# 14190) was pipetted directly onto the transwell membrane to facilitate hydration of the membrane. 1x PBS was removed from the transwell and an acellular basement collagen layer was prepared by quickly adding 1 mL of ice-cold collagen matrix solution (DMEM (Gibco, Cat# 11995) + 33% collagen CellMatrix Type I-A (Nitta Gelatin Inc., Cat# 631-00653) + 10 mM HEPES pH 7.4 (Thermo Fisher Scientific, Cat# 15630106)) was added to each well, maintained at 4 °C to prevent solidification prior to plating. The basement collagen was then incubated at 37 °C + 5% $CO_2$ for 1.5 h

to allow for collagen matrix solidification. A fresh batch of 1 mL of ice-cold collagen matrix was then prepared and layered on top of the transwell-solidified collagen matrix layer. Immediately, using a large bore pipette tip, the cancer-fibroblast spheroids ($N = 3$ per transwell) were transplanted into the upper aqueous collagen matrix. In the SPIKE assay, the collagen-embedded spheroids were then incubated at 37 °C + 5% $CO_2$ for 1.5 h to allow for collagen matrix solidification. The transplanted spheroids were equilibrated prior to drug treatment by placing 1.2 mL of DMEM + 1× anti-anti + 1% hiFBS in the exterior reservoir of the transwell in the 6-well plate and maintained at 37 °C + 5% $CO_2$. After 8-18 h, spheroids were treated with DMSO (0.3%), 10 nM fulvestrant, 10 µM afatinib, or 10 µM fulvestrant + 10 µM afatinib in DMEM + 1× anti-anti + 1% hiFBS. After 4 days of treatment, media in exterior reservoir was replaced with fresh drug treatment media treatments increasing fulvestrant to 25 nM.

SPIKE assay coculture spheroid growth was observed after 7 days by florescent imaging using an Axio Observer 7 inverted microscope (Carl Zeiss) controlled with Zen software (3.6 Blue edition), equipped with a 5 × 0.16NA Plan-Apochromat objective, and captured on an Axiocam 702 mono camera. Z-stacks and tiles were acquired as $2 \times 2$ tiles with 5 slices at 50 µm intervals. Excitation and emission were provided by a Colibri 7 and Filter Set 90 HE and images were captured under EGFP (CAMA-1 (ETR), Venus), Cy3 (MRC5, mCherry), DAPI (CAMA-1 (CRR), cerulean) and Brightfield channels. Images were stitched, orthogonally projected (maximum), and subset images of ROIs were created with Zen (3.6 Blue edition).

### Isolation of patient-derived cancer-associated fibroblasts (CAFs)

Primary breast cancer samples were collected from malignant pleural effusions with proper written informed consent and ethical compliance under IRB #41030 and 89989 (University of Utah) and utilized under IRB #20357 (City of Hope). Briefly, after fluid drainage, cells were pelleted at 300×g for 5 min, and resuspended in TAC buffer (17 mM Tris, pH 7.4, and 135 mM NH4Cl), followed with incubation at 37 °C for 5 min and another centrifugation (300×g for 5 min). After repeating these steps several times until red blood cells were completely depleted from pellet, the remaining cells were washed with PBS twice and frozen in 90% FBS with 10% dimethyl sulfoxide (DMSO). For CAF isolation, the thawed vials were added into 9 ml of media in a 15 ml conical tube and centrifuged at 300×g for 5 min. After removal of the supernatant, cell pellets were resuspended in 10 ml of media in a 100-mm dish and incubated for 30 min at 37 °C. Then, the suspended cells were removed again, and the adherent cells were washed with PBS (2x) and cultured with fresh media (Renaissance Essential Tumor Medium with Supplement, Cellaria #CM-0001, plus 5% FBS and 1% anti-anti) at 37 °C for 1–2 weeks until the dish became confluent for CAFs collection. To verify CAFs purity, the adherent cells were collected with trypsin-EDTA once, washed once with flow buffer (PBS with 1% BSA), and stained with FITC labeled anti-human CD90 (Miltenyibiotec, #130-117-684) and PE labeled anti-human EpCAM (Miltenyibiotec, #130-113-264) followed by flow cytometry testing for double positive selection. CAFs were also subjected to qPCR assays for fibroblast makers identification.

### Predicting dynamical consequences of a cancer-fibroblast mutualism

The mutualistic signaling between cancer and fibroblast populations was described by a system of ordinary differential equations. This dynamic model describes the division of cancer and fibroblast cells ($r_C$ and $r_F$ indicate intrinsic division rates of cancer and fibroblast cells in monoculture). Cell division is assumed to be regulated by density-dependent competition for resources (e.g. growth factors/nutrients/oxygen) as the population approaches the carrying capacity ($K$). Cancer cell (but not fibroblast) division is assumed to be reduced by fulvestrant therapy ($X$) following a hyperbolic decay function. Fulvestrant doses were rescaled so that $X$ ranges between zero (fulvestrant=0 nM; DMSO control) and 1 (fulvestrant=5 nM). Mutualism between cell types is introduced by allowing each cell population to stimulate the division of cells of the opposite type (mutualism effects on cancer and fibroblast cells are $\beta_C$ and $\beta_F$ respectively). Cell death ($\mu$) is assumed to be constant. The initial abundance of each cell type (experimental seeding number: $C_0$ and $F_0$) are assumed to be known. This led to the following system of differential equations:

$$\frac{dC}{dt} = r_C \left( \frac{1 + \beta_C F}{1 + X} \right) C \left( 1 - \frac{C}{K_C} \right) - \mu C$$

$$\frac{dF}{dt} = r_F (1 + \beta_F C) F \left( 1 - \frac{F}{K_F} \right) - \mu F$$

Intrinsic division of each cell type and their death rate (both units=cells per hour) were estimated from the exponential growth measurements of monoculture populations. Linear models were used to quantify log-linear population growth with and without fulvestrant treatment. Assuming that proliferation is halted by endocrine therapy, whilst cell death is relatively unaffected, we quantified the death rate as the log-linear population decline under fulvestrant treatment. Intrinsic division was then measured as the log-linear population growth under control conditions plus the estimated death rate. Mutualism effects (units=per cocultured cell) were similarly estimated by contrasting log-linear growth rates of cocultures, during the exponential population growth phases, with the growth of cancer monocultures. The difference in growth rate was divided by the other cell type's abundance to arrive at the estimates of $\beta_C$ and $\beta_F$. The carrying capacity of each cell type ($K_C/K_F$; units=cells) was fixed to be 100 times $C_0$.

The model was used to simulate continuous time population abundances of cancer and fibroblast cells using numerical integration. The lsoda solver of the "deSolve" R package was used to switch automatically between stiff and non-stiff methods. To evaluate the evolution of the system from initial conditions to steady state, under a given set of parameters, we used the "rootSolve" R package's steady function to apply an iterative steady-state solver using the Newton–Raphson method.

### Quantitative real-time PCR analysis of treatment-induced gene expression changes

We cultured replicate monoculture populations of fluorescently labeled ER+ breast cancer cell lines (endocrine or combination ribociclib-resistant CAMA-1, MCF-7, and T47D) under cell cycle inhibitor (ribociclib), endocrine (fulvestrant) and combination therapy and DMSO control conditions for 7 days with 1% FBS. We then compared qPCR-measured mRNA expression of signaling ligands and GF receptor genes, using linear mixed-effects models and ANOVA. Monoculture fibroblast (MRC5) populations were cultured under a series of doses of TGFβ1 (0, 5, 20, 100 ng/mL; Peprotech, #100-21) for 3 days and patient-derived fibroblasts ($n = 6$ patient-derived populations) were cultured with or without TGFβ1 (20 ng/mL) for 4 days, respectively. We then compared mRNA expression of signaling ligands and fibroblast differentiation markers between inactive (untreated) and activated (TGFβ1 primed with 20 ng/mL) replicates. For gene expression of 3D cocultured cancer and fibroblast cells, FACS sorting was performed to isolate the fluorescence-labeled cells following spheroid disassociation with trypsin-EDTA after 3 days drug treatments. Monocultures of cancer and fibroblast cell were 3D cultured under the same conditions for an equal duration to serve as the control. To evaluate mRNA expression, RNA extraction was performed using the RNeasy Mini kit from QIAGEN. cDNA synthesis was then carried out utilizing the Maxima First Strand cDNA Synthesis Kit from Thermo Scientific. The resulting cDNA was analyzed via quantitative real-time PCR (qRT-PCR) using the Fast SYBR Green Master Mix (Applied Biosystems, Thermo Scientific) as per the manufacturer's protocol. The mRNA expression levels of various genes were determined relative to RPLP0, with primer sequences detailed in Table EV4.

### Immunoblotting for endocrine induction of ERBB protein phosphorylation

Cells were washed and collected with cold PBS, then total proteins were extracted using the RIPA Lysis Buffer (Thermo Scientific) that contained 1 mM PMSF and a HaltTM protease and phosphatase inhibitor cocktail (Thermo Scientific). The protein concentration in the extracted lysates was determined using the BCA assay (Thermo Scientific). Equal amounts of denatured proteins were then separated using SDS-PAGE (4–20% Tris-Glycine Gel from Bio-Rad). Visualization was carried out using a chemiluminescence technique (Thermo Scientific) with a standard protocol. The separated proteins were then immunoblotted overnight in cold room using specific antibodies against ErbB protein phosphorylation such as Phospho-EGF Receptor (Tyr1068) (#3777), Phospho-HER2/ErbB2 (Tyr1221/1222) (6B12) (#2243), Phospho-HER3/ErbB3 (Tyr1289) (D1B5)) (#2842), Phospho-HER4/ErbB4 (Tyr1284)/EGFR (Tyr1173) (21A9) (#4757), HER2/ErbB2 Antibody (#2242), HER3/ErbB3 (1B2E) (#4754), HER4/ErbB4 (111B2) (#4795), from Cell Signaling, β-actin (C4) (sc-47778) from Santa Cruz Biotechnology, EGFR Antibody (EPR806Y) from Millipore Sigma.

To quantify protein levels, western blot intensities were measured using ImageJ. Intensities were normalized to β-actin and we then performed log2 transformation and calculated the log2 fold changes in normalized intensities compared to DMSO control in each cell line (i.e Fulvestrant =0 nM).

Linear mixed-effects models were applied to data for each protein across the six cell lines to identify the effects of fulvestrant ($\beta_{Fulv}$) and afatinib ($\beta_{Afat}$) treatment on their relative western intensities. The fixed effects component of the model was thus:

$$\mu_{fixed} = \beta_0 + \beta_{Fulv} \sqrt{Fulvestrant} + \beta_{Afat} Afatinib.$$

Here Fulvestrant and Afatinib represent drug concentrations and $\beta_0$ captures the baseline intensity.

To account for the replicate-specific variation in Western intensities, both in the DMSO control and the trend with

fulvestrant and afatinib treatment, we used a random slope and intercept random effects component in the model. Here the random replicate effects were nested within cancer cell line, leading to the random component:

$$\mu_{random} = (u_{0,L} + v_{0,L,r}) + (u_{Fulv,L} + v_{Fulv,L,r})Fulvestrant$$
$$+ (u_{Afat,L} + v_{Afat,L,r})Afatinib.$$

Here, $u_{x,L}$ indicates the random effect of lineage $L$ on parameter $\beta_x$. In addition, $v_{x,L,r}$ indicates the nested random effect of replicate $r$ within lineage $L$ on parameter $\beta_x$. Random effects $u_{x,L}$ and $v_{x,L,r}$ were assumed to follow multivariate normal distributions, and the residual error was assumed to be normally distributed, leading to the following model on normalized intensities ($Y_{L,r}$):

$$\log 2(Y_{L,r}) = \mu_{fixed} + \mu_{random} + \epsilon$$

$$\epsilon \sim N(0, \sigma^2)$$

The Satterthwaite method was used to perform degree of freedom, t-statistic and $P$-value calculations.

## Data availability

The datasets and computer code produced in this study are available in the following databases: Pre-processed single-cell RNA-seq gene expression data and relevant metadata: GEO (the Gene Expression Omnibus) under accession code GSE211434; Custom code used in analyses and to produce Figs. 1–6: GitHub (https://github.com/U54Bioinformatics/FELINE_project/tree/master/FELINE_ERBB_fibroblast_facilitation). Source data are provided with this paper.

The source data of this paper are collected in the following database record: biostudies:S-SCDT-10_1038-S44320-025-00104-6.

## Peer review information

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

## Acknowledgements

We thank the anonymous patients from the trial that made this study possible, and Patricia Bild for inspiring this research. We thank Anne O'Dea, Priyanka Sharma, Cynthia Ma, Meghna Trivedi, Kevin Kalinsky, Kari B Wisinski, Ruth O'Regan, Issam Makhoul, Laura M Spring, Aditya Bardia, Yuan Yuan, Lauren Nye, Onalisa Winblad, Jamie Wagner-Berbel, Kelsey Larson, Christa Balanoff, Gregory Crane, Fang Fan, Allison Aripoli, Amanda Amin, Richard McKittrick, Marc Hoffmann, Marc Inciardi, Cory Bivona, Mia Hard, Manana Elia, and Mark Redick for conducting the trial and contributing patient samples. We thank Adam Cohen for providing clinical insights. JG, AHB, PAC, AN, FA, and JC were supported by the National Cancer Institute of the National Institutes of Health (NIH) under award number U54CA209978 and U01CA264620. The content is solely the authors responsibility and does not necessarily represent the official views of the NIH. The High-Throughput Genomics Shared Resource was supported by the NIH Award Number P30CA042014. JTC was supported by a Cancer Prevention Research Institute of Texas Core Facility Support Award (RP170668). The authors also thank JKTG for providing funding for this research.

## Author contributions

**Jason I Griffiths**: Conceptualization; Resources; Data curation; Software; Formal analysis; Funding acquisition; Investigation; Visualization; Methodology; Writing—original draft; Writing—review and editing. **Feng Chi**: Formal analysis; Validation; Investigation; Methodology; Writing—original draft; Project administration; Writing—review and editing. **Elena Farmaki**: Validation; Methodology; Writing—review and editing. **Eric F Medina**: Data curation; Software; Validation. **Patrick A Cosgrove**: Validation; Methodology; Writing—original draft; Project administration. **Kimya L Karimi**: Validation; Investigation; Methodology; Writing—review and editing. **Jinfeng Chen**: Data curation; Software; Formal analysis. **Vince K Grolmusz**: Validation; Investigation; Methodology. **Frederick R Adler**: Conceptualization; Writing—review and editing. **Qamar J Khan**: Resources; Data curation; Project administration. **Aritro Nath**: Data curation; Software; Formal analysis; Writing—review and editing. **Jeffrey T Chang**: Conceptualization; Data curation; Formal analysis. **Andrea H Bild**: Conceptualization; Resources; Supervision; Funding acquisition; Investigation; Methodology; Writing—original draft; Writing—review and editing.

Source data underlying figure panels in this paper may have individual authorship assigned. Where available, figure panel/source data authorship is listed in the following database record: biostudies:S-SCDT-10_1038-S44320-025-00104-6.

## Disclosure and competing interests statement

Qamar Khan declares research funding from Novartis. Andrea Bild is a founder of Unravel Genomics, which builds biomarkers of drug response. Aritro Nath is co-founder of Unravel Genomics. The remaining authors declare no competing interests.

# Expanded View Figures

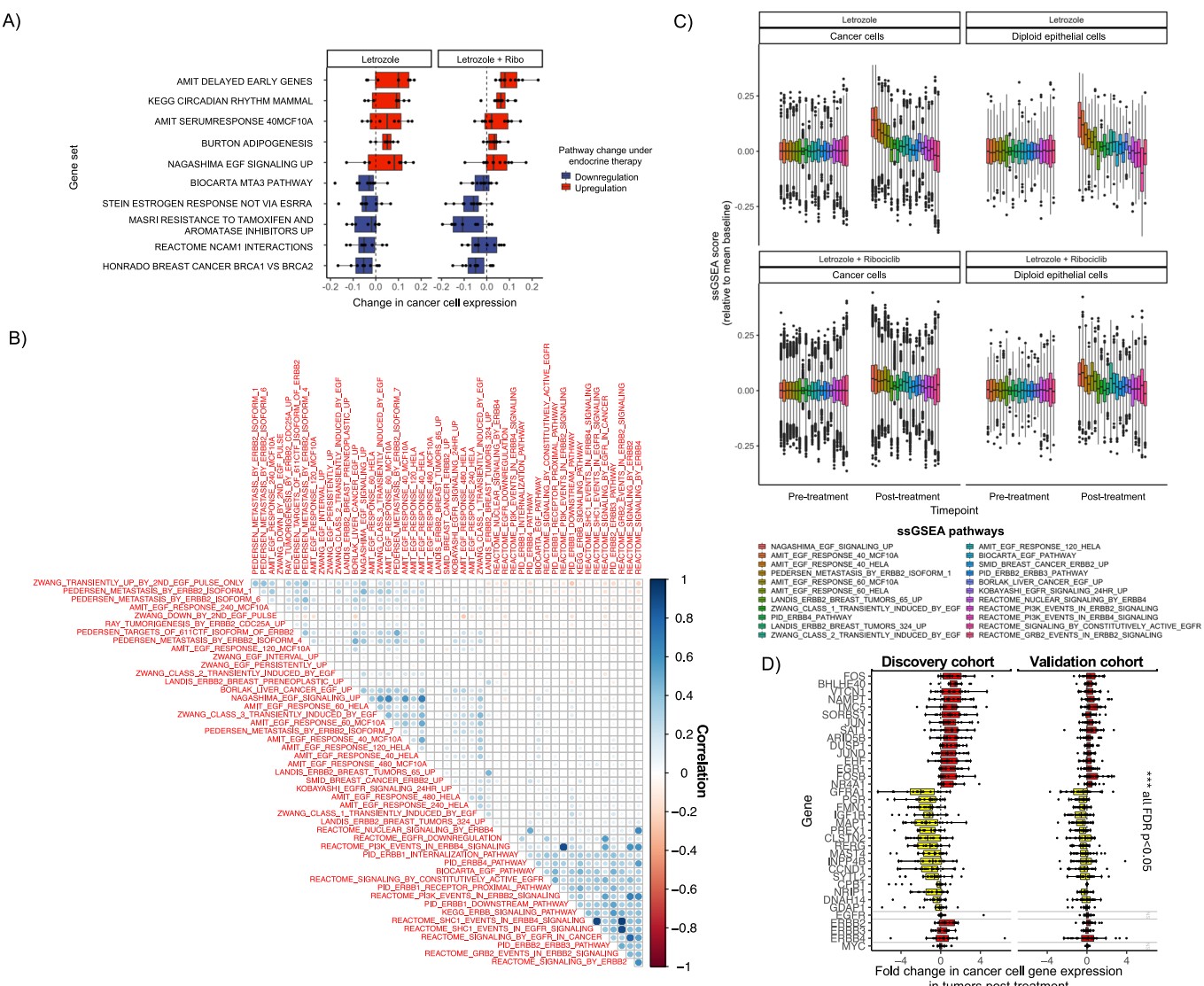

**Figure EV1. ERBB pathway upregulation during treatment in endocrine +/- CDK4/6i-resistant cancer cells of patient tumors.**

(A) Cancer phenotypic changes during endocrine and combination ribociclib treatment. Summary of the top 5 ssSGEA pathways upregulated (red) and downregulated (blue) following endocrine therapy (ET) in patients treated with letrozole alone (left) and combination ribociclib (ribo) treatment (right). Points show the difference in mean ssGSEA pathway scores of cancer cells in a tumor pre- versus post treatment (positive values = increase activity within tumor post treatment). Box elements show median (center line), upper and lower quartiles (hinges) and 1.5*interquartile range (whiskers=minima/maxima) of changes in pathway activity across patient tumors. (B) Positive correlation among ERBB pathway scores across cancer cells. Correlation plot among 47 ssGSEA pathways (axes labels) contributing to the overall ERBB activation score. Coloration indicates the strength of correlation between pairs of pathways (blue=high, red=low correlation). Point sizes proportional to the absolute value of the correlation coefficient. (C) ERBB pathway activation in cancer and diploid epithelial cells during treatment. Boxplots showing post-treatment ERBB pathway activation of cancer (left) and diploid epithelial cells (right) resistant to letrozole alone (top) or combination ribociclib treatment (bottom). Points show ERBB pathway scores (y axis) of single cells as measured by 22 different ssGSEA scores (colors), all showing overall significant increases in post-treatment resistant cancer cells. The extent of pathway activation in diploid epithelial cells was consistent with that observed in cancer cells. Box elements show median (center line), upper and lower quartiles (hinges) and 1.5*interquartile range (whiskers=minima/maxima) of changes in pathway activity across patient tumors. (D) Genes modulated in resistant patient cancer cells. Boxplot of the top 15 most consistently upregulated and downregulated genes (red versus yellow) in cancer cells following treatment. Across patient tumors, cancer cells resistant to treatment (sampled post treatment) showed consistent upregulation (red) of transcription factors downstream of ERBB signaling (e.g., FOS, FOSB, JUN and JUND) and downregulation (yellow) of genes involved in estrogen-dependent growth (e.g., PGR and MAPT). At the bottom, we present the changes in the four ERRB receptor genes and MYC, a key downstream transcription factor. Points indicate the fold change in the average gene expression of cancer cells of a tumor post treatment compared to pre-treatment (positive values= increased expression during treatment). Box elements represent median (center line), upper/lower quartiles (hinges), 1.5*interquartile range (whiskers=minima/maxima) of gene expression changes across tumors from the discovery cohort. All FDR-corrected ANOVA P values less than 0.05. Gene specific statistics (including exact P values) are provided in Table EV3. Sample sizes in (A–D): Discovery:: 57,403 tumor-derived cancer cells (36,825 pre-treatment, 20,578 post treatment) from 16 patients with paired pre- and post-treatment samples with >20 cancer cells (n = 32 samples), Validation:: 163245 tumor-derived single cells (51,088 pre-treatment, 25,727 post treatment) from 25 patients with paired pre- and post-treatment samples with >20 cancer cells (n = 50 samples).

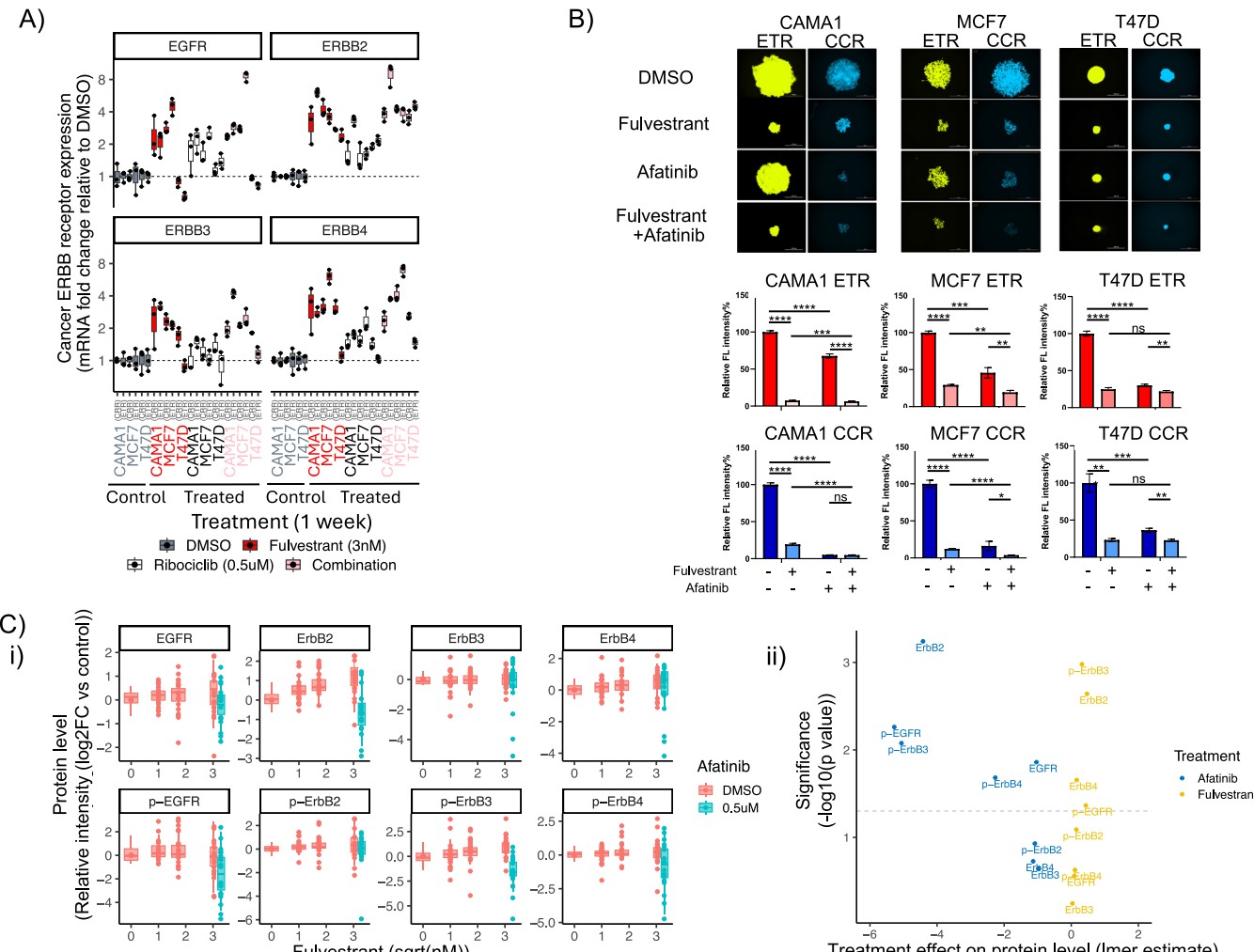

◀  **Figure EV2.  In vitro validation of ERBB pathway activation during endocrine or CDK4/6i treatment and the control of cancer cell growth by targeted therapy inhibiting the ERBB pathway.**

(**A**) Validation in vitro that ER+ breast cancer cells broadly upregulate ERBB receptor gene expression under endocrine and CDK4/6 inhibition treatment. Fold change of ERBB growth factor receptor mRNA expression (panels), measured by qPCR in three paired cancer cell lines (x axis: CAMA-1, MCF-7 and T47D endocrine therapy-resistant (ETR) and combination ribociclib-resistant (CRR)) under endocrine therapy (fulvestrant 3 nM), CDK4/6 inhibition (ribociclib 0.5 μM) or combination treatments (fulvestrant 3 nM plus ribociclib 0.5 μM) compared to DMSO control. Endocrine and endocrine+CDK4/6i combination treatment, but not CDK4/6i alone, broadly increased ERBB growth factor receptor expression after 7 days of treatment compared to DMSO control (linear mixed model expression change under:: Endocrine:Est=1.83, se=0.19, df=276, $t = 9.77$, $P = $ 2e-16, CDK4/6i:Est=0.64, se= 0.19, df=276, $t = 3.41$, $P = $ 7.4e-4, Combination: Est=2.43, se=0.19, df=276, $t = 12.96$, $P = $ 2e-16; Sample size=288 qPCR samples, 6 cell lines, 3 replicates per gene ($n = 4$) and treatment ($n = 4$)). Box elements represent median (center line), upper/lower quartiles (hinges), 1.5*interquartile range (whiskers=minima/maxima) of ERBB receptor fold change versus control. (**B**) Combination of fulvestrant and afatinib inhibits spheroid growth of 3D cancer monocultures. (**A**) Representative images of spheroid growth of sensitive (Venus, green) or resistant (CFP, blue) cells cultured in fulvestrant (5 nM), afatinib (1.25 μM for CAMA-1, 1.5 μM for MCF-7, and 2.5 μM for T47D), or combination treated media for 18 days. Bars=1000 μm. (**B**) Fluorescence intensity of sensitive and resistant cells under various treatment conditions. Data are represented as average of three replicates ± standard deviation (SD). *$P < 0.05$, **$P < 0.01$, ***$P < 0.001$, ****$P < 0.0001$. T test: differential intensity between treatments:: CAMA-1 ETR: Fulvestrant vs. DMSO $P = 0.0001$, Afatinib vs.DMSO $P = 0.0001$, Fulvestrant+Afatinib vs. Afatinib $P = 0.0001$, Fulvestrant+Afatinib vs. Fulvestrant $P = 0.0006$; MCF-7 ETR: Fulvestrant vs. DMSO $P = 0.0001$, Afatinib vs. DMSO $P = 0.0002$, Fulvestrant+Afatinib vs. Afatinib $P = 0.0036$, Fulvestrant+Afatinib vs.Fulvestrant $P = 0.0029$; T47D ETR: Fulvestrant vs.DMSO $P = 0.0001$, Afatinib vs.DMSO $P = 0.0001$, Fulvestrant+Afatinib vs. Afatinib $P = 0.0019$, Fulvestrant+Afatinib vs. Fulvestrant $P = 0.1246$; CAMA-1 CCR: Fulvestrant vs.DMSO $P = 0.0001$, Afatinib vs.DMSO $P = 0.0001$, Fulvestrant+Afatinib vs. Afatinib $P = 0.346$, Fulvestrant+Afatinib vs. Fulvestrant $P = 0.0001$; MCF-7 CCR: Fulvestrant vs. DMSO $P = 0.0001$, Afatinib vs.DMSO $P = 0.0001$, Fulvestrant+Afatinib vs. Afatinib $P = 0.0265$, Fulvestrant+Afatinib vs. Fulvestrant $P = 0.0001$; T47D CCR: Fulvestrant vs.DMSO $P = 0.0004$, Afatinib vs.DMSO $P = 0.0009$, Fulvestrant+Afatinib vs. Afatinib $P = 0.0013$, Fulvestrant+Afatinib vs. Fulvestrant $P = 0.808$. (**C**) Western blot quantification shows ERBB receptor protein levels increased under fulvestrant treatment and reduced under afatinib across cancer cell lines and experimental replicates. Western blot intensities were measured using ImageJ. Relative intensities of each protein were determined by first normalizing to B-actin and calculating log2 fold changes compared to DMSO control within each cell line. Panel i shows the positive effect of fulvestrant (x axis) and the inhibitory effect of afatinib (color) on the levels of total (top row) and phosphorylated (bottom row) ERBB receptor protein across family members (columns). Points indicate measured relative intensities (quantified western blot bands). Linear mixed-effects regressions (lmer) were fitted for each protein, using data from across treatments, cell lines and experimental replicates to quantify the fulvestrant and afatinib effect sizes, whilst accounting for the cell line and replicate-specific variation in Western intensities, both in the DMSO control and the treatment effects. Boxplots indicate the range of fitted model estimates across replicates within each treatment group (central line=median prediction; lower/upper hinges correspond to first/third quartiles; lower/upper whisker (minima/maxima) extends from the hinge to the largest value less than 1.5 * IQR from the lower/upper hinge). The goodness of fit to the data supports significance estimates of treatment effects. Panel ii shows the lmer (Linear mixed-effects regression) estimated impacts of fulvestrant treatment (yellow) and afatinib treatment (blue) on ERBB protein levels (x axis) and accompanying p values for each ERBB protein (y axis). Across proteins, fulvestrant treatment consistently increased protein levels. Conversely, afatinib consistently reduced them. Sample size = 720 Western blots: 6 cell lines, 8 proteins, 5 treatments, 3 experimental replicates.

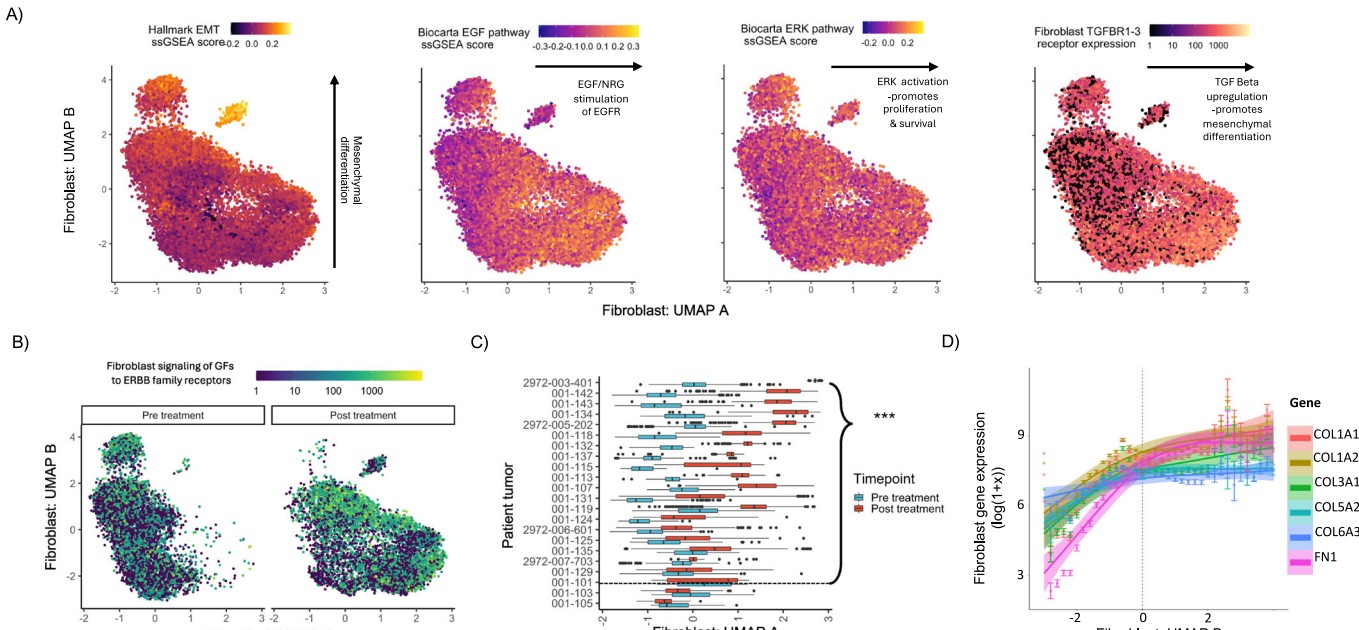

**Figure EV3.  Single-cell phenotypic heterogeneity and ERBB growth factor signaling of fibroblasts of endocrine +/- CDK4/6i treated patient tumors.**

(A) Fibroblast differentiation across the phenotype landscape described by UMAP projection. Points represent single-cell phenotypes. Cells with similar phenotypes are positioned more closely together. The major axes of phenotypic variation correspond to fibroblast EGFR-activated proliferation (UMAP A: *x* axis: high Biocarta EGF and ERK pathway activation) and mesenchymal fibroblast differentiation (UMAP B: *y* axis: high hallmark EMT pathway activation). The biological interpretation of the UMAP dimensions (axis labels) were determined by assessment of the genes, gene sets and communications that dynamically changed (nonlinearly) along each dimension. Biological interpretation of fibroblast phenotypes supported by unsupervised clustering and differential expression analysis (Appendix Figs. S3 and S4). EGFR-activated fibroblasts showed upregulation of TGFβ receptors that drive SMAD signaling and further mesenchymal differentiation indicating that EGFR activation supports mesenchymal fibroblast differentiation (right panel Color= total single-cell expression across TGFβ receptors 1–3). (B) Fibroblast ERBB ligand growth factor signaling activated during treatment in mesenchymal fibroblast cells and a subpopulation of EGFR-activated fibroblasts (color= total single-cell expression of ERBB ligands: NGR1-3, EGF, HBEGF, AREG, TGFA, EMA4D, HLA-A, EFNB1, ICAM1, GNAI2, CDH1, AREG, ANXA1 and ADAM17. (C) Fibroblast EGFR-activated proliferation increased consistently across tumors during endocrine treatment (19/22 letrozole treated patient tumors in the discovery cohort) (Linear mixed effect:: Est=1.23, se=0.19, df=21.00, *t* = 6.37, *P* = 2.6e-6; asterisk signifies significance). Box elements represent median (center line), upper/lower quartiles (hinges), 1.5*interquartile range (whiskers=minima/ maxima) of fibroblast EGFR activation proliferation (measured by fibroblast UMAP A) within a tumor pre or post treatment. Similarly, the fraction of fibroblasts classified (using unsupervised clustering) as EGFR-activated cells increased during treatment. Patients with a greater fraction of EGFR-activated fibroblasts post treatment exhibited greater residual tumor size at day 180 (pathological measurement of longest length; Appendix Fig. S3). (D) Mesenchymal fibroblast differentiation is associated with increased expression of fibronectin and various collagens which drive tumor fibrosis and cancer cell proliferation through extracellular matrix modification. Fibroblasts were finely classified by differentiation state across the phenotypic landscape (mesenchymal fibroblast differentiation: UMAP B) and the mean (points) and standard error (error bars) of collagen and fibronectin gene expression (color) is shown for each level of differentiation. The distribution of fibroblast differentiation states was discretized into 36 equal sized classes to provide an average of 329 cells per level of differentiation. Generalized additive models (smooth curves with uncertainty regions shaded) characterize the trend in gene expression during differentiation. Sample size in (A–D): *n* = 22,916 fibroblast cells of the discovery cohort from 22 patient tumors at two paired timepoints (pre/post treatment).

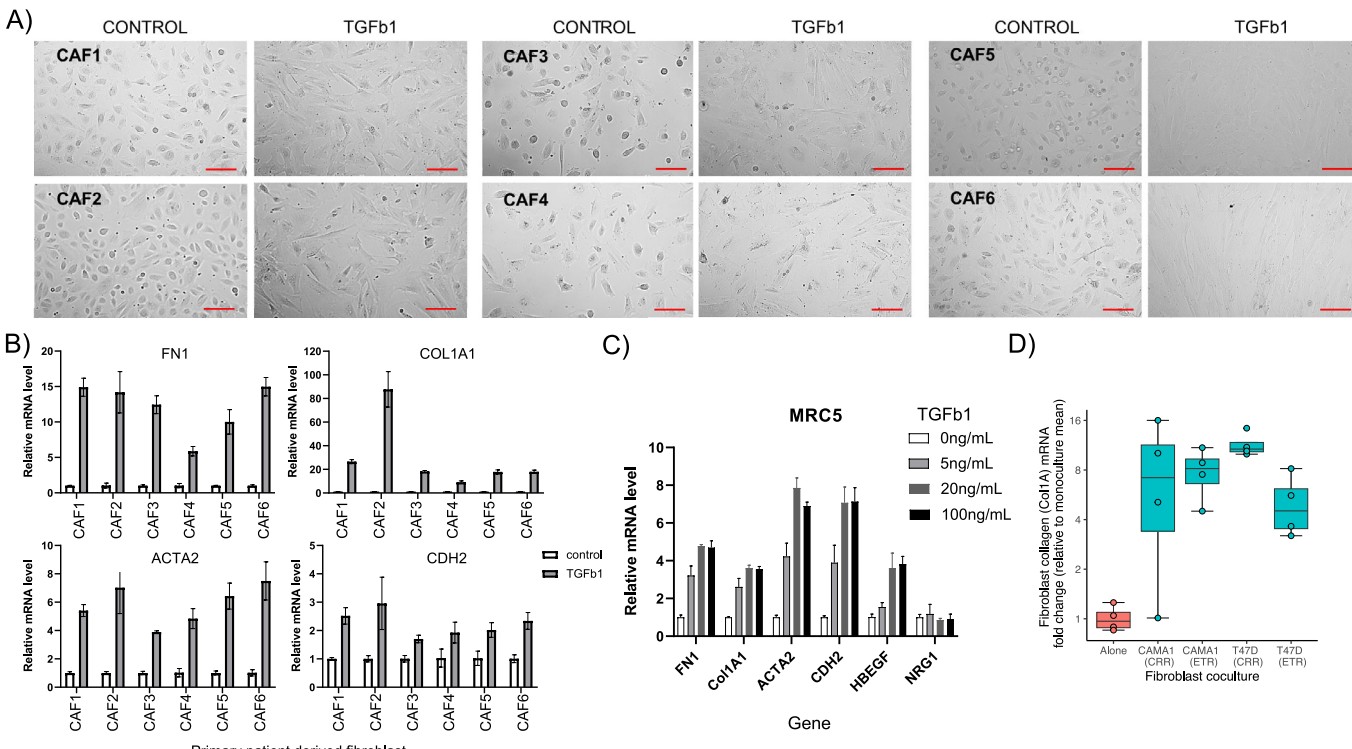

**Figure EV4.  TGFβ1 or cancer coculture activates fibroblasts differentiation and promotes production of growth factors.**

(A) Primary patient-derived fibroblast cells ($n = 6$) were isolated from six patients (CAF1–6) and were treated with TGFb1 (20 ng/mL) for 3 days in 2D cultures with medium containing 10% FBS and compared to untreated controls. Images showing morphological change due to TGFb1 treatment compared to control were captured under microscopy (200 x) prior to cell collection for gene expression quantification by qPCR. Scale bars = 50 μm. (B) Increased mRNA levels of fibrosis makers (FN1, Col1A1, ACTA2) and EMT maker (CDH2) under TGFb1 treatment (gray) versus control (white) in all ($n = 6$) primary patient-derived fibroblast populations (max CAF p-value: FN1 = 0.0015, COLA1 = 0.0006, ACTA2 = 0.0047, CDH2 = 0.033, HBEGF = 0.0243, NRG1 = 0.0067). Error bars = mean +/− standard error. (C) TGFb1 similarly increased mRNA levels of fibrosis makers (FN1, Col1A1, ACTA2), EMT maker (CDH2) and growth factor ligands (HBEGF) in MRC5 fibroblasts treated with TGFb1 (0, 5, 20, 100 ng/mL) in a dose-dependent manner (color) ($n = 3$ replicates per TGFb1 treatment level) (p-value: FN1 = 0.0001, COLA1 = 0.0001, ACTA2 = 0.0001, CDH2 = 0.0001, HBEGF = 0.0004, NRG1 = 0.60). Error bars = mean +/− standard error. (D) Cancer cell coculturing stimulates fibroblast differentiation. Fibroblasts (MRC5) increased mesenchymal fibroblast phenotype (measured by Collagen type I alpha 1 (Col1A)) when cocultured with endocrine therapy-resistant (ETR) or combination ribociclib-resistant (CRR) CAMA-1 or T47D cancer cells (blue) compared to when grown alone (red) for 3 days. (ANOVA: Effect of coculture with: T47D (CRR) est = 1.68, se = 0.43, t = 3.9, P = 0.0014; T47D (ETR) est = 2.02, se = 0.43, t = 4.7, P = 0.0003; CAMA-1 (CRR) est = 2.43, se = 0.43, t = 5.7, P = 4.6e-5; CAMA-1 (ETR) est = 1.60, se = 0.43, t = 3.65, P = 0.002). Box elements represent median (center line), upper/lower quartiles (hinges), 1.5*interquartile range (whiskers = minima/maxima) for collagen mRNA levels in replicate fibroblast populations (points = replicates) relative to the monoculture mean. Sample size = 20 populations (5 compositions with 4 replicates).

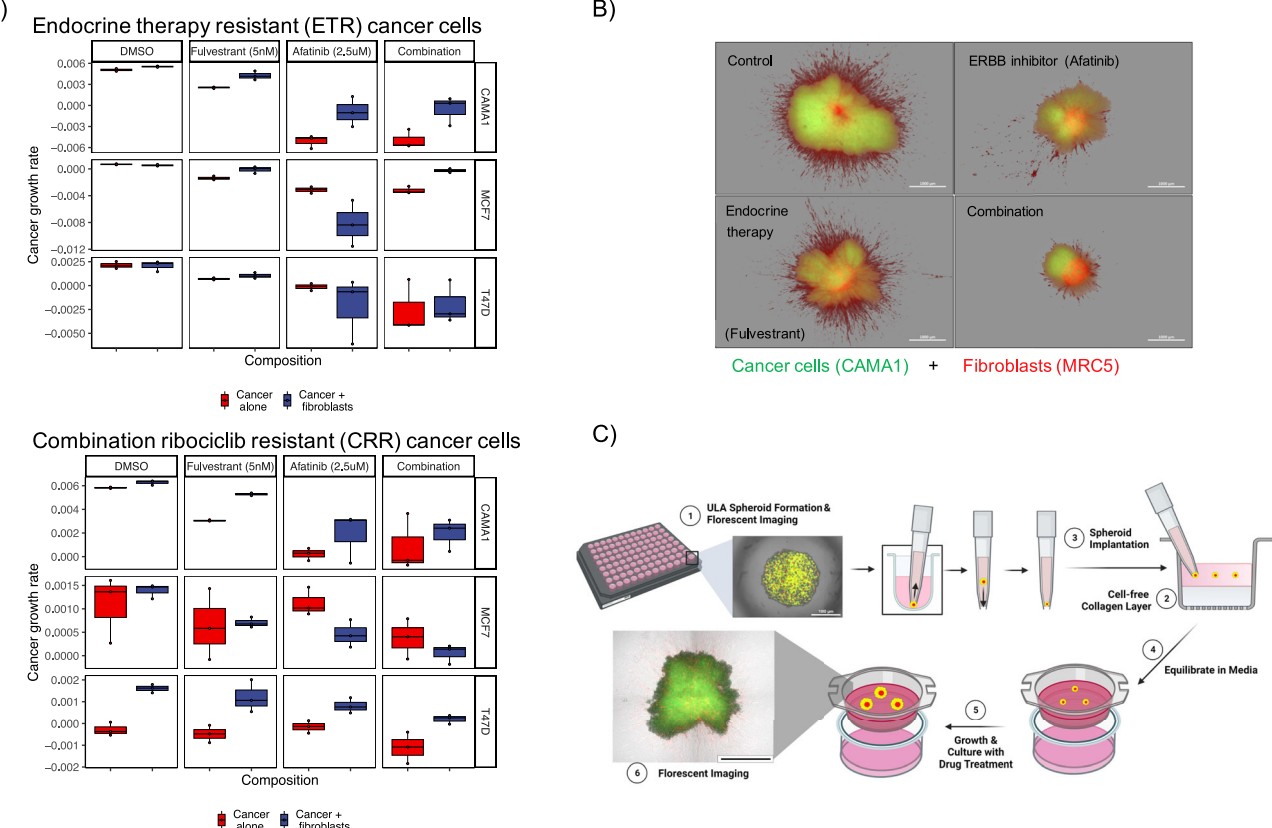

**Figure EV5. ERBB pathway inhibition blocks fibroblast facilitation of cancer, control spheroid outgrowth, cancer proliferation and the invasive spiked morphology.**

(A) ERBB pathway inhibition blocks fibroblast facilitation of cancer promoted by endocrine therapy in both endocrine therapy-resistant (ETR: top) and combination ribociclib-resistant (CRR: bottom) cancer cells. Fibroblast facilitation of cancer population growth blocked by Afatinib (ERBBi). Cancer cells growth rate analysis of CAMA-1, MCF-7 and T47D cancer cells (rows) when monocultured (red) or cocultured with fibroblasts (MRC5) (blue) in 3D spheroids under treatments of afatinib (2.5 μM), fulvestrant (5 nM) or combination (afatinib 2.5 μM + fulvestrant 5 nM) at the indicated concentrations. For all figures, endocrine therapy-resistant cancer cells were labeled with Venus (green), and combination ribociclib-resistant cells were labeled with Cerulean (blue) for CAMA-1, MCF-7 and T47D. MRC5 fibroblasts were seeded together with cancer cells at 1:1 ratio in medium containing 1% FBS. Spheroid Images were captured every 3 or 4 days before medium and drug changes for up to 10 days. Fluorescence intensity was calculated for each image to determine cell abundance and exponential rate of change of cell number measured using least squares. Box elements represent median (center line), upper/lower quartiles (hinges), 1.5*interquartile range (whiskers=minima/maxima) of cancer growth rate (cells/cell/day) across replicate spheroids of a cell line within each treatment. Facilitation of ETR (top panel) and CRR (bottom panel) cancer cell lines (rows; CAMA-1, MCF-7 and T47D) evident by their accelerated growth when cocultured with fibroblasts compared to cancer monocultures under DMSO treatment (first column) and more strongly under fulvestrant treatment (second column). Blockage of facilitation evident under afatinib (third column) and combination treatment (fourth column) by the equivalent or slower growth rate of fibroblast cocultured cancer cells compared to cancer monocultures. Afatinib also slows and often completely prevents the growth of resistant and sensitive cancer cells, especially in combination with fulvestrant. Sample size: $n = 72$ spheroids (3 replicates, 6 cell lines, 2 fibroblast abundances (present/absent), 2 treatments) measured across 5 timepoints. (B) Combining endocrine therapy with an ERBB inhibitor controls spheroid outgrowth. Representative images of cancer-fibroblasts spheroid cocultures (SPIKE assay) composed of endocrine therapy-resistant CAMA-1 cancer cells (green: CAMA-1 (ETR)) and fibroblasts (red; MRC5), two weeks after being embedded in matrix and treated with DMSO (control; top left), pan-ERBB inhibitor (afatinib 10 μM; top right), endocrine therapy (fulvestrant 10 nM; bottom left) or combination treatment (bottom right). When cocultured with fibroblasts, control-treated cancer spheroids have a distinct invasive morphology, with cancer cells growing along collagen spikes. Endocrine treated cancer spheroids maintain the spikey morphology, but cancer growth is partially decreased. ERBB inhibition blocks cancer proliferation and the invasive spiked morphology. Combination endocrine and ERBB inhibitor therapy blocks growth and prevents invasive cancer morphology. (C) SPIKE assay overview. Cancer-fibroblast coculture spheroids are established for 48 h (1), a transwell culture insert is prepared with an acellular collagen layer (2) followed by spheroid transplanting into an upper collagen layer (3). Spheroids are equilibrated in culture media for 18 h (4) prior to culture media/drug treatment added via the exterior media reservoir (5). Cocultures of cancer cells (green/yellow) and fibroblasts (red) can be monitored by florescent imaging in ULA plates ((1), BioTek Cytation 5, scale = 1,000 μm) or by widefield microscopy (6) (Axio Observer 7, scale = 1000 μm). (Created with Biorender.com).

