## [Peer Review File · Molecular Systems Biology]

Blocking cancer-fibroblast mutualism inhibits proliferation of endocrine resistant breast cancer.

Jason Griffiths, Feng Chi, Elena Farmaki, Eric Medina, Patrick Cosgrove, Kimya Karimi, Jinfeng Chen, Vince Grolmusz, Frederick Adler, Qamar Khan, Aritro Nath, Jeffrey Chang, and Andrea Bild

Corresponding author(s): Jason Griffiths (jgriffiths@coh.org) , Andrea Bild (abild@coh.org)

Review Timeline:

Submission Date:	10th Jul 24
Editorial Decision:	12th Aug 24
Revision Received:	2nd Dec 24
Editorial Decision:	21st Jan 25
Revision Received:	18th Feb 25
Editorial Decision:	21st Mar 25
Revision Received:	24th Mar 25
Accepted:	10th Apr 25

Editor: Poonam Bheda

Transaction Report:

after three months if you have not completed it, to update us on the status.

I look forward to receiving your revised manuscript.

Yours sincerely,

Poonam Bheda

Poonam Bheda, PhD
Scientific Editor
Molecular Systems Biology

Reviewer #1:

ER+ and PR+ breast cancers develop resistance to targeted ER/CDK inhibitors through activation of alternative proliferation pathways such as ERBB. Here the authors performed scRNAseq on serial (time series) biopsies of patients to uncover mechanisms of how resistance emerges. The primary finding is that cancer cells up regulate ERBB receptor signaling and promote fibroblasts to produce the corresponding ligand.

Endocrine therapy resistance develops in majority of ER+ breast cancer metastatic cases. Adding CDK4/6 inhibitors to endocrine therapy can improve met outcomes but less effective in early stage disease. A previous smaller study by the group showed rewiring of cancer cell signaling and a shift to alternative proliferative pathways, of which there are a variety of options.

Cancer cells can themselves produce growth factors to promote self--proliferation but such ligands can also come from the TME e.g. fibroblasts and endothelial cells. Feedback between cancer and the TME can accentuate this loop, and may be encouraged by more stressful treatment conditions, which might lead to faster development of resistance.

The present study hypothesizes that endocrine therapy can promote such a feedback loop, and investigates this through scRNAseq on serial patient biopsies during treatment. This facilitates understanding of phenotypes and crosstalk, and uses complementary assays to assess mechanisms of interactions.

Dataset of over 400k cells from 173 biopsies from 62 patients prior, during, and after therapy.

Multiple patients converged on the same crosstalk pathway with fibroblasts. Endocrine therapy unregulated ERBB on cancer cells as well as production of ligands by fibroblasts. Blocking this reduced cancer cell growth.

Post-menopausal node+ or >2cm tumor patients were selected from the FELINE trial. 120 patients were randomized to endocrine alone or endocrine with CDK4/6. Half were used for discovery, and half for validation experiments. Samples acquired at day 0, 14, and ~180 and subject to 10X genomics. 35 were from discovery cohort, and 27 from validation cohort. These were subject to standard processing techniques for clustering, labeling cells.

Analysis was performed to compare pre-treatment cancer cells to those remaining after treatment in the same patients. Additionally cell-cell communications were inferred both to cancer cells and from cancer cells to the TME. Computational inferences were tested in 6 organoid models - these were developed from initially responsive cell lines, and also ones which no longer responded following extended treatment to induce resistance mechanisms.

Majority of post-treatment resistant cancer cells activated ERBB signaling (via gene set analysis) including up regulation of downstream effector transcription factors such as cJun/FOS. Moreover they had lost expression of endocrine dependent proliferation genes. Concurrently, Western blots confirmed phosphorylated activation of ERBB signaling, consistent with RNA level changes.

Ligand-receptor computational analysis also showed enhanced signaling from the TME to multiple components of the ERBB apparatus on cancer cells. The largest contribution came from fibroblasts. Moreover, this was specifically coming primarily from a "myCAF" subtype of mesenchymal fibroblasts which emerged through differentiation during therapy. This could be induced by TGFb/EGF signaling and it turns out that the main source of these signals was the cancer cells, supporting development of a

positive feedback loop.

In vitro experiments supported the findings of the analyses of primary tumor samples via scRNAseq etc.

Finally, the authors developed a differential equation based model of the interaction which fit the observed dynamics. They additionally tested whether inhibiting ERBB signaling could reduce cancer cell growth and fibroblast differentiation in spheroids. This confirmed that myCAF production was largely stopped, and spheroid growth was inhibited; thus supporting the primary hypothesis.

There is little to criticize in this paper. It's carefully done, addresses an interesting and important topic, and the findings are important from both understanding cancer-TME signaling in general and in the specific context of development of endocrine therapy resistance in these specific breast cancers. The methodologies are appropriate, and the experimental validations in organic models (as well as in vitro Western blots etc) provide support to the genomic findings, going well beyond what is normally done. The power comes from the patient cohort and the serial biopsies, together with the large size of the dataset generated.

Reviewer #2:

Summary of story

Drug combinations of antiestrogens and CDK4/6i have variable efficacy in early stage BrCa. The authors have previously shown that therapy resistance involves a switch from ER-driven to growth factor-driven signaling. In this manuscript the authors present evidence to suggest that this growth factor-driven signaling is a result of mutualism between tumor cells and a myoepithelial cancer-associated fibroblast (myCAF) phenotype and that this mutualism is sensitive to inhibition of the ErbB family of receptor tyrosine kinases with afatinib. The authors leverage a large sample cohort (the FELINE clinical trial from which 173 tumor biopsies from 62 patients were taken prior to, during and after treatment with neoadjuvant antiestrogen therapy (letrozole) with or without a CDK/6 inhibitor (ribociclib)). They obtained single-cell RNA-seq measurements and used complex statistical approaches to identify the bidirectional cell-cell communication between cancer-associated fibroblasts and cancer cells that are upregulated in the 6 month-treated samples. They also used cell line models of therapy resistant BrCa and ex vivo cultured cancer-associated fibroblasts (CAF) to enable experimental validation of the effects of this cell-cell communication on tumor growth and response to treatment.

The authors' hypothesis that mutualism is driving the ability of BrCa tumors to resist ER and CDK4/6 inhibitors is intriguing, albeit not especially novel, as cancer-associated fibroblasts have long been known to facilitate tumor growth and resistance to therapy. However, this manuscript provides detailed evidence for the involvement and potential co-evolution/adaptation of cancer cells and cancer-associated fibroblasts facilitating therapy resistance through bidirectional cell-cell communication. As such, it should be of reasonably high interest to the broader research community and has direct clinical relevance. The ex vivo experiments provide reasonable support for the observations from the clinical samples and are further bolstered by their mathematical description of the hypothesized mutualism. The authors' description of mutualism within the context of cancer treatment resistance is of broad interest in the field but the manuscript could be strengthened by more in-depth quantitative assessment of mutualism within their experimental system, some in silico predictions using their mathematical model, and further validation using the clinical samples.

Major critiques

Overall text. This paper was somewhat difficult to read; I frequently found myself getting caught up in unnecessary technical details in the main text that made it hard to follow the underlying story. The use of highly technical (and nonstandard) terms for specific methodological approaches is distracting and unnecessary in the results section making it more challenging to determine what aspects of the work is important/relevant to the story-especially Figure 1, which is hard to follow. An example of the addition of technical details that confuse the message: "Assessment of fibroblast-to-cancer cell crosstalk using linear models showed consistent activation of ERBB communication pathways across patient tumors via ERBB receptors (Fig 3B)." This wording elicits questions as to exactly how the quantification of ErbB pathway activity was measured using linear models and appears to focus more on the approach than on the result. Suggested text, simplified for clarity: Fibroblast-to-cancer cell ErbB pathway activation was consistently identified across patient tumors.

Summary description of experimental approach. The text in Section 1 describing the prior published work by the same authors (Nature Cancer 2021) is important and should be expanded to clarify exactly how the current work extends it, but it would seem appropriate to put at the end of the Introduction. Important questions that should be addressed include, 1) did any of the patient demographics or summary statistics of treatment response change after the addition of the validation cohort? 2) Were any changes in sample or data acquisition introduced? 3) Since the current work focused only on samples with sufficient single-cell data obtained after 180 days of treatment, was any sample bias introduced when compared to the entire cohort? E.g., were more treatment-sensitive tumors preferentially excluded? Other text within Section 1 may be relevant to include in the Results but it could be summarized with fewer of technical details that seem more appropriate for the Methods. For example, several nonstandard terms of methodological approaches (if kept) must be more clearly defined, explained, reworded or removed,

endocrine drug effects are described as X rather than μ .

Reviewer #3:

The authors performed an important study of early-stage ER+ BC patients treated with a CDK4/6 inhibitor in the neoadjuvant setting. Longitudinal studies where tumor samples are collected and profiled prior to treatment as well as during treatment are especially valuable, and the authors further distinguish their study by performing single-cell RNA sequencing, which provides an unprecedented level of insight into the molecular characteristics of the tumor samples.

This data set enabled the team to distinguish between changes to tumor cells and the tumor microenvironment, a novel and important analysis. In addition, the team also performed in vitro studies using cancer cell lines to further study the hypotheses generated from the clinical data.

A major finding that the authors highlight is that the ERBB pathway was activated in patients upon treatment, and that the mechanism of ERBB pathway activation was not solely through upregulation of the ERBB receptor. The authors analyze the single-cell RNA sequencing data to separately characterize the tumor cells and immune and other cells in the tumor microenvironment, and propose that fibroblasts are inducing ERBB signaling in the tumor through tumor-fibroblast crosstalk.

Despite these strengths, the manuscript suffers from significant deficiencies that much be addressed prior to publication.

1. Throughout the text (including the title, the abstract, the body, figures), the authors refer to "endocrine therapy resistant breast cancer" without any evidence that the samples they are studying are endocrine therapy resistant.

o The cohort that the authors have examined is neoadjuvant and presumably has not received prior endocrine therapy. The cohort is then treated with endocrine +/- CDK4 inhibitors, and samples are collected post treatment. The authors do not classify the response of the tumors or seek to correlate mechanistic biomarkers with markers of sensitivity or resistance; instead they compare samples collected during treatment with pre-treatment samples. Hence it is not clear to this reviewer why the authors claim that this cohort is ET resistant nor what evidence they have to associate the molecular pathways with sensitivity to endocrine therapy. For example in the abstract the authors state "In most patient tumors, resistant cancer cells increased ERBB growth pathway activity during treatment, only partially through ERBB receptor upregulation " Unless the authors correlate response to therapy with molecular features, what evidence do they have that the molecular features are associated with resistance? Do the authors intend to communicate that they are examining the molecular changes in response to endocrine therapy in early-stage ER+ BC, which may or may not be associated with sensitivity - if so they should clarify throughout the manuscript and change the title. If the authors agree, this conceptual change in the manuscript has important implications for the interpretation of the results that the authors should state and that will require rewriting.

2. The authors do not appear to separate the samples collected prior to surgery at day 14 from those collected after surgery at day 180, and group both together as post treatment. One might imagine that there are significant differences between samples collected at day 14 and day 180 - have the authors investigated these differences and why were they grouped together?

3. The samples were from patients treated with ribociclib plus letrozole as well as letrozole monotherapy. The authors did not report any comparisons between the two treatment arms to assess any differences between them. In principle, it would be interesting to perform these comparison and identify any potential differences. It might also identify mechanisms that are unique to the combination arm.

4. The authors note that "Activation of the ERBB signaling pathway was also observed in normal diploid epithelial cells of these post treatment tumor samples, indicating that it may be induced by treatment and TME communication changes rather than selected for during subclonal cancer evolution". How do the authors align this observation with their conjecture that ERBB upregulation is a mechanism of tumor escape to treatment? Would the null hypothesis simply be that ERBB upregulation is a response to endocrine treatment and not selected as a mechanism of resistance?

5. The relatively fast upregulation of ERBB2 signaling in vitro at 1 week suggests that this made simply be a marker of treatment - how can the authors demonstrate that it leads to treatment resistance?

6. Figure 1 is a conceptual overview of the study with a long caption that recapitulates some of the body of the paper. The authors do not provide detail on the number of samples collected at each time point in the figure - making it difficult to interpret the subsequent work. In the text, the authors distinguish samples collected prior to, during and after treatment "...serially collected from 173 tumor biopsies from 62 patients taken prior to, during and after treatment." The authors do not state the during of treatment and should have a overview of the cohort that includes this information - a missed opportunity is Figure 2 where no distinction is made between the 14 day timepoint and the 180 day timepoint and so it is unclear if "post-treatment" refers to one or both of those timepoints.

7. The authors established cell models of endocrine and endocrine + CDK4/6 inhibitor resistance. By construction, the cell

models are resistant to treatment and not simply the result of exposure to endocrine and/or endocrine + CDK4/6 inhibitors, where as noted above it appears to this reviewer that there is no evidence that the analysis the authors performed on the samples collected from patients provides into mechanisms of resistance rather than simply response to treatment. The connection between the in vitro models and the human cohort is not obvious and should be clarified, and statements such as "Patient derived insights, validated across independent cohorts, and multiple in vitro model systems of endocrine and ribociclib resistance support the ecological prediction that the oncogenic mutualisms of cancer and fibroblasts intensify under treatment." must be justified or eliminated.

8. The authors note that the resistant cell lines upregulate ERBB pathway activity. In the patients, the authors propose a mechanism whereby the tumor secretes TGF β which in turn stimulates fibroblasts to release EGF ligands that lead to ERBB pathway activation. If this is the mechanism of ERBB activation, what is the relevance of the in vitro cell model system that does not include fibroblasts, and likewise what is the relevance of the fibroblast mutualism if ERBB activation occurs in vitro in the absence of fibroblasts? The fact that the in vitro cell model upregulates ERBB through a purely tumor intrinsic mechanism suggests that the mutualism is not the exclusive mechanism of ERBB upregulation and begs the question of how relevant the effect of the fibroblasts is. The studies of tumor-fibroblast coculture reported in Fig 6 may be able to demonstrate the importance of mutualism: the authors should clarify how this connects with the original resistance models generated in the absence of fibroblasts.

Response to reviewers' comments for: " Blocking cancer-fibroblast mutualism inhibits proliferation of endocrine therapy resistant breast cancer"

Dear reviewers,

We thank all reviewers and the editor for providing extremely encouraging, positive and constructive appraisals of the manuscript. In this document, we address reviewers' comments (*italicized*) sequentially, with a point-by-point description (*in blue colored text*) of how each comment or question about the manuscript has been addressed. Changes to the manuscript have been highlighted.

REVIEWER COMMENTS

Reviewer #1:

Study summary) *"ER+ and PR+ breast cancers develop resistance to targeted ER/CDK inhibitors through activation of alternative proliferation pathways such as ERBB. Here the authors performed scRNAseq on serial (time series) biopsies of patients to uncover mechanisms of how resistance emerges. The primary finding is that cancer cells up regulate ERBB receptor signaling and promote fibroblasts to produce the corresponding ligand. Endocrine therapy resistance develops in majority of ER+ breast cancer metastatic cases. Adding CDK4/6 inhibitors to endocrine therapy can improve met outcomes but less effective in early-stage disease. A previous smaller study by the group showed rewiring of cancer cell signaling and a shift to alternative proliferative pathways, of which there are a variety of options. Cancer cells can themselves produce growth factors to promote self-proliferation but such ligands can also come from the TME e.g. fibroblasts and endothelial cells. Feedback between cancer and the TME can accentuate this loop, and may be encouraged by more stressful treatment conditions, which might lead to faster development of resistance.*

The present study hypothesizes that endocrine therapy can promote such a feedback loop and investigates this through scRNAseq on serial patient biopsies during treatment. This facilitates understanding of phenotypes and crosstalk and uses complementary assays to assess mechanisms of interactions. Dataset of over 400k cells from 173 biopsies from 62 patients prior, during, and after therapy.

Multiple patients converged on the same crosstalk pathway with fibroblasts. Endocrine therapy unregulated ERBB on cancer cells as well as production of ligands by fibroblasts. Blocking this reduced cancer cell growth.

Post-menopausal node+ or >2cm tumor patients were selected from the FELINE trial.

120 patients were randomized to endocrine alone or endocrine with CDK4/6. Half were used for discovery, and half for validation experiments. Samples acquired at day 0, 14, and ~180 and subject to 10X genomics. 35 were from discovery cohort, and 27 from validation cohort. These were subject to standard processing techniques for clustering, labeling cells.

Analysis was performed to compare pre-treatment cancer cells to those remaining after treatment in the same patients. Additionally, cell-cell communications were inferred both to cancer cells and from cancer cells to the TME. Computational inferences were tested in 6 organoid models - these were developed from initially responsive cell lines, and also ones which no longer responded following extended treatment to induce resistance mechanisms.

Majority of post-treatment resistant cancer cells activated ERBB signaling (via gene set analysis) including up regulation of downstream effector transcription factors such as cJun/FOS. Moreover, they had lost expression of endocrine dependent proliferation genes. Concurrently, Western blots confirmed phosphorylated activation of ERBB signaling, consistent with RNA level changes.

Ligand-receptor computational analysis also showed enhanced signaling from the TME to multiple components of the ERBB apparatus on cancer cells. The largest contribution came from fibroblasts. Moreover, this was specifically coming primarily from a "myCAF" subtype of mesenchymal fibroblasts which emerged through differentiation during therapy. This could be induced by TGFb/EGF signaling and it turns out that the main source of these signals was the cancer cells, supporting development of a positive feedback loop.

In vitro experiments supported the findings of the analyses of primary tumor samples via scRNAseq etc.

Finally, the authors developed a differential equation-based model of the interaction which fit the observed dynamics. They additionally tested whether inhibiting ERBB signaling could reduce cancer cell growth and fibroblast differentiation in spheroids. This confirmed that myCAF production was largely stopped, and spheroid growth was inhibited; thus supporting the primary hypothesis."

Response) Thank you for your thorough summary. We are pleased that the key take home messages of the study were conveyed in the manuscript and that the details of the study were also communicated to the expert reviewer.

Overall Remarks) "There is little to criticize in this paper. It's carefully done, addresses an interesting and important topic, and the findings are important from both

understanding cancer-TME signaling in general and in the specific context of development of endocrine therapy resistance in these specific breast cancers. The methodologies are appropriate, and the experimental validations in organic models (as well as in vitro Western blots etc) provide support to the genomic findings, going well beyond what is normally done. The power comes from the patient cohort and the serial biopsies, together with the large size of the dataset generated."

Response) Our team greatly appreciates these positive comments, and thanks the reviewer for their time, effort and advice. We agree that the power of the study comes from the serial biopsies that were kindly provided by many patients during the FELINE trial. We also concur with the reviewers point that the combination of mathematical and computational modeling and detailed experimental validation supports the molecular findings.

Reviewer #2:

Study summary) "Drug combinations of antiestrogens and CDK4/6i have variable efficacy in early stage BrCa. The authors have previously shown that therapy resistance involves a switch from ER-driven to growth factor-driven signaling. In this manuscript the authors present evidence to suggest that this growth factor-driven signaling is a result of mutualism between tumor cells and a myoepithelial cancer-associated fibroblast (myCAF) phenotype and that this mutualism is sensitive to inhibition of the ErbB family of receptor tyrosine kinases with afatinib. The authors leverage a large sample cohort (the FELINE clinical trial from which 173 tumor biopsies from 62 patients were taken prior to, during and after treatment with neoadjuvant antiestrogen therapy (letrozole) with or without a CDK/6 inhibitor (ribociclib)). They obtained single-cell RNA-seq measurements and used complex statistical approaches to identify the bidirectional cell-cell communication between cancer-associated fibroblasts and cancer cells that are upregulated in the 6 month-treated samples. They also used cell line models of therapy resistant BrCa and ex vivo cultured cancer-associated fibroblasts (CAF) to enable experimental validation of the effects of this cell-cell communication on tumor growth and response to treatment. The authors' hypothesis that mutualism is driving the ability of BrCa tumors to resist ER and CDK4/6 inhibitors is intriguing, albeit not especially novel, as cancer-associated fibroblasts have long been known to facilitate tumor growth and resistance to therapy. However, this manuscript provides detailed evidence for the involvement and potential co-evolution/adaptation of cancer cells and cancer-associated fibroblasts facilitating therapy resistance through bidirectional cell-cell communication. As such, it should be of reasonably high interest to the broader research community and has direct clinical relevance. The ex vivo experiments provide reasonable support for the observations from the clinical samples and are further bolstered by their

mathematical description of the hypothesized mutualism. The authors' description of mutualism within the context of cancer treatment resistance is of broad interest in the field but the manuscript could be strengthened by more in-depth quantitative assessment of mutualism within their experimental system, some in silico predictions using their mathematical model, and further validation using the clinical samples.”

Response) We are very pleased that this and other reviewers have found our work to be of high interest to the research community. We are grateful that the reviewer appreciated the extensive, high quality and valuable clinically relevant data that we generated. It is encouraged that they value the detailed evidence that this data provides when combined with appropriate statistical models and in vitro experimental validation.

Comment/Question) – Overall text:

“This paper was somewhat difficult to read; I frequently found myself getting caught up in unnecessary technical details in the main text that made it hard to follow the underlying story. The use of highly technical (and nonstandard) terms for specific methodological approaches is distracting and unnecessary in the results section making it more challenging to determine what aspects of the work is important/relevant to the story-especially Figure 1, which is hard to follow. An example of the addition of technical details that confuse the message: ‘Assessment of fibroblast-to-cancer cell crosstalk using linear models showed consistent activation of ERBB communication pathways across patient tumors via ERBB receptors (Fig 3B).’ This wording elicits questions as to exactly how the quantification of ErbB pathway activity was measured using linear models and appears to focus more on the approach than on the result. Suggested text, simplified for clarity: Fibroblast-to-cancer cell ErbB pathway activation was consistently identified across patient tumors.”

Response) We have revised the manuscript to move overly technical details from the results into the methods section. We thank the reviewer for their additional effort to provide suggestions to clarify the text. This specific section now reads: “Across patient tumors, fibroblast-to-cancer cell ERBB communication was consistently increased via various ERBB receptors (**Fig 3B**).”

We have followed the reviewers advice of focusing the results on the biology and provided references to the appropriate methods section when technical details were previously given.

Comment/Question) – Summary description of experimental approach:

“The text in Section 1 describing the prior published work by the same authors (Nature Cancer 2021) is important and should be expanded to clarify exactly how the current

work extends it, but it would seem appropriate to put at the end of the Introduction. Important questions that should be addressed include,

- 1) did any of the patient demographics or summary statistics of treatment response change after the addition of the validation cohort?*
- 2) Were any changes in sample or data acquisition introduced?*
- 3) Since the current work focused only on samples with sufficient single-cell data obtained after 180 days of treatment, was any sample bias introduced when compared to the entire cohort? E.g., were more treatment-sensitive tumors preferentially excluded?"*

Response) We agree that this is an important piece of information to outline more clearly in the introduction. We have incorporated this addition into the second paragraph of the paper. The subsequent sentences then explain the limitation of the previous work, namely that we still needed to identify the “alternative signals driving reactivation of proliferation represent direct therapeutic targets to overcome resistance”. This change helps motivate the work in this paper and we are pleased that you pointed this out.

To address these specific three questions:

- 1) Patient demographics and tumor treatment response metrics of the discovery and validation cohort were determined during the Griffiths et al. (2021) study and remained unchanged.*
- 2) A consistent sample preparation and data acquisition pipeline was used*
- 3) As patients in the FELINE trial received neoadjuvant treatment, even those that were responsive (had shrinking tumors) still had pathologically detectable amounts of residual disease at the end of treatment. Due to this, we were able to obtain samples from these responding and non-responding tumors, preventing a strong sampling bias linked to response. Across the study, 48% of patients were classified as responders with tumors exhibiting shrinkage. Very similarly, in the serially sampled molecular dataset, 47% of tumors are treatment sensitive. No evidence of preferential exclusion seen.*

We have included these points in the methods, stating:

- 1) “The 120 patients were previously divided into two equally sized cohorts: a hypothesis generating discovery cohort and a validation cohort (Griffiths et al, 2021). Patient demographics metrics and tumor treatment response outcomes were determined at the onset of that study and remained unchanged.”*
- 2) “Sample preparation and data acquisition were consistent between cohorts.”*
- 3) “No sample bias was detected when we compared to the fraction of responding/non-responding tumors between the patients with paired samples and the full cohort.”*

Comment/Question) – “Other text within Section 1 may be relevant to include in the Results but it could be summarized with fewer of technical details that seem more appropriate for the Methods. For example, several nonstandard terms of methodological approaches (if kept) must be more clearly defined, explained, reworded or removed, including: 1) phenotype evolution analysis, 2) hierarchical random regression, 3) extended expression product communication. These phrases are currently meaningless without more context. A graphical summary of the methodological approach may be better suited to the authors' objectives of the text in Section 1 and Figure 1.”

Response) We appreciate the need to clearly convey our findings without extensive technical detail. All the terms that you raised have been taken out of Section 1. We have moved these details into the methods (where they are explained in detail). In Section 1, we provided pointers to the relevant methods section so that interested readers can navigate there more easily.

Comment/Question) – *ErbB signaling pathway activity score.*

“It is unclear how the results shown in Fig S3 (gene sets identified as recurrently dysregulated in cancer cells of post treatment tumor samples) led to the authors' results shown in Fig 2A (activation of the ERBB growth-factor signaling pathway) since only one of the 48 ERBB activation gene sets shown in Fig S4 was identified in Fig S3. There was no description of the progression. Moreover, in the legend to Figure 2 the authors state, "Pathway scores are a composite of 48 ERBB activation ssGSEA signatures from the C2 collection (listed in Fig S4)," but there is no description of how the composite score was calculated in Methods. If this calculation is derived from the statistical modeling of the treatment-induced effects described in the Methods section titled "Identifying consistent resistance phenotypes across tumors," it should be made clear and explicit. However, it may be best to add another section within Methods specifically for this calculation.”

Response) We thank the reviewer again for highlighting a place where the description of the studies progression could be improved. We have addressed this advice in the results by providing a brief description of how we first identified that dysregulated pathways were linked to ERBB signaling, then identified that pathway activity was correlated within cells and due to this created on composite ERBB score that measures ERBB activity across these pathways. To avoid technical details (as proposed above), we follow the reviewers suggestion and provide a description of this approach in a new methods section.

In the methods we state that: “*Measuring composite phenotypes scores:* For pathways with correlated activity within cells, UMAP was used to perform non-linear dimension

reduction. Higher dimensional pathway data was projected into a low-dimensional latent space, preserving the global structure and providing a summary of the dominant patterns of variation across single cells. The first UMAP dimension was used to measure the composite phenotype score as it quantifies the major axis of phenotypic variation.”

Comment/Question) – *Within-cell-type phenotypic heterogeneity- myCAF phenotype: “The section in the Methods describing the characterization of within-cell-type phenotypic heterogeneity requires more explanation, especially as it pertains to what the authors refer to as myCAF cells. The authors write, “For each broad cell type, we generated a cell-type specific UMAP based on ssGSEA profiles with the intrinsic UMAP dimensionality determined using the packing number estimator. We then subdivide each cell type into subtypes of at least 30 cells with coherent phenotypes and of equal interval width along each phenotype axis.” This is not sufficiently clear. What is meant by a ssGSEA profile? Are you only considering genes from specific MSigDB gene sets identified as enriched by ssGSEA? If so, which specific gene sets were used to discriminate the fibroblast cells and how were they identified?”*

Response) We have clarified the highlighted methods section. Specifically, we have removed the term ssGSEA profile as it is not sufficiently clear or descriptive. Instead, we have described the list of ssGSEA pathway scores that are provided as the model input (c2+hallmark normalized ssGSEA pathway scores).

We state that: “For each broad cell type, we generated a cell-type specific UMAP based on ssGSEA pathway scores (4775 pathways per cell; c2+hallmark). This contributed to dimension reduction and the zinbwave’ normalization process also corrected for biases associated with single cell read counts and gene length and GC-content.”

Comment/Question) – *“Figure S9A and B show the expression scores of specific MSigDB gene sets of the individual cells comprising the fibroblast cell population visualized in UMAP-embedded space with the UMAP dimensions labeled as Fibroblast EGFR activated proliferation and myCAF differentiation, but it is unclear where these axes labels came from. Is this based on a ssGSEA score for a specific MSigDB gene set involving myCAF differentiation? Since UMAP embedding is an unsupervised approach, how are the specific UMAP dimensions being attributed to the specific axis labels? Which cells are specifically considered myCAFs and why? What proportion of the fibroblast population would be considered myCAF? Are they specifically enriched in the cells present after 180 days of treatment versus other fibroblast subpopulations?”*

Response) We have clarified the description of how the biological interpretation of the UMAP dimensions were obtained. In the legend of figure S9 we explain that: “the biological interpretation of UMAP dimensions (axis labels) were determined by assessment of the genes, gene sets and communications that correlated with each dimension.”

As shown in Figure S9, we see a relatively continuous distribution of mesenchymal fibroblast differentiation. By using the communication analysis to integrate the signaling strength across fibroblast subpopulations, we have circumvented the need for what would be a fairly arbitrary cutoff to define specific sub populations of cells within the manuscript. Nevertheless, there is evidence in the single cell data to propose a tentative classification cut off around the zero point on the mesenchymal differentiation UMAP axis. Figure S10 shows that the expression of mesenchymal marker genes increases asymptotically across the differentiation gradient. A plateau of the increase in these markers can be seen above a mesenchymal differentiation score of around zero. We have indicated in the methods that this cutoff is what we used to define high and low mesenchymal differentiation states. Here, we do not specifically wish to claim that this is an optimal classification cut point. We are aware that there is a large body of research characterizing fibroblast cell subpopulations and much more thorough investigation is warranted. We would be excited to take this on as a follow-up project with interested collaborators.

Comment/Question) – *“In the current manuscript, the use of the term myCAF is not well supported (or even necessary). If the authors want to focus specifically on this subtype of fibroblasts, more evidence needs to be provided to demonstrate the relevance and appropriateness of using the term myCAF (possibly by a myCAF-specific MSigDB signature score). The foremost consideration would be the expression of α -smooth muscle actin, which is considered a hallmark of this cell type. An advantage of validating the expression of this marker as relevant to the mutualistic interaction with the treatment-resistant cancer cells would be its utility in immunohistochemical or immunofluorescence staining of fixed tissue to confirm the spatial co-localization of cell types (although this could be done with any relevant marker of the identified fibroblast subpopulation).*

A related question involves the ability of fibroblasts to activate ErbB signaling, specifically: what proportion of the signaling to ErbB receptors from the entire tumor can be attributed to fibroblasts, and more specifically, to the myCAF subtype of fibroblasts? The results shown in Fig 3A suggest that signals from fibroblasts predominate the activation of ErbB signaling. For comparison, however, it would be helpful to show the same results as shown in Fig 3B but considering the cells identified as myCAF or diploid epithelial cells as the source of ligands (possibly as supplemental figures).”

Response) We agree with the reviewer that it is not necessary to apply the myCAF label to the highly mesenchymal fibroblast population. As they point out, the conclusions are not reliant on the inclusion and focus on this subtype. The mesenchymal signatures that our data driven analyses identify perform well in stratifying fibroblast heterogeneity. Figures S9/S10 provide relevant pathway and gene level markers that distinguish cells along this axis of differentiation. We have removed the use of the term “myCAF” throughout the manuscript. This has been replaced with descriptions of the observed phenotype: “highly mesenchymal differentiated fibroblasts”.

Comment/Question) – *“The results shown in Fig 3C are unclear. The label within the figure refers to "Fibroblast myCAF differentiation" with a graphic representation of "high" appearing more mesenchymal than "low" and each secreting ErbB ligands as a key for the colors associated with the box and whisker plots shown on the right. However, it is unclear exactly what the individual points represent. The legend states that each point is an averaged value (for example from the Discovery dataset of n=238620 fibroblasts grouped by differentiation within tumor into 307 populations). Are the 307 populations of equal size? How are the myCAF values distributed among these 307 populations? Are all 307 points represented on the graph? If so, how many fall into the "high" and "low" myCAF phenotypes? Do each of the 307 subpopulations contain mixtures of cells from the different time points and patient tumors? Are myCAF differentiation-high cells overrepresented in the day 180 samples and does myCAF enrichment correspond to the increased ErbB activation in the cancer cells of each individual patient?”*

Response) We have revised this figure for clarity and biological interpretability. The reviewer correctly pointed out that representing multiple populations within a sample made the interpretation of individual datapoints less clear. The revised version now has the improved interpretation that each point reflects the mean ERBB ligand signaling of fibroblasts cells of one cell type (either highly or lowly differentiated) in a specific tumor sample. Analysis of this data gives the same conclusion; that highly mesenchymal differentiated fibroblasts contribute more ERBB signals in both the discovery and validation cohort. Statistics have been updated in the legend and pleasingly this simpler representation of the data provided greater statistical support. We have also updated the legend text of Fig 3C, with a focus on more precisely describing what the data points represent in relation to the patient data.

We have also addressed the reviewer's suggestion (below) to remove the term “myCAF” and instead describe the biological process as mesenchymal differentiation. This better connects the schematic to the cell types presented. We have provided a breakdown of the number of tumor samples with highly and lowly differentiated

fibroblast cells in the legend. Cells are not averaged across patients or timepoints. Each tumor sample will contain a unique number of highly/lowly differentiated fibroblasts and so the average will be across unequal sized samples.

Finally, we have not observed a clear overrepresentation of mesenchymal differentiated fibroblast cells post-treatment samples and instead see enhanced signaling per fibroblast. Instead, temporal changes in cellular composition of the tumors are more strongly driven by two other cell types: cancer cell populations (as expected) and immune cell types (the subject of a separate study into immune modulation).

Comment/Question) – *“The experimental results relating to induction of myCAF differentiation in response to antiestrogen treatment (Figures 4 & 5) suggest that myCAF induction should occur early in response to treatment. Have any attempts to validate this been attempted using the 14-day patient-matched treatment samples? Only data from 0 and 180-day treatment samples were described.”*

Response) We examined the ERBB signaling transition at day 14 and found that the phenotype transitions take longer in the patient setting than they do *in vitro*. Across patients, the directionality of the transition is consistent with the longer-term outcome, but the size of the effect is still small and varied across patients. There are several potential reasons that we have hypothesized may contribute to this, including:

- Pharmacokinetic limitation of drug distribution and limitations to tissue penetration.
- Scaling of signaling to concentrations needed to drive tissue-level differentiation may introduce greater delays.
- Compensatory mechanisms to maintain homeostatic regulation (e.g. counteracting signaling feedbacks).
- Environmental factors (hypoxia and resource availability) driving differences in tissue function and cellular differentiation (e.g. metabolism).

Ideally, an intermediate timepoint between days 14 and 180 would have been available to address the important question of how long the process takes in general within the patient setting.

Comment/Question) – *“Do the fibroblast cells in the coculture experiments that demonstrate mutualism exhibit increased expression of markers associated with myCAF differentiation? Is there any evidence for the mutualism to be extended to different subpopulations of fibroblasts within the same tumor (high myCAF differentiation and low EGFR activation with low myCAF differentiation and high EGFR activation)?”*

Response) In response to one of the above suggestions from this reviewer, we have switched from using the term myCAF and instead describe the mesenchymal differentiation state of the fibroblasts. In S12 we have shown that the *in vitro* fibroblasts show increased markers of mesenchymal differentiation (Collagen: COL1A RNA expression) when cocultured with cancer cells. Figure 3 and Figures S11 also show that the mesenchymal differentiation phenotype (elongation) is observable by microscopy when treated with TGFB1.

Comment/Question) – *“Experimental analysis and mathematical model of mutualism. The experimental observations of growth facilitation when fibroblasts and cancer cells are cocultured and this is affected by endocrine therapy is a strong result and the mathematical model of mutualism provides a useful tool for quantitatively assessing the strength of this effect. However, very little information about how the model was simulated, or how the parameters were estimated were provided and the model output of “cancer abundance” is hard to compare to experimental data. The change in growth rate, as shown in Fig 5C&D, would provide a much easier output to interpret (at a minimum to show how the model can effectively be parameterized to explain the experimental results). There is no explanation as to why there is a death term in the model since death does not appear to have been explicitly measured. Additionally, the stated death rate ($\mu=1e-3$) is 10X greater than the growth rate ($r_c=1e-4$); it is unclear how these parameter values are consistent with continued growth of the population. Adding a better description of the modeling (i.e., to the Methods) and providing model predictions of how interfering with the facilitated growth between cell populations would affect the effective growth rate would significantly strengthen its contribution to supporting the authors' interpretations. Ideally, the model could be used to more strongly link the ex vivo experimental results with the patient-derived data.”*

Response) We have expanded our description of this modeling component within the methods to address these questions. We also clarified and corrected the figure legend description to explain that the net proliferation rate was the value that of $1e-4$ (this is the estimated division rate ($1.1e-3$) minus the death date ($1e-3$)). We thank the reviewer for pointing this out.

We have stated that:

“Quantitative real time-PCR analysis of treatment induced gene expression changes
The mutualistic signaling between cancer and fibroblast populations was characterized by a system of ordinary differential equations. This dynamic model describes the division of cancer and fibroblast cells (intrinsic division rates of cancer and fibroblast cells in monoculture are r_c and r_f), which we assume to be regulated by density dependent competition for resources (e.g. space/growth factors/nutrients/oxygen) as the

population approaches the carrying capacity (K). Cancer cell (but not fibroblast) division is also assumed to be reduced by fulvestrant therapy (X) following hyperbolic decay function. Fulvestrant doses were rescaled so that X ranges between zero (fulvestrant = 0nM; DMSO control) and 1 (fulvestrant= 5nM). Mutualism between cell types is introduced by allowing each cell population to stimulate the division of cells of the opposite type (mutualism effects on cancer and fibroblast cells are β_C and β_F respectively). Cell death (μ) is assumed to be constant. The initial abundance of each cell type (experimental seeding number: C_0 and F_0) are assumed to be known. This led to the following system of differential equations:

$$\begin{aligned}\frac{dC}{dt} &= r_C \left(\frac{1 + \beta_C F}{1 + X} \right) C \left(1 - \frac{C}{K} \right) - \mu C \\ \frac{dF}{dt} &= r_F (1 + \beta_F C) F \left(1 - \frac{F}{K} \right) - \mu F\end{aligned}$$

Intrinsic division of each cell type and their death rate were estimated from the exponential growth measurements of monoculture populations. Linear models were used to quantify log-linear population growth with and without fulvestrant treatment. Assuming that proliferation is halted by endocrine therapy, whilst cell death is relatively unaffected, we quantified the death rate as the log linear population decline under fulvestrant treatment. Intrinsic division was then measured as the log-linear population growth under control conditions plus the estimated death rate. Mutualism effects were similarly estimated by contrasting log-linear growth rates of cocultures, during the exponential population growth phases, with the growth of cancer monocultures. The difference in growth rate was divided by the other cell type's abundance to arrive at the estimates of β_C and β_F . The carrying capacity was fixed to be 100 times C_0 .

The model was used to simulate continuous time population abundances of cancer and fibroblast cells using numerical integration. The lsoda solver of the “deSolve” R package was used to switch automatically between stiff and non-stiff methods. To evaluate the evolution of the system from initial conditions to steady state, under a given set of parameters, we used the “rootSolve” R package's steady function to apply an iterative steady-state solver using the Newton-Raphson method.”

Comment/Question) – “Computational code and data availability. The availability of all code used to generate the figures in the manuscript and access to relevant data is a major strength supporting the appropriate use of statistical and computational approaches used by the authors. However, the code does not allow for true reproducibility of the figures (at least not without substantial effort to reproduce the relevant preprocessing and processing steps required to filter and clean the data for further analysis). None of the data files referenced within the code were provided/available, other than the primary processed files accessible from GEO (with the appropriate reviewer token). The code frequently references data files generated by

other code (possible available somewhere within the repository) but that are only available on the local computer of the code author. It would be most useful to provide an overview and specific order of steps required to generate the files needed to reproduce the manuscript figures (e.g., a README document added to the git repository that provides the details necessary to reproduce the figures and links to preprocessed data, if possible).”

Response) Source data files for each figure have been revised and provided. We have revised the code repository and provided the source data that accompanies the manuscript to allow reproduction of each of the subpanels of the manuscript’s figures.

Source data (named: SourceData_Figure#_BriefTitle.csv(or .tiff etc.) is now provided for each figure, following instructions provided in the source data checklist. Each input csv file provides the data underpinning each analysis presented in the figures. We also provide image files of the representative images presented and the experimental replicates (western blots and microscope images).

Source code has been refactored and partitioned into separate scripts to perform analyses relating to each subpanel of the manuscript figures (using source data as the input to retrieve subpanel specific Figures/outputs).

The github has been updated to make this code available and the README document has been extended to provide more description of the analysis steps.

Minor Comment/Question):

“On page 7: ‘Using fluorescence microscopy, we observed induction of the elongated mesenchymal morphology of myCAFs in all $TGF\beta1$ primed patient-derived fibroblast populations but not in the unprimed controls (Fig 4E)’ There is no Figure 4E. The authors probably mean Fig 3D, but it is a transmitted light image, not fluorescence.
On page 7: ‘In all six patient-derived populations, $TGF\beta1$ primed fibroblasts upregulated both neuregulin-1 (NRG1) and human basic epidermal growth factor (HBEGF) (Fig 4F).’ There is no Figure 4F. The authors appear to mean Fig 3E.

Response) The reviewer is correct. These two sentence have been corrected and now read:

“Using transmitted-light brightfield microscopy, we observed induction of the elongated mesenchymal morphology of myCAFs in all $TGF\beta1$ primed patient-derived fibroblast populations but not in the unprimed controls (Fig 3D)” and

"In the legend of Fig 5, the authors state, "The endocrine drug effect was fixed to the half maximal value ($\mu=0.5$)," but the endocrine drug effects are described as X rather than μ ."

Response) Thank you for spotting this. The endocrine drug effect has been corrected to: *"The endocrine drug effect was fixed to the half maximal value ($X=0.5$)", matches the use in the description of panel B.*

Reviewer #3:

Study summary) "The authors performed an important study of early-stage ER+ BC patients treated with a CDK4/6 inhibitor in the neoadjuvant setting. Longitudinal studies where tumor samples are collected and profiled prior to treatment as well as during treatment are especially valuable, and the authors further distinguish their study by performing single-cell RNA sequencing, which provides an unprecedented level of insight into the molecular characteristics of the tumor samples.

This data set enabled the team to distinguish between changes to tumor cells and the tumor microenvironment, a novel and important analysis. In addition, the team also performed in vitro studies using cancer cell lines to further study the hypotheses generated from the clinical data.

A major finding that the authors highlight is that the ERBB pathway was activated in patients upon treatment, and that the mechanism of ERBB pathway activation was not solely through upregulation of the ERBB receptor. The authors analyze the single-cell RNA sequencing data to separately characterize the tumor cells and immune and other cells in the tumor microenvironment and propose that fibroblasts are inducing ERBB signaling in the tumor through tumor-fibroblast crosstalk."

Response) We are pleased that the reviewer feels that this is a novel and important study and analysis and that they appreciate the value of the longitudinal single cell patient-derived data that we have generated and analyzed. We agree that studying tumor responses during treatment is an especially valuable source of information to allow highly detailed molecular characterization of single cell changes within the cancer and non-cancer components of a tumor.

Comment/Question) – *"Throughout the text (including the title, the abstract, the body, figures), the authors refer to "endocrine therapy resistant breast cancer" without any evidence that the samples they are studying are endocrine therapy resistant.*

The cohort that the authors have examined is neoadjuvant and presumably has not received prior endocrine therapy. The cohort is then treated with endocrine +/- CDK4

inhibitors, and samples are collected post treatment. The authors do not classify the response of the tumors or seek to correlate mechanistic biomarkers with markers of sensitivity or resistance; instead they compare samples collected during treatment with pre-treatment samples. Hence it is not clear to this reviewer why the authors claim that this cohort is ET resistant nor what evidence they have to associate the molecular pathways with sensitivity to endocrine therapy. For example in the abstract the authors state "In most patient tumors, resistant cancer cells increased ERBB growth pathway activity during treatment, only partially through ERBB receptor upregulation " Unless the authors correlate response to therapy with molecular features, what evidence do they have that the molecular features are associated with resistance? Do the authors intend to communicate that they are examining the molecular changes in response to endocrine therapy in early-stage ER+ BC, which may or may not be associated with sensitivity - if so they should clarify throughout the manuscript and change the title. If the authors agree, this conceptual change in the manuscript has important implications for the interpretation of the results that the authors should state and that will require rewriting."

Response) This study builds on prior work (Griffiths et al. 2021) in which we have analyzed the response of tumors during treatment and classified tumors into categories that are either growing or shrinking over the trial period. We do not claim that this cohort is endocrine resistant. On the contrary, we identified tumors that were both growing and shrinking during treatment. In line with the reviewer's suggestions, in that study we linked the tumor's growth during treatment to phenotypic evolution and identified the convergent evolutionary shift away from estrogen dependent proliferation. We also identified that by the end of treatment cancer cells across tumors exhibited cell cycle reactivation, as measured clinically by KI-67 antigen expression and tumor size trajectories and detected transcriptionally.

Here we assess how, at the single cell level, individual cancer cells are resistant. We examine how cancer cells alter their phenotype during treatment to overcome treatment induced cytostatic effects. Throughout the manuscript we have reviewed our wording to clarify that we are examining cellular resistance (across tumors) and contrast cells that are treatment naive with those that survived post treatment and that have been found to have reinitiated proliferation.

Comment/Question) – *"The authors do not appear to separate the samples collected prior to surgery at day 14 from those collected after surgery at day 180, and group both together as post treatment. One might imagine that there are significant differences between samples collected at day 14 and day 180 - have the authors investigated these differences and why were they grouped together?"*

Response) Pre- and post-treatment samples were kept separate from those at day 14. Phenotype evolution analyses were conducted at the single cell level, with hierarchical model structures used to account for the non-independence of cells within a sample. Similarly, communication analyses measured signaling within tumor samples. “Post treatment” refers specifically to samples collected at day 180, following treatment. In contrast day 14 samples are referred to as “early follow-up” samples. We have investigated temporal changes between cancer cell phenotypes between pre-treatment, early follow-up and post-treatment samples. Post-treatment samples showed much more striking upregulation of ERBB signaling compared to day 14. For this reason and to simplify the message, we focus in the main text on the contrast between pre- and post-treatment cancer cells.

We have described this explicitly in the methods, stating:

“During treatment, patients provided three mandatory core tumor biopsies (via 14-gauge needle) at screening (pre-treatment: Day 0), cycle 1 follow up (early follow-up: Day 14), and at end of trial (Post-treatment: Day 180).”,

“Cancer cells from these patients (n=134218 cells (87913 pre-treatment; 46305 post-treatment), 41 paired pre-/post-treatment tumor samples) were analyzed using hierarchical regression to assess how the distribution of single cell phenotypes within each tumor evolved pre- and post-treatment, allowing identification of consistently emerging phenotypes”.

Comment/Question) – *“The authors note that “Activation of the ERBB signaling pathway was also observed in normal diploid epithelial cells of these post treatment tumor samples, indicating that it may be induced by treatment and TME communication changes rather than selected for during subclonal cancer evolution”. How do the authors align this observation with their conjecture that ERBB upregulation is a mechanism of tumor escape to treatment? Would the null hypothesis simply be that ERBB upregulation is a response to endocrine treatment and not selected as a mechanism of resistance?”.*

Response) We have clarified that the degree of ERBB upregulation in cancer cells is much greater than that seen in diploid epithelial cells. This does support the hypothesis that it is a tumor mechanism of resistance that selects for upregulation of the growth pathways that diploid epithelial cells may utilize. There are multiple hypotheses for why diploid epithelial cells may show this upregulation (to a smaller extent) that we are actively investigating. First, it is possible, especially with ER+ breast cancers with minimal genetic rearrangements (known to occur), that some diploid epithelial cells are cancer subclones with few copy number alterations. Secondly, it is possible that the upregulated ERBB signaling in the tumor microenvironment impacts all cells in/around

the tumor, independent of whether they are cancer or normal epithelial cells. Importantly, without the capability of classifying all diploid epithelial cells as non-cancer or cancer in this experiment, we cannot rule out these hypotheses and require the development of more sophisticated computational tools to classify these cells. We do agree with the reviewer that this limitation prevents us from reaching such strong conclusions from this subset of cells and have removed the statement from the text.

Comment/Question) – *“The relatively fast upregulation of ERBB2 signaling in vitro at 1 week suggests that this made simply be a marker of treatment - how can the authors demonstrate that it leads to treatment resistance?”*

Response) We performed the in vitro experiments blocking compensatory ERBB upregulation to demonstrate that ERBB signaling provides treatment resistance (Figure 6; Figure S13-S14). Experimental results showed that when the ERBB inhibitor binds to the upregulated ERBB receptors and blocks signal transduction, endocrine sensitivity is increased. The ERBB inhibitor effectively targets diverse resistant cancer cell lines, by blocking proliferation and by controlling fibroblast guided invasion.

We have highlighted this in the results of the main text stating that compared to endocrine alone combination afatinib and endocrine treatment led to: “slower cancer growth rates and spheroid shrinkage [being] frequently achieved (Fig EV5). This finding demonstrates that blocking ERBB signaling can increase endocrine sensitivity.”

Comment/Question) – *“Figure 1 is a conceptual overview of the study with a long caption that recapitulates some of the body of the paper. The authors do not provide detail on the number of samples collected at each time point in the figure - making it difficult to interpret the subsequent work. In the text, the authors distinguish samples collected prior to, during and after treatment ‘...serially collected from 173 tumor biopsies from 62 patients taken prior to, during and after treatment.’ The authors do not state the duration of treatment and should have an overview of the cohort that includes this information - a missed opportunity is Figure 2 where no distinction is made between the 14 day timepoint and the 180 day timepoint and so it is unclear if ‘post-treatment’ refers to one or both of those timepoints.”*

Response) In the legend of Figure 1, we have clarified the time points of the pre- and post-treatment samples that we examined and the duration of the trial. We also included the number of patient tumors with paired pre- and post-treatment cancer samples for the discovery and validation cohort.

We state that:

resistance. Further, the ability to reverse resistance by inhibiting this pathway provides evidence of this pathway's role in ET/CDKi drug response.

This approach was used to test a key prediction of the patient derived analyses, namely that activation of ERBB signaling may contribute to endocrine therapy resistance through the alternative ERBB-ERK proliferative pathway. Thus, the endocrine resistant cell lines were chosen as they represent the lack of response to endocrine therapy seen in the post-treatment patient tumors that continue to proliferate during treatment. As a further connection between the single cell patient data and in vitro results, we have clarified the connection of experimental treatments and our patient derived predictions. We state that: "we assessed the therapeutic efficacy of inhibiting the identified mechanisms of GF signal-mediated proliferation to i) target the resistance phenotype of persistent cancer cells observed in patient tumors and ii) block growth promoting cancer-non-cancer cell communications underpinning an oncogenic mutualism that fuels cancer cell growth."

Lastly, as the reviewer requested, we edited the statement for clarity: "*Patient derived insights, validated across independent cohorts, and multiple in vitro model systems of endocrine and ribociclib resistance support the role of ERBB signaling activation in ER+ breast cancer growth during endocrine therapy.*"

Comment/Question) – *"The authors note that the resistant cell lines upregulate ERBB pathway activity. In the patients, the authors propose a mechanism whereby the tumor secretes TGFB which in turn stimulates fibroblasts to release EGF ligands that lead to ERBB pathway activation. If this is the mechanism of ERBB activation, what is the relevance of the in vitro cell model system that does not include fibroblasts, and likewise what is the relevance of the fibroblast mutualism if ERBB activation occurs in vitro in the absence of fibroblasts? The fact that the in vitro cell model upregulates ERBB through a purely tumor intrinsic mechanism suggests that the mutualism is not the exclusive mechanism of ERBB upregulation and begs the question of how relevant the effect of the fibroblasts is. The studies of tumor-fibroblast coculture reported in Fig 6 may be able to demonstrate the importance of mutualism: the authors should clarify how this connects with the original resistance models generated in the absence of fibroblasts."*

Response) Our tumor-wide communication analysis and in vitro analyses both suggest that the general level of ERBB ligands depends on the contribution of both cancer and fibroblast cell types. We see evidence in both the patient and in vitro setting that autocrine ERBB signaling plays a role. We show this in figure 3A. Facilitation of cancer cells by neighboring cancer cells is something that we have measured in these cell lines (see Emond et al. Nature Communications 2023).

References

Emond R, Griffiths JI, Grolmusz VK, Nath A, Chen J, Medina EF, Sousa RS, Synold T, Adler FR & Bild AH (2023) Cell facilitation promotes growth and survival under drug pressure in breast cancer. *Nat Commun* 14: 3851

Farmaki, Elena, et al. (2023) ONC201/TIC10 enhances durability of mTOR inhibitor everolimus in metastatic ER+ breast cancer. *Elife* 12: e85898.

Griffiths, JI., et al. (2021) Serial single-cell genomics reveals convergent subclonal evolution of resistance as patients with early-stage breast cancer progress on endocrine plus CDK4/6 therapy. *Nature cancer* 2.6 658-671.

Grolmusz VK, Chen J, Emond R, Cosgrove PA, Pflieger L, Nath A, Moos PJ & Bild AH (2020) Exploiting collateral sensitivity controls growth of mixed culture of sensitive and resistant cells and decreases selection for resistant cells in a cell line model. *Cancer Cell Int* 20: 253

21st Jan 2025

Manuscript Number: MSB-2024-12513R

Title: Blocking cancer-fibroblast mutualism inhibits proliferation of endocrine resistant breast cancer.

Author: Jason Griffiths

Dear Dr. Griffiths,

Thank you again for submitting your revised work to Molecular Systems Biology. We have now heard back from two of the original three reviewers who we asked to re-evaluate your study. As you will see below, the reviewers are supportive on the resource value of your datasets as well as the novel analyses and validation in vitro in cancer cell lines. However, both Reviewers 2 and 3 point out some of their comments were not fully addressed, a few potentially due to ambiguous wording. We would therefore ask you to address their concerns in a revision. Please let me know in case you would like to discuss in further detail any of the any of the reviewer comments, I would be happy to schedule a call.

We remind you that we have the following formatting requirements:

1) A .docx formatted version of the manuscript text (including legends for main figures, EV figures and tables). Please make sure that the changes are highlighted to be clearly visible. Alternatively you may choose to submit your manuscript as a LaTeX file.

4) A .docx formatted letter INCLUDING the reviewers' reports and your detailed point-by-point responses to their comments. As part of the EMBO Press transparent editorial process, the point-by-point response is part of the Peer Review File (PRF), which will be published alongside your paper.

5) A complete author checklist, which you can download from our author guidelines (<https://www.embopress.org/page/journal/17574684/authorguide#submissionofrevisions>). Please insert information in the checklist that is also reflected in the manuscript. The completed author checklist will also be part of the PRF.

6) Please note that all corresponding authors are required to supply an ORCID ID for their name upon submission of a revised manuscript.

7) It is mandatory to include a 'Data Availability' section after the Materials and Methods. Before submitting your revision, primary datasets produced in this study need to be deposited in an appropriate public database, and the accession numbers and database listed under 'Data Availability'. Please remember to provide a reviewer password if the datasets are not yet public (see <https://www.embopress.org/page/journal/17574684/authorguide#dataavailability>).

In case you have no data that requires deposition in a public database, please state so in this section as follows: "This study includes no data deposited in external repositories". Note that the Data Availability Section is restricted to new primary data that are part of this study.

8) All Materials and Methods need to be described in the main text using our 'Structured Methods' format, which is required for all research articles. According to this format, the Methods section includes a Reagents and Tools Table (listing key reagents, experimental models, software and relevant equipment and including their sources and relevant identifiers) followed by a Methods and Protocols section describing the methods using a step-by-step protocol format. The aim is to facilitate adoption of the methodologies across labs. Please upload the Reagents and Tools table as a separate document when submitting your revised manuscript. More information on how to adhere to this format as well as a downloadable template (.docx) for the Reagents and Tools Table can be found in our author guidelines:

<https://www.embopress.org/page/journal/17444292/authorguide#structuredmethods>

9) For data quantification: please specify the name of the statistical test used to generate error bars and p-values, the number (n) of independent experiments (specify technical or biological replicates) underlying each data point and the test used to

Reviewer #2:

Overall, the reviewers generally addressed the specific critiques and comments and improved the manuscript accordingly. The updates to the code repository, the inclusion of the specific code for reproducing the figures, and especially the updated README file that includes the full description and appropriate links for the multiple code resources applied, made following the methods much easier. I reiterate that the public access to the code and data is a major strength of the manuscript, and I encourage even more annotation that facilitates others investigating this exceptional resource.

However, there remains several aspects of the manuscript that should be addressed prior to publication. Following are the authors' specific responses to original reviewer comments/critiques that require further clarification/correction and a few new questions that arose from the revised manuscript.

Questions about cancer-associated fibroblasts

Reviewer #2 Comment/Question

"Do the fibroblast cells in the coculture experiments that demonstrate mutualism exhibit increased expression of markers associated with myCAF differentiation? Is there any evidence for the mutualism to be extended to different subpopulations of fibroblasts within the same tumor (high myCAF differentiation and low EGFR activation with low myCAF differentiation and high EGFR activation)?"

Authors' Response

In response to one of the above suggestions from this reviewer, we have switched from using the term myCAF and instead describe the mesenchymal differentiation state of the fibroblasts. In S12 we have shown that the in vitro fibroblasts show increased markers of mesenchymal differentiation (Collagen: COL1A RNA expression) when cocultured with cancer cells. Figure 3 and Figures S11 also show that the mesenchymal differentiation phenotype (elongation) is observable by microscopy when treated with TGFB1.

Reviewer's Rebuttal: The original question may not have been worded well. It was asking whether there was any mutualism detected between fibroblast subpopulations. For example, do low-mesenchymal fibroblasts respond to high-mesenchymal fibroblasts, or vice versa? Also, phenotypic changes and collagen expression are suggestive of more mesenchymal features but scRNA-seq would seem to be a much more powerful way of defining the specific changes induced by coculture and treatment that could subsequently be investigated in the human tumor data to further support the authors' hypothesis of mutualism in the clinical cohort.

Reviewer #2 question (related to question above)

What proportion of the fibroblast population would be considered myCAF? Are they specifically enriched in the cells present after 180 days of treatment versus other fibroblast subpopulations?

[no response was provided by the authors.]

It remains an open question whether the abundance/proportion of the more mesenchymal fibroblasts is greater after 180 days of treatment (compared to pretreatment in the same patient). Similarly, is there any correlation of tumor size at 180 days with more mesenchymal fibroblasts? That would seem to be a prediction of the mutualism model.

Reviewer #3 comment/question

The fact that the in vitro cell model upregulates ERBB through a purely tumor intrinsic mechanism suggests that the mutualism is not the exclusive mechanism of ERBB upregulation and begs the question of how relevant the effect of the fibroblasts is.

Response:

Our tumor-wide communication analysis and in vitro analyses both suggest that the general level of ERBB ligands depends on the contribution of both cancer and fibroblast cell types. We see evidence in both the patient and in vitro setting that autocrine ERBB signaling plays a role. We show this in figure 3A. Facilitation of cancer cells by neighboring cancer cells is something that we have measured in these cell lines (see Emond et al. Nature Communications 2023).

Rebuttal:

The question remains about the relative contribution of ErbB activation emanating from fibroblasts vs tumor (or other) cells. Can this be quantified in any way?

Minor critiques

Wording is occasionally unclear and imprecise. For example, in the introduction the authors wrote: we revealed that endocrine and CDK4/6 inhibition therapy resistance involves cancer cell signaling shifting from estrogen to alternative growth signal-mediated proliferation

Is the following sentence what the authors intended to mean? - we revealed that cancer cell resistance to endocrine and CDK4/6 inhibition therapy involves shifting from estrogen-mediated signaling to signaling driven by alternative growth factors

The authors refer to the pooling of samples from the discovery and validation cohorts as a "greatly expanded dataset." This should be removed.

The number of values (n value) represented in each boxplot should be provided for all figures.

On page 15 of the revised manuscript the authors wrote, "Single-cell RNA-sequencing was performed on nuclei extracted from OCT embedded core tumor biopsies (as previously described (Griffiths et al, 2021; Sei et al, 2018). Single cell RNA-sequencing was performed on single nuclei suspensions using 10X Genomics Chromium platform (as previously described in Griffiths et al, 2021)."

Presumably, these sentences should be combined into one. It is unclear why it is repeated with slight variation.

In general, it would have been helpful to point out specific changes to the wording applied to the main text either by highlighting, bolding, etc., directly in the revised manuscript and by pointing out the specific changes in the response to reviewers.

Reviewer #3:

From the first submission:

The authors performed an important study of early-stage ER+ BC patients treated with a CDK4/6 inhibitor in the neoadjuvant setting. Longitudinal studies where tumor samples are collected and profiled prior to treatment as well as during treatment are especially valuable, and the authors further distinguish their study by performing single-cell RNA sequencing, which provides an unprecedented level of insight into the molecular characteristics of the tumor samples.

This data set enabled the team to distinguish between changes to tumor cells and the tumor microenvironment, a novel and important analysis. In addition, the team also performed in vitro studies using cancer cell lines to further study the hypotheses generated from the clinical data.

A major finding that the authors highlight is that the ERBB pathway was activated in patients upon treatment, and that the mechanism of ERBB pathway activation was not solely through upregulation of the ERBB receptor. The authors analyze the single-cell RNA sequencing data to separately characterize the tumor cells and immune and other cells in the tumor microenvironment, and propose that fibroblasts are inducing ERBB signaling in the tumor through tumor-fibroblast crosstalk.

The authors addressed the comments on the first submission from this reviewer. Remaining comments below are relatively minor.

The authors state that there are a total of 62 patient tumors of which 35 are in the discovery and 27 in the validation cohort, with a total of 39 from patients receiving combination ribociclib and 23 receiving letrozole alone. Of these, the authors state that there are a total of 41 matched tumor samples, and according to Fig 2, 16 of these are in the discovery cohort and 25 in the validation cohort. It would be more clear for the authors to note that their analysis combined patients treated with combination ribociclib with those patients receiving letrozole alone.

The authors note that "As post-treatment cancer cells show activation of the ERBB signaling pathway, we took a parallel experimental approach to test whether endocrine and CDK4/6i treatments drive compensatory ERBB receptor expression in vitro." This reviewer assumed that the CRR cell lines are intended to correspond to the resistant tumor samples post-treatment with CRR. If that is the case, would one expect that resistance mechanisms such as increased ERBB signalling would already be upregulated in the CRR cell lines even in the absence of endocrine treatment. Could the authors comment on this - e.g. one way to test would be to compare the levels of ERBB genes and signaling between baseline ETR cell lines and the induced resistance CRR derivatives?

The Results sections states that "Furthermore, endocrine therapy induced greater phosphorylation of ERBB receptors, showing rapid dose-dependent compensatory activation of ERBB signaling across endocrine and combination ribociclib resistant cell lines." And the Figure caption reiterates "Broad ERBB activation and phosphorylation across all cancer cell lines under endocrine therapy." This is not consistently apparent in Fig 2C. Can the authors be more clear about their claim regarding

activation of ERBB signaling under endocrine treatment both in the Results section and the figure caption?

Fig 4.B -Validation cohort - The results sections states "The increased TGF β 1-3 cancer to fibroblast crosstalk was found in both the discovery and validation cohort tumors." However there appears to be much lower TGF β 1,2,3 signaling upregulation in fibroblasts compared to discovery cohort?

Fig 5 - Nice to demonstrate facilitation of tumor+fibroblast relative to tumor alone. Do you have negative controls where you tested tumor + other immune component to show lack of mutualism?

In section 5, the authors show that there is statistically significant increase in tumor and fibroblast growth rates in coculture relative to monoculture. It is difficult to interpret the magnitude of the effect on growth rate in Figure 5. Can the authors help clarify - as well as provide units for the graphs?

In section 5, the authors note that "The model describes the cancer and fibroblast population growth (r_i and K_i represent intrinsic growth rate and carrying capacity of cell type i), with each cell type facilitating the proliferation of the other (β_C = cancer facilitation/fibroblast; β_F = fibroblast facilitation/cancer cells)." which implies that there is both a carrying capacity K for the cancer cells and a separate carrying capacity for the fibroblasts. However, there is only a single K variable listed in Fig. 5E - Can the authors clarify? It would also help if the authors clarified the units and the interpretation of Fig 5E.

Similar comments for Section 6 and Figure 6 regarding how to interpret growth rate differences as well as the units.

In Section 6, the authors claim that "Under afatinib treatment, cocultured fibroblasts provided some support to accelerate cancer growth of several cell lines compared to afatinib treated cancer monocultures." In Figure 6, it appears that afatinib has comparable effect on coculture vs. monoculture cancer cells - can the authors clarify?

Minor suggestions

- Page 1 change "Resistant cancer cells upregulate ERBB growth factor-receptors and stimulate fibroblasts to provide corresponding ligands."
- Page 3 change "Ecological theory predicts that such oncogenic mutualisms may become stronger under more stressful treatment conditions (Bertness & Callaway, 1994; Hammarlund & Harcombe, 2019; Michalet et al, 2014)."
- Page 5 change "Activation of the ERBB signaling pathway was also observed to a lesser extent in diploid epithelial cells of these post treatment tumor samples (Fig EV1/C)."
- Fig.1 - Upper right claims that "Test results in separate patient cohort". The wording is potentially confusing as the results are tested in a validation cohort that is from the same clinical study. Recommend changing to "Test results in validation cohort".
- Fig. 1 - Lower panel entitled "Cancer Non-cancer Mutualism". Would it be more clear to title this section as something like "Test cancer mutualism in vitro"?

Point-by-point responses to reviewers' reports for: "Blocking cancer-fibroblast mutualism inhibits proliferation of endocrine therapy resistant breast cancer"

We appreciate the reviewer's detailed assessment and constructive appraisal of our manuscript. In this document, we address the remaining concerns (italicized) sequentially, with a point-by-point description (in blue colored text) of how each comment or question about the manuscript has been addressed. Manuscript changes have been highlighted.

REVIEWER COMMENTS

Reviewer #2:

Reviewer #2 summary: *"Overall, the reviewers generally addressed the specific critiques and comments and improved the manuscript accordingly. The updates to the code repository, the inclusion of the specific code for reproducing the figures, and especially the updated README file that includes the full description and appropriate links for the multiple code resources applied, made following the methods much easier. I reiterate that the public access to the code and data is a major strength of the manuscript, and I encourage even more annotation that facilitates others investigating this exceptional resource. However, there remains several aspects of the manuscript that should be addressed prior to publication. Following are the authors' specific responses to original reviewer comments/critiques that require further clarification/correction and a few new questions that arose from the revised manuscript."*

Response) We are glad that we were broadly able to address the constructive criticisms raised by the reviewer and that our updates to code, data availability and documentation improved the manuscript. Below we will provide further clarification/correction regarding outstanding points.

Comment/Question 1) – "Reviewer #2 Comment/Question: *Do the fibroblast cells in the coculture experiments that demonstrate mutualism exhibit increased expression of markers associated with myCAF differentiation? Is there any evidence for the mutualism to be extended to different subpopulations of fibroblasts within the same tumor (high myCAF differentiation and low EGFR activation with low myCAF differentiation and high EGFR activation)?"*

Authors' Response: *In response to one of the above suggestions from this reviewer, we have switched from using the term myCAF and instead describe the mesenchymal differentiation state of the fibroblasts. In S12 we have shown that the in vitro fibroblasts show increased markers of mesenchymal differentiation (Collagen: COL1A RNA expression) when cocultured with cancer cells. Figure 3 and Figures S11 also show that the mesenchymal differentiation phenotype (elongation) is observable by microscopy when treated with TGFB1.*

Reviewer's Rebuttal: *The original question may not have been worded well. It was asking whether there was any mutualism detected between fibroblast subpopulations. For example, do low-mesenchymal fibroblasts respond to high-mesenchymal fibroblasts, or vice versa? Also, phenotypic changes and collagen expression are*

suggestive of more mesenchymal features but scRNA-seq would seem to be a much more powerful way of defining the specific changes induced by coculture and treatment that could subsequently be investigated in the human tumor data to further support the authors' hypothesis of mutualism in the clinical cohort."

Response) "We have carefully considered this question. While it is indeed interesting, addressing it is currently outside our abilities in the experimental system. Specifically, the performance of experiments to detect mutualism between fibroblast cell subtypes would require that we can coculture multiple labelled fibroblast populations, each with a stable *low-mesenchymal or high-mesenchymal phenotype*. As the in vitro (and in vivo) fibroblast phenotypes are quite plastic, we do not currently have the ability to "fix" the phenotype during the experiment or to track phenotypic change from mutualistic/antagonistic effects during the coculture. Due to this limitation, we feel that presently we are unable carry out this experiment with high confidence in the results. As such, we respectfully note that this falls beyond current capabilities and the scope of the current work, which is focused on cancer cell and fibroblast interactions.

We agree that scRNAseq is a powerful tool for defining specific changes induced by treatment, and the signatures developed within the study of our current patient cohorts can be applied to broader human cancer patient cohorts to assess their translational utility. While we are comfortable using experimental systems to study focused hypotheses, we believe that signatures identified from the FELINE patient setting provide a more general and representative assessment of fibroblast phenotype continuum and diversity with patient tumors with which to develop relevant predictors of patient outcomes. These can indeed be applied to future validation cohorts when additional patient-derived scRNAseq samples are developed.

Comment/Question 2) – "Reviewer #2 Comment/Question: *What proportion of the fibroblast population would be considered myCAF? Are they specifically enriched in the cells present after 180 days of treatment versus other fibroblast subpopulations?*

Authors' Response: *[no response was provided by the authors.]*

Reviewer's Rebuttal: *It remains an open question whether the abundance/proportion of the more mesenchymal fibroblasts is greater after 180 days of treatment (compared to pretreatment in the same patient). Similarly, is there any correlation of tumor size at 180 days with more mesenchymal fibroblasts? That would seem to be a prediction of the mutualism model."*

Response) We apologize for not providing the reviewer with an answer to this question in the previous revision. To address this, we have applied the unbiased clustering-based approach, suggested by this reviewer (in Comment/Question 5 below), and identified clusters of fibroblasts that are enriched in the respective signals of EGFR activated, mesenchymal and resting/progenitor fibroblasts (Appendix Fig S4B; full figure added to appendix). We grouped higher resolution clusters into these three fibroblast categories by applying hierarchical clustering to the Euclidean distance in PCA (linear embedding) space between cluster centroids. The clustering-based results were highly consistent with results from our prior analyses of the dynamic changes in gene expression across the UMAP representation of fibroblast heterogeneity.

Appendix Figure S4B) Overlay of fibroblast cluster annotations (color) onto the single cell fibroblast UMAP landscape (axes; small points=single cells) shows strong agreement in inferred fibroblast similarity between approaches. Fibroblast cluster centroids (large points) calculated by the median UMAP dimension scores. Biological interpretation of clusters (bold text) determined by biological function of marker genes (non-bold text).

We next quantified the percentage of fibroblasts within each tumor sample that were classified into resting/progenitor, EGFR activated or mesenchymal states. We used a linear mixed effects regression model to detect changes in logit-transformed frequency of each fibroblast cell class during treatment, whilst accounting for patient specific variation in initial fibroblast composition. The analysis showed that across patients, the fraction of EGFR activated fibroblast was significantly increased during treatment (see Appendix Figure S4D below). These cells showed the activation of downstream ERK transcription factors and a strong proliferation signal in the ssGSEA data. When looking at cell fractions, as opposed to absolute numbers, it makes sense that we see the increased representation of the highly proliferative cell type. The expansion of this proliferative population is in line with the prediction of our mutualism hypothesis that both cancer and fibroblasts should benefit from the exchange of growth factor communications. We also see this enhanced proliferation of fibroblasts in cancer cocultures (vs fibroblast monocultures) during in vitro experiments. Interestingly, despite the proliferation in EGFR activated cells, we still observe no significant change in the proportion of mesenchymal fibroblasts, tentatively suggesting a possible increase in their absolute abundance in many tumors. In future work, we will be interested to verify this tentative suggestion, perhaps through assessment of mesenchymal marker presenting fibroblast abundance per unit area of FFPE slides from endocrine treated tumors. The result that mesenchymal cells dominate the signaling provision to cancer cells, despite not showing such significant increases in abundance reinforces their substantially increased cellular production of growth factor ligands.

Appendix Figure S4D) Ternary plot showing the fibroblast composition of tumor samples pre- (blue) and post-treatment (red) in terms of the fraction of fibroblasts classified using an unsupervised clustering into three phenotypes with marker genes reflecting: resting/progenitor, EGFR activated and mesenchymal states respectively. Points= fibroblast composition in a specific tumor sample. Significant increase in the logit-proportion of EGFR activated fibroblasts post-treatment (est=5.7, se=1.57, df=24.28, z=3.62, p=0.001). No significant change in the logit-proportion of mesenchymal fibroblasts post-treatment (est=-3.74, se=1.95, df=28.8, z=-1.928, p=0.06). Sample size (n): 15908 fibroblasts from 33 patient tumors with pre- and post-treatment fibroblast samples.

Finally, we followed the reviewer's additional suggestion to assess the relationship between fibroblast subtype frequency and post-treatment (day 180) residual tumor size (Appendix Figure S4E; presented in the appendix). We used and generalized additive model (GAM) to assess the non-linear association between these variables. Following the identification of proliferation of the EGFR activated population post-treatment, we specifically assessed the association between the frequency of this fibroblast subtype in a tumor after therapy and the post-treatment pathological longest length measurement (mm). In support of the mutualism model, tumors with a greater fraction of EGFR activated proliferative fibroblasts were significantly larger post treatment.

Appendix Figure S4E) Scatterplot showing the relationship between post-treatment EGFR activated fibroblast fraction (logit scale) and the tumor's post-treatment pathological longest length measurement (mm). Tumors with a greater fraction of fibroblasts in an EGFR activated state after treatment (day 180) were significantly larger post treatment (Generalized additive model (GAM) non-linear trend: $\text{eff.df}=1, F=6.46, p=0.019$). Solid line=nonlinear relationship between post-treatment fibroblast fraction and tumor size. Shaded area=95% confidence interval. ($\pm 1.96 \times \text{SE}$). Sample size= 22 tumors with matched post-treatment measurement of EGFR activated fibroblast fraction and pathological assessment of tumor size.

We have summarized these results in the main text by stating: “We observed a significant increase in the fraction of EGFR activated fibroblasts post-treatment and found that an increased frequency of this cell type was associated with greater post-treatment residual tumor burden (pathologically measured longest tumor length) (Appendix Fig S3).”

Comment/Question 3) – “Reviewer #3 Comment/Question: The fact that the *in vitro* cell model upregulates ERBB through a purely tumor intrinsic mechanism suggests that the mutualism is not the exclusive mechanism of ERBB upregulation and begs the question of how relevant the effect of the fibroblasts is.

Authors' Response: Our tumor-wide communication analysis and *in vitro* analyses both suggest that the general level of ERBB ligands depends on the contribution of both cancer and fibroblast cell types. We see evidence in both the patient and *in vitro* setting that autocrine ERBB signaling plays a role. We show this in figure 3A. Facilitation of cancer cells by neighboring cancer cells is something that we have measured in these cell lines (see Emond et al. *Nature Communications* 2023).

Reviewer's Rebuttal: The question remains about the relative contribution of ErbB activation emanating from fibroblasts vs tumor (or other) cells. Can this be quantified in any way?”

Response) In the patient setting, we estimate that fibroblast cells provide the majority of supplementary ERBB ligand signals. This is quantified for both the discovery and validation cohort by the line width of the network diagram in figure 3A. These findings are consistent with prior in vitro results showing that ERBB ligand expression by fibroblasts is significantly higher than in our cancer cell lines (T47D and MCF7) (Berdier-Acer et al. 2021). In this work, the NRG1 ligand expression of epithelial and stromal cells was also compared between different laser capture microdissected (LCM) breast cancer datasets (GSE10797, GSE14548, GSE35019 and GSE83591). Again, stromal cells showed enhanced ligand production. Together this quantitatively supports our scRNA-seq derived patient insight that fibroblasts are the major contributor of supplementary ERBB growth signals.

We have included a statement describing this quantitative support for the role of fibroblasts as dominant signal contributors in the results section by stating: “These results are consistent with prior findings that ERBB ligand expression is significantly higher in fibroblasts than in cancer cell lines (T47D and MCF7) and higher in stromal versus epithelial compartments of various laser capture microdissected (LCM) breast cancer tumor samples (Berdier-Acer et al. 2021).”

References

Berdier-Acer M, Maia A, Hristova Z, Borgoni S, Vetter M, Burmester S, Becki C, Michels B, Abnaof K, Binenbaum I, *et al* (2021) Stromal NRG1 in luminal breast cancer defines pro-fibrotic and migratory cancer-associated fibroblasts. *Oncogene* 40: 2651–2666

Comment/Question 4) – “Reviewer #2 Comment/Question: *[Authors wrote] Pathway scores are a composite of 48 ERBB activation ssGSEA signatures from the C2 collection (listed in Fig S4), but there is no description of how the composite score was calculated in Methods.*

Authors' Response: *Measuring composite phenotypes scores: For pathways with correlated activity within cells, UMAP was used to perform non-linear dimension reduction. Higher dimensional pathway data was projected into a low-dimensional latent space, preserving the global structure and providing a summary of the dominant patterns of variation across single cells. The first UMAP dimension was used to measure the composite phenotype score as it quantifies the major axis of phenotypic variation.*

Reviewer's Rebuttal: *But in the revised Methods the authors state: ‘For pathways with correlated activity within cells, PCA was used to perform dimension reduction. The higher dimensional set of pathway scores related to a key phenotype (e.g. ERBB signaling) was projected into a low-dimensional latent space, preserving the global structure and providing a summary of the dominant patterns of variation across single cells. Top principal component dimensions were used to measure the composite phenotype score, by quantifying the major axis of phenotypic variation that correlates with diverse ERBB ssGSEA pathways.’ This inconsistency must be fixed. Presumably the authors meant to use “the first principal component” in both their response and in the main text, but there is no way to know which is correct from the code provided in the*

referenced GitHub repository since only steps after preprocessing were provided. The UMAP dimensions are nonlinear embeddings and, as such, are inappropriate for interpretation in the way the authors may be applying it (effectively as a linear correlate). The interpretation of UMAP dimensions also relates to the authors' use of the UMAP embeddings to interpret heterogeneity of the fibroblast mesenchymal and proliferative state in the next critique."

Response) We thank the reviewer for pointing out this error in the previous point-by-point response document. The main text provided the correct description of the method used, which as the reviewer suggests was PCA. We overlooked to correct this in the response document and apologize for the previous inconsistency between the two documents. Thank you for your careful examination of both texts.

Comment/Question 5) – “Reviewer #2 Comment/Question: [regarding the UMAP axis labels in Fig S9A (now Figure EV3A?)] how are the specific UMAP dimensions being attributed to the specific axis labels?

Authors' Response: We have clarified the description of how the biological interpretation of the UMAP dimensions were obtained. In the legend of figure S9 we explain that: "the biological interpretation of UMAP dimensions (axis labels) were determined by assessment of the genes, gene sets and communications that correlated with each dimension."

Reviewer's Rebuttal: The label of the UMAP dimensions is inappropriate, especially considering that the axes themselves are arbitrary nonlinear embeddings and represent more than what the authors suggest with their label. Especially important is that, while the data presented suggest the variation exists across the UMAP dimensions, the UMAP dimension values cannot be attributed to the authors' labels (i.e., a UMAP1 value of 3 cannot be interpreted as having more EGF stimulation than a value of 2 due to the nonlinear nature of the embedded space). A separate annotation assisting the reader to interpret the gradient along this axis may be appropriate, but not the axis label itself. This gradient of expression across the cells represented in a UMAP-embedded space may be more easily interpreted if an unbiased clustering approach was performed (e.g. Leiden) and specific clusters were shown to be enriched or depleted of the respective signals. (The authors may be doing something like this in their approach, but it should be clarified.) This would also facilitate addressing specific questions about the relative abundance of cells within clusters that can be compared for various phenotypes (e.g. mesenchymal differentiation, EGF proliferative signaling, etc.) and relative abundance of these cells within each independent tumor sample and whether or not specific phenotypes are enriched after 180 days of treatment; i.e., this could address the question posed by reviewer #2, asking: Are [author-identified fibroblast populations] specifically enriched in the cells present after 180 days of treatment versus other fibroblast subpopulations?"

Similarly, in their response to reviewers the authors wrote that they clarified their description of how they assessed cellular heterogeneity with the scores derived from ssGSEA pathway information and included: We state that: "For each broad cell type, we generated a cell-type specific UMAP based on ssGSEA pathway scores (4775

pathways per cell; c2+hallmark). This contributed to dimension reduction and the zinbwave' normalization process also corrected for biases associated with single cell read counts and gene length and GC-content."

This wording seems to suggest that the authors may have used UMAP dimension as quantitative values of heterogeneity rather than as an aid to visualizing the high dimensional data. This must be further clarified. The use of linear embeddings would be more appropriate than nonlinear embeddings, but the alternative suggestion above to compare clusters of cells with similar features would facilitate the quantitative assessment of gene expression changes without relying on nonlinear embedding dimensions themselves."

Response) First, we have edited the axis labels of figure EV3A to indicate that they are showing UMAP dimensions. As suggested, we added a separate annotation arrow assisting the reader to interpret the gradients of pathway activation and mesenchymal differentiation along each axis.

Second, we followed the reviewers suggestion to use unsupervised clustering and differential expression analysis as a complementary line of evidence to support our interpretation of the major axes of fibroblast phenotypic heterogeneity. This analysis was described in the response to Comment/Question 2 (above). We obtain consistent results to those obtained previously (Appendix Fig S3/S4). Fibroblast clusters were segmented across UMAP space, indicating that linear (PCA) and non-linear embeddings were broadly consistent. The most differentially expressed genes in the major clusters supported our biological interpretation of the phenotypes of cells in each fibroblast category. This allowed us to define three major fibroblast states (Resting/progenitor, EGFR activated and mesenchymal) that directly correspond to the fibroblast phenotypes identified in the prior analyses reported in the manuscript. We thank the reviewer for their suggestion.

We have stated in the main text that: "To support this interpretation, unbiased clustering was performed to identify fibroblast subpopulations and differential expression confirmed the upregulation of mesenchymal and EGFR response markers genes in clusters that aligned closely with the major axes of the UMAP landscape (Appendix Fig S4)."

Third, we have clarified that unsupervised clustering and differential expression analysis was used as part of the cell type annotation pipeline. We state in the "*Single nuclei RNA sequencing, processing and cell type annotation*" methods section that: "Cancer and non-cancer cell types were annotated using unbiased clustering and differential expression analysis in combination with application of singleR, ImmClassifier and inferCNV pipeline. Annotations were verified using UMAP analysis and cell type specific marker genes expression (Aran *et al*, 2019; Tickle *et al*, 2019; Becht *et al*, 2018; Liu *et al*, 2021) (annotation workflow detailed in (Griffiths *et al*, 2021, 2022))."

In Griffiths *et al*. (2021) we describe this clustering annotation step in more detail:

“Cell clusters were identified using ‘FindClusters’ method (resolution=0.8) in R package Seurat v3.1.1.9023. Third, each cell cluster was defined as epithelial cells, stromal cells (fibroblasts, endothelial cells) or immune cells (macrophages, T cells, B cells) based on the most frequent cell type annotation by ‘SingleR’.”

Fourth, to further support the attribution of fibroblast EGFR pathway activation and mesenchymal differentiation along the UMAP dimensions, we provide greater description and present additional results of our trajectory-based analyses of the major axes of variation in the fibroblast scRNAseq data. This approach follows the tradeSeq workflow of Van den Berge *et al.* (2020) to discover genes that are non-linearly associated with phenotypic differentiation. This provides a framework to exploit the continuous resolution of scRNAseq phenotypic heterogeneity, to identify gradients of differential expression across UMAP space and to illuminate the underlying biological processes.

In the methods section ‘*Fibroblast phenotype reconstruction*’, we now state:

“The phenotype reflected by each dimension were identified through trajectory-based analysis, to make use of the near continuous resolution of scRNAseq phenotypic heterogeneity. Following the tradeSeq approach (Van den Berge *et al.*, 2020) , generalized additive models were used to identify genes with non-linearly and dynamically (smoothly) changing expression along each axis. Significance of dynamical changes in gene expression were determined by the F value of the predictor smooth term (penalized thin plate regression spline). The 25 most dynamically varying genes were used to interpret the biology of phenotypic gradients. These dynamic changes in expression were verified by examining correlation with ssGSEA pathway scores and through gene overlay of marker genes. The key phenotypes reflected mesenchymal fibroblast differentiation (high hallmark EMT ssGSEA scores, and high collagen, fibronectin and ECM secretion protein expression) (Wu *et al.*, 2020; Glabman *et al.*, 2022) and activation of EGFR driven proliferation (high biocarta EGF and ERK proliferation ssGSEA scores) (Meran *et al.*, 2011; Midgley *et al.*, 2013) (Fig EV3A) (Appendix Fig S3/4).”

In Appendix Fig S3, we show results from the trajectory-based analysis of dynamic (non-linear) gene expression change across UMAP space. The heatmaps show EGFR transcriptional activation across UMAP A and ECM modification gene expression across UMAP B.

Appendix Figure S3) Trajectory-based differential expression analysis results identify the core transcriptional program defining fibroblast differentiation across the UMAP landscape. Genes dynamically expressed across UMAP A axis (left panel) and UMAP B axis (right panel) were identified using a general additive model to test the potentially nonlinear relationships between scaled gene expression ($\log(1+\text{CPM})$) and UMAP coordinates. Heatmap shows the single cell RNAseq gene expression of the top 25 most dynamically changing genes (rows; grouped by hierarchical clustering) that typically showed monotonically increasing or decreasing expression along the UMAP axes. Columns represent single fibroblast cells that are ordered by rank order along a UMAP axis (annotation bars above indicates fibroblast subpopulation cluster annotation obtained from gaussian mixture model clustering with cluster number determined using Bayesian information criteria). Darker+ redder tile coloration signifies higher expression. Genes dynamically changing along UMAP A include EGFR expression, various downstream EGFR response transcription factors (FOS, FOXO3, FOSB, DUSP1, KLF6, NR4A1). Genes dynamically increasing along UMAP B are involved in extracellular matrix formation (collogens: COL1A1, COL1A2, COL12A1, COL11A1, COL10A1; fibronectin: FN1; ITGA11), extracellular matrix protein secretion (POSTN, B2M, MMP11), mesenchymal fibroblast differentiation (INHBA, KIF26B). Genes dynamically downregulated along UMAP B are involved in inhibition of stromal differentiation (NFIA, DCLK1) and regulation of proliferation (ABI3BP, CELF2).

We found that CAMA1 and MCF7 showed inconsistent baseline differences in ERBB signaling between ETR and CRR lineages. In the MCF7 cell line, CRR cells show higher expression of EGFR and ERBB4 but lower expression of ERBB3-4. The ssGSEA pathway analysis showed overall upregulation of the ERBB signaling (e.g. via the GOBP and KEGG ERBB signaling pathways). In contrast, in the CAMA1 cell line, CRR cells showed downregulation of ERBB3-4 genes but increased activity of some ERBB ssGSEA pathways (e.g. GOBP ERBB signaling pathway) but not others (KEGG ERBB signaling pathway).

We have an ongoing project that is examining the dynamical activation/deactivation of ERBB, MAPK, MYC and cell cycle pathways during three weeks of treatment (using time course RNA expression data +/- treatment) in these cell lines. The findings are being written up as a separate manuscript as we see striking cell cycle deactivation and rapid reactivation in resistant lineages, as opposed to sustained deactivation in sensitive cells. Crucially, ERBB inhibitors can maintain durable cell cycle deactivation. We look forward to sharing more of these details shortly. Overall, our findings suggest that the ERBB resistance mechanism can be induced upon endocrine +/- ribociclib treatment.

Comment/Question 3) – *“The Results sections states that “Furthermore, endocrine therapy induced greater phosphorylation of ERBB receptors, showing rapid dose-dependent compensatory activation of ERBB signaling across endocrine and combination ribociclib resistant cell lines.” And the Figure caption reiterates “Broad ERBB activation and phosphorylation across all cancer cell lines under endocrine therapy.” This is not consistently apparent in Fig 2C. Can the authors be more clear about their claim regarding activation of ERBB signaling under endocrine treatment both in the Results section and the figure caption?”*

Response) We thank the reviewer for pointing out the need for clarification of our description regarding the impact of endocrine treatment on ERBB signaling activation. To more appropriately convey the details of the results, we have revised the language relating to this figure in both the main text and figure legend.

In the figure legend, we now provide the following, more detailed, description of the activation of ERBB signaling under endocrine treatment:

“Overall ERBB activation and phosphorylation increased under endocrine therapy across ribociclib resistant (CRR) and sensitive (ETR) cell lines, but with mixed extent in specific cancer cell lines (greater in CAMA-1 and T47D; lesser in MCF-7). Afatinib inhibits phosphorylation and reduces total concentration of ERBB receptors in each of the cancer cells.”

Likewise, we revised the main text to read: “Using Western blot analysis, we confirmed that fulvestrant treatment induced higher levels of ERBB receptors across cancer cell lines, although the extent was much greater in CAMA-1 and T47D cell lines and less in MCF-7 (Fig 2C). Furthermore, endocrine therapy induced greater phosphorylation of ERBB receptors, showing rapid dose-dependent compensatory activation of ERBB

signaling across endocrine and combination ribociclib resistant cell lines (again with more varied activation in MCF-7).”

Comment/Question 4) – *“Fig 4.B -Validation cohort - The results sections states “The increased TGF β 1-3 cancer to fibroblast crosstalk was found in both the discovery and validation cohort tumors.” However there appears to be much lower TGFB1,2,3 signaling upregulation in fibroblasts compared to discovery cohort?”*

Response) The asterisks on figure 4B indicate the significance of the increases in TGFB1-3 signaling in the cohorts. The reviewer is correct that the effect sizes of the increases in signaling are substantially smaller in the validation cohort than in the discovery cohort, but the significant effects in the predicted direction were still found. As mentioned in the manuscript and in our previous response, we hypothesize that some of the diploid cells that were found in patients of the validation cohort and contributed significantly more to the TGFB signaling (versus diploid cells in the discovery cohort) may have been cancerous but lack the distinct copy number alteration that would allow them to be classified as cancer cells. This would explain why there is more diploid epithelial and less cancer cell signaling in the validation cohort. We endeavored not to modify the cell annotation process between cohorts and to have high confidence in our classified cancer cells. We are pleased that we were able to retrieve the quantitatively consistent outcome without needing to modify the computational pipeline, which we feel would have eroded the value of the validation analyses.

Comment/Question 5) – *“Fig 5 - Nice to demonstrate facilitation of tumor+fibroblast relative to tumor alone. Do you have negative controls where you tested tumor + other immune component to show lack of mutualism?”*

Response) Here we have provided the results from applying our mutualism analysis to data collected in the lab from a supplementary 7-day coculture experiment with multiple negative controls (coculturing cancer cells with CD8+ T cells and/or macrophages) (Figure R1 below).

To perform the experiment, we collected immune cells from patient blood samples as follows. Leukocyte Reduction System (LRS) cones were obtained from a healthy blood donor at City of Hope, Duarte, CA under Institutional Review Board (IRB # 17387) approval. Blood from LRS cones was transferred to K₂EDTA blood collection tube (BD Biosciences), and centrifuged for 10 minutes at room temperature and 800 x g. Plasma was removed and buffy coat was collected and diluted to 5mL in 1x PBS without MgCl₂ (Gibco) + 2% hiFBS (heat inactivated Fetal Bovine Serum). Red blood cells were removed by immunomagnetic depletion using 50 μ L of EasySep RBC Depletion Reagent (Stemcell Technologies) per mL of sample according to manufacturer instructions and froze as viable peripheral blood mononuclear cells in 50% RPMI-1640 (Gibco) + 40% heat inactivated FBS (hiFBS, Sigma-Aldrich) + 10% DMSO (Fisher Scientific) and stored in liquid nitrogen vapor phase. Subsequently, CD8+ T cells were isolated from

buffy coat using EasySep Human CD8+ T Cell Isolation Kit (Stemcell Technologies) by immunomagnetic negative selection according to manufacturer instructions. Isolated CD8+ T cells were centrifuged for 5 minutes at room temperature, 300 x g. CD8+ T cells were then resuspended to 0.5e6 cell/mL in RPMI-1640 (Gibco) + 10% hiFBS + 1x antibiotic-antimycotic (Gibco) and stimulated for activation for 4 days supplemented with 20ng/mL IL-2 (Miltenyi Biotec) and CD3/CD28 Dynabeads Human T-Activator (Gibco) in a 6-well tissue culture treated plate (Corning) and maintained in 37°C humidified incubator + 5% CO₂. Prior to co-culture with cancer cells, CD8+ T cells were collected and CD3/CD28 beads removed using DynaMag-15 magnet (Gibco). Purified activated CD8+ T cells were centrifuged at 300 x g 5min, at room temperature and resuspended to > 2.0e6 cell/mL in fresh RPMI-1640 complete culture media. Patient-derived CD14+ monocytes were isolated using a Stemcell CD14 Isolation Kit. Monocytes were differentiated towards an M2 macrophage state by first treating them for 3 days with 2ul/ml M-CSF and then treating them for a further 4 days with M-CSF 20 ng/ml, IL-4 20 ng/ml, IL-13 20 ng/ml and IL-6 20 ng/ml (Biolegend:574802,B368019;574002,B341744;5711102,B262843). Cancer-immune coculture spheroids were plated with 5000 cancer cells (CAMA-1 or MCF7) and CD8+ T cells were added in a (1:1:1) ratio after 24 hours (or with DMSO control addition if cell not involved in this coculture treatment combination). Three replicate spheroids were cultured under 1%FBS and monitored over 7 days on day 0,1,2,3,4 and 7.

We directly applied the existing mutualism quantification analysis to this in vitro spheroid growth data. This showed that, as expected of a negative control, no mutualism was detected under any condition. Instead, we identified significant suppression of growth across cancer cell line under many of the immune coculture combinations.

We state in the methods section '*Quantifying cell abundance and facilitation*' that: "We applied the mutualism quantification analysis to spheroid growth data from cocultures with patient-derived macrophages and CD8+T cells. As expected, these negative control cell types showed no mutualism, significant growth suppression being detected across cancer cell lines in CD8+ T cell cocultures."

quantify the proportional increase (positive values) or decrease (negative values) in cells per cell per hour (change in exponential growth rate) under coculture conditions compared to the monoculture baseline.” In the figure labels, we have indicated the treatment effect on growth rate as having units: Δ cells/cell/hour.

Comment/Question 7) – “In section 5, the authors note that “The model describes the cancer and fibroblast population growth (r_i and K_i represent intrinsic growth rate and carrying capacity of cell type i), with each cell type facilitating the proliferation of the other (β_C = cancer facilitation/fibroblast; β_F = fibroblast facilitation/cancer cells).” which implies that there is both a carrying capacity K for the cancer cells and a separate carrying capacity for the fibroblasts. However, there is only a single K variable listed in Fig. 5E - Can the authors clarify? It would also help if the authors clarified the units and the interpretation of Fig 5E.”

Response) We have added the appropriate subscripts to the equations presented in Fig 5E, with cancer and fibroblast carrying capacities being indicated by K_C and K_F respectively. In the legend, we have clarified that the same value of the carrying capacity was used across both cancer and fibroblast populations. We have also provided x/y label annotations of the units for each of the parameters presented in figure 5E. Units of parameters and their interpretation are provided in the methods section: *Predicting dynamical consequences of a cancer-fibroblast mutualism.*

Comment/Question 8) – “Similar comments for Section 6 and Figure 6 regarding how to interpret growth rate differences as well as the units.”

Response) Above, in response to comment/question 8, we explained how we have clarified the description/interpretation of growth rates in the results. As for Figure 5, we have indicated in Figure 6 that the treatment effect on growth rate have units: Δ cells/cell/hour.

We note that we have provided corresponding fluorescence images of cancer-fibroblast spheroids sizes after 7 days of treatment with DMSO or afatinib (Figure 5C) as a complementary visual way to present the treatment effect sizes. We found this to be a useful way to show drug impacts on spheroid growth.

Comment/Question 9) – “In Section 6, the authors claim that “Under afatinib treatment, cocultured fibroblasts provided some support to accelerate cancer growth of several cell lines compared to afatinib treated cancer monocultures.” In Figure 6, it appears that afatinib has comparable effect on coculture vs. monoculture cancer cells - can the authors clarify?”

Response) We are thankful that the reviewer raised this and have clarified our description of this result. The revised text reflects the reviewers observation that afatinib

removes and systematic coculture benefit. We now state that: “Cocultured cancer growth rate was typically reduced to well below that of the untreated (DMSO) cancer monocultures. Under afatinib treatment, cocultured fibroblasts provided weak support to accelerate cancer growth of several cancer cell lines (CAMA-1 CRR/ETR) but hampered the growth of other cell lines (MCF-7 CRR/ETR) when compared to afatinib treated cancer monocultures. As a result, no significant benefit of fibroblast coculture was detectable across cancer cell lines under afatinib treatment.”

Minor critiques

A) – “Page 1 change “Resistant cancer cells upregulate ERBB growth factor-receptors and stimulate fibroblasts to provide corresponding ligands.””

Response) This section of text now reads exactly as the reviewer suggests.

B) – “Page 3 change “Ecological theory predicts that such oncogenic mutualisms may become stronger under more stressful treatment conditions (Bertness & Callaway, 1994; Hammarlund & Harcombe, 2019; Michalet et al, 2014).””

Response) This section of text reads exactly as the reviewer indicates.

C) – “Page 5 change “Activation of the ERBB signaling pathway was also observed to a lesser extent in diploid epithelial cells of these post treatment tumor samples (Fig EV1/C).””

Response) This section of text now reads: Activation of the ERBB signaling pathway was also observed to a lesser extent in diploid epithelial cells of these post-treatment tumor samples (Fig EV1/C).

D) – “Fig.1 - Upper right claims that “Test results in separate patient cohort”. The wording is potentially confusing as the results are tested in a validation cohort that is from the same clinical study. Recommend changing to “Test results in validation cohort”.”

Response) Following the reviewers suggestion, we have edited the statement in the upper right box to read: “Test results in validation patient cohort”.

E) – “Fig. 1 - Lower panel entitled “Cancer Non-cancer Mutualism”. Would it be more clear to title this section as something like “Test cancer mutualism in vitro”?”

Response) Following the reviewers suggestion, we have edited the left-hand panel title to read: “Test Cancer-Non-cancer Mutualism in vitro”.

Point-by-point responses to reviewers' reports for: "Blocking cancer-fibroblast mutualism inhibits proliferation of endocrine therapy resistant breast cancer"

We are pleased to have fully addressed all reviewers' questions and concerns and are excited to be informed of the pending acceptance. In this document, we address remaining reviewer and then editor comments (italicized) with a point-by-point description (in blue colored text) of how each point has been addressed. Manuscript changes have been highlighted.

REVIEWER COMMENTS

Reviewer #2:

Reviewer #2 summary: "The authors have sufficiently addressed all my concerns."

Response) We thank the reviewer for their time and effort to carefully and constructively examine the manuscript. Acting on their recommendations has clearly strengthened the manuscript.

Reviewer #3:

Reviewer #3 summary: "The authors have fully addressed the questions and incorporated suggestions and this reviewer supports publication without additional changes. Below are some minor wording changes that the authors should incorporate:"

Response) We are pleased to have been able to successfully address their questions and comments and will address their remaining minor wording changes below. We thank the reviewer for once more carefully assessing the text and identifying these sections to clarify.

Minor wording changes

Comment/Question 1) – *"Typo page 3: (Either delete "within" or "in" in this sentence) This approach integrates patient-derived single-cell RNA sequencing to explore acquired resistant phenotypes and intercellular crosstalk within in tumors as patients undergo therapy and utilizes cell-based assays to assess mechanisms of resistance and interactions between cancerous and non-cancerous cells."*

Response) We have removed "in" from this sentence. Thank you for noticing this typo.

Comment/Question 2) – *"Awkward wording page 4: Consistent growth-factor signaling phenotypic dysregulation of resistant cancer cells across treatments and patients was determined using single sample Gene Set Enrichment Analysis (ssGSEA) scores (Liberzon et al, 2011; Hänzelmann et al, 2013) (Fig 1; top right)."*

Response) This sentence has been revised and now reads: "Consistent dysregulation of growth-factor signaling in resistant cancer cells during treatments across patients was determined using single sample Gene Set Enrichment Analysis (ssGSEA) scores (Liberzon et al, 2011; Hänzelmann et al, 2013) (Fig 1; top right)."

Comment/Question 3) – “Page 11: (Change "support" to "supported") Further, identification of physiologically relevant resistance traits can be supported by integrated analyses of patient data and experimental systems.”

Response) We have changed "support" to "supported". Thank you for noticing this typo.

EDITOR COMMENTS

1) We require an institutional email address in the manuscript and in our submission system for corresponding authors - currently Jason Griffiths does not have an institutional email address listed. It is an option to keep an additional/non-institutional email as a secondary email address. This has recently been made mandatory for all EMBO Press journals, and is included as a requirement upon submission (for more information please see the authorship guidelines in our guide to authors:

<https://www.embopress.org/page/journal/17574684/authorguide>).

We can also link your ORCID ID in our system to a profile displaying both email addresses.

Response) We have provided the following institutional email address for Jason Griffiths: jgriffiths@coh.org

2) In the main manuscript file, please include keywords to max. 5.

Response) We have added a keywords section.

3) Please combine the information in the Code Availability section into the Data availability section and format according to the example below:

"The datasets and computer code produced in this study are available in the following databases:

- Chip-Seq data: Gene Expression Omnibus GSE46748

(<https://www.ncbi.nlm.nih.gov/geo/query/acc.cgi?acc=GSE46748>)

- Modeling computer scripts: GitHub

(<https://github.com/SysBioChalmers/GECKO/releases/tag/v1.0>)

- [data type]: [full name of the resource] [accession number/identifier] ([doi or URL or identifiers.org/DATABASE:ACCESSION])"

Response) Code availability combined into the data availability section. This has been reformatted following the example provided. It now reads:

The datasets and computer code produced in this study are available in the following databases:

- Pre-processed single cell RNA-seq gene expression data and relevant metadata: GEO (the Gene Expression Omnibus) under accession code GSE211434

(<https://www.ncbi.nlm.nih.gov/geo/query/acc.cgi?acc=GSE211434>)

- Custom code used in analyses and to produce Figures 1-6: GitHub

(https://github.com/U54Bioinformatics/FELINE_project/tree/master/FELINE_ERBB_fibroblast_facilitation)

- *As indicated in the author checklist, please ensure that a statement on whether or not blinding was done is included in the Methods even if no blinding was done.*

Response) We have added statements to the methods indicating that:

- “Sample procurement conformed to the principles set out in the Department of Health and Human Services Belmont Report.”
- “Blinding was achieved through placebo administration.”

9) *Please remove the Reagents and Tools Table in the Methods section of the main manuscript file and upload it as a separate file choosing the file type "Reagent Table".*

Response) The “Reagents and Tools Table” has been removed from the main manuscript methods and uploaded as a separate file.

10) *Please remove the Statement of Significance from the manuscript, as this is not one of the sections included in our paper format.*

Response) The “Statement of Significance” has been removed.

11) *Please place individual sections of the manuscript in the following order: Title page - Abstract & Keywords - Introduction - Results - Discussion - Methods - Data Availability - Acknowledgements - Disclosure and Competing Interests Statement - References - Figure Legends - Expanded View Figure Legends.*

Response) Individual manuscript sections have been ordered as requested.

12) *For the figures and figure legends, please take care of the following:*

- *Please make sure to update the callouts of all figures in the main manuscript text. Currently there is a callout for a Fig S4, but no such file uploaded to our submission system.*

Response) We have searched the main text and there is no call out for “Fig S4”. Call outs do exist for “Appendix Fig S4” and this is uploaded in the Appendix.

- *Please note that the legend for figure EV3 D is missing in the manuscript. This needs to be rectified.*

Response) Apologies that the label “D” within the figure EV3 legend text was removed in error. The legend is present, and we have rectified the issue by adding back the label. The legend for figure EV3 D now reads:

“D) Mesenchymal fibroblast differentiation is associated with increased expression of fibronectin and various collagens which drive tumor fibrosis and cancer cell proliferation through extracellular matrix modification. Fibroblasts were finely classified by differentiation state across the phenotypic landscape (mesenchymal fibroblast differentiation: UMAP B) and the mean (points) and standard error (error bars) of collagen and fibronectin gene expression (color) is shown for each level of differentiation. The distribution of fibroblast differentiation states was discretized into 36

- Please note that the box plots need to be defined in terms of minima, maxima in the legends of figures 2A, D; 3B, C; 4B, C, D; 5A, B, C, D; 6A, B; EV1 A, C, D; EV2 C, EV4 D.

Response) We have defined the whiskers (minima/maxima) as 1.5*inter-quartile range throughout the manuscript. This is noted in the legends of figures 2A, 2D, 3B, 3C, 4B, 4C, 4D, 5A, 5B, 5C, 5D, 6A, 6B, EV1A, EV1C, EV1D, EV2C, and EV4D.

For example, in figure 2A we have stated that: “Box elements represent median (center line), upper/lower quartiles (hinges), 1.5*inter-quartile range (whiskers=minima/maxima) of ERBB activity within a tumor sample.”

- Please note that information related to n is missing in the legends of figures 2D, 3B; EV1A, C; EV3 D; EV4C; EV5 A.

Response) We have provided n values in the legend for figures: 2D, 3B, EV1A, EV1C, EV3D, EV4C and EV5A.

For example in figure 2D, we state: “Sample size: Discovery:: 32 patient tumors (n=51 pre-/post-treatment samples), Validation:: 28 patient tumors (n=54 pre-/post-treatment samples).”

Likewise for figure 3, we state: “Sample sizes in A-C: Discovery:: 76246 tumor-derived single cells (47275 pre-treatment, 28971 post treatment) from 20 patients with paired pre- and post-treatment samples (n=40), Validation:: 163245 tumor-derived single cells (96659 pre-treatment, 66586 post-treatment) from 28 patients with paired pre- and post-treatment samples (n=56).”

- Please note that the error bars are not defined in legends of figures EV3D, EV4B, C.

Response) We have defined error bars in figure legends: EV3D, EV4B, EV4C.

For all figures, error bars have been defined as showing standard error around mean.

- Please note that scale bar and its definition are missing for figures 3D, EV4 A.

Response) We have added scale bars in figures: 3D, and EV4A.

13) Tables: Tables EV1-EV4 should be uploaded as individual Excel files and the legends should be removed from the main file. Additionally there are callouts in the manuscript for Table S1/S2, but no such files have been uploaded to your submission - please double check these callouts.

Response) Tables EV1-EV4 have been uploaded as individual Excel files with legend in the first row of these files instead of in the main text. Callouts to Table S1/S2 has been replaced with Table EV1/EV2.

14) Appendix file: Please include page numbers for the listed items in the Table of Contents; also the nomenclature should be Appendix Figure S1-S2 in figure legends.

Response) Page numbers included in Appendix and figure legend nomenclature corrected.

15) *Synopsis text: Please shorten the standfirst to a maximum of 300 characters including spaces. Additionally, please shorten the 4th bullet point to a maximum of 30 words.*

Response) Synopsis shortened to under 300 words and 4th bullet reduced to 23 words.

16) *As part of the EMBO Publications transparent editorial process initiative (see our policy here: https://www.embopress.org/transparent-process#Review_Process), Molecular Systems Biology will publish online a Peer Review File (PRF) to accompany accepted manuscripts. This file will be published in conjunction with your paper and will include the anonymous referee reports, your point-by-point response and all pertinent correspondence relating to the manuscript. Let us know whether you agree with the publication of the PRF and as here, if you want to remove or not any figures from it prior to publication. Please note that the Authors checklist will be published at the end of the PRF.*

Response) We agree with the publication of the PRF as is provided here.

17) *After your paper is published, we will promote it on social media. If you have any handles or hashtags for Bluesky you would like included, please let us know.*

Response) Thanks for the promotion. Not handles/hashtags for Bluesky to report.

10th Apr 2025

Manuscript number: MSB-2024-12513RRR

Title: Blocking cancer-fibroblast mutualism inhibits proliferation of endocrine resistant breast cancer.

Dear Dr Griffiths,

Thank you again for sending us your revised manuscript. We are now satisfied with the modifications made and I am pleased to inform you that your paper has been accepted for publication.

Yours sincerely,

Sincerely,

Poonam Bheda, PhD
Scientific Editor
Molecular Systems Biology
